# REVISITING AND EXPANDING TARGETED UNIVERSAL ADVERSARIAL PERTURBATIONS

## ABSTRACT

Universal adversarial perturbations (UAPs) have deepened the vulnerability concern of Deep Neural Networks (DNNs) after the initial intriguing discovery of vanilla single-model-single-image adversarial attacks. However, the landscape of UAPs has not been thoroughly investigated. In this paper, we revisit and expand UAPs for white-box targeted attacks along three axes simultaneously: the model-axis, the data-axis, and the target-axis. For the target-axis, we adopt the most aggressive ordered top-$K$ attack protocol ($K \geq 1$) to expand the traditional top-1 attack setting in the prior art of learning UAPs. Our proposed method is thus dubbed as **AllAttacK**. In implementation, our AllAttacK is built on two state-of-the-art single-model-single-image ordered top-$K$ attack methods, the KL divergence based adversarial distillation method and the more recently proposed quadratic programming based method. We propose a simple yet effective joint mini-data-batch and mini-model-batch optimization strategy in learning UAPs for a large number of models (e.g., up to 18 disparate DNNs) and a large number of images (e.g., 1000 images). We test our AllAttacK on the ImageNet-1k classification task using an ensemble of disparate models such as Convolutional Neural Networks and their adversarially-robustified versions, Vision Transformers, CLIP vision encoders, and MLP-Mixers. Our learned AllAttacK perturbations are doubly transferable across training and testing models, and across training and testing images, and they also show intriguing yet sensible looking.

## 1 INTRODUCTION

Visual perception is robust with human vision, and is aimed to be similarly, if not more, robust with computer vision (Palmer, 1999). Computer vision has witnessed remarkable progress by end-to-end representation learning using Deep Neural Networks (DNNs) (LeCun et al., 1998; Krizhevsky et al., 2012; He et al., 2016; Huang et al., 2017; Dosovitskiy et al., 2020). However, adversarial attacks can easily fool well trained image classification DNNs to classify a dog image as a cat by adding visually-imperceptible perturbations (Nguyen et al., 2015; Szegedy et al., 2014; Athalye & Sutskever, 2017; Carlini & Wagner, 2016; Goodfellow et al., 2015; Kannan et al., 2018; Madry et al., 2017; Xie et al., 2019; Madry et al., 2018). Initially perceived as mere anomalies, adversarial attacks have rapidly evolved, posing increasingly intricate challenges (Geirhos et al., 2020) for the reliability and trustworthiness of AI systems, especially in high-stake applications.

Among many other aspects, *Universal Adversarial Perturbations (UAPs) that are often quasi-imperceptible have introduced even deeper troubles for DNNs since they are doubly transferable across DNNs and images.* UAPs have been studied both for un-targeted top-1 attacks (Moosavi-Dezfooli et al., 2017; Shafahi et al., 2020) and targeted top-1 attacks (Liu et al., 2016), but tested with convolutional neural networks only including CaffeNet (Jia et al., 2014), VGGNets (Chatfield et al., 2014; Simonyan & Zisserman, 2015), GoogLeNet (Szegedy et al., 2015) and ResNets (He et al., 2016). With the recent development of DNNs with new architectures such as Vision Transformers (Dosovitskiy et al., 2020), ConvNeXt Woo et al. (2023) and MLP-Mixers (Tolstikhin et al., 2021), and with new and more powerful training recipes such as the contrastive language-image pretraining (CLIP) (Radford et al., 2021) and further combined with masked image modeling (MIM) as in the EVA2 model (Fang et al., 2023), it is unclear whether targeted UAPs can retain their attacking power for ensembles of those disparate DNNs, as well as adversarially-robustified counterparts (Croce et al., 2020).

Figure 1: Illustration of the proposed AllAttacK. See text for details.

In the meanwhile, with respect to the number of attack targets beyond the conventional top-1 setting, there are three types of settings with increasing difficulty levels: i) *Untargeted Top-$K$ Adversarial Attacks* (Easiest): Ground-truth labels shouldn't be in the top-$K$ classes, Top-$K$ classes can be anything but ground truth. This is often achievable as by-product via existing un-targeted attack methods as pointed out in (Zhang & Wu, 2020). ii) *Unordered Top-$K$ Targeted Adversarial Attacks* (Hu et al., 2021; Zhang et al., 2022; Tursynbek et al., 2022; Kumano et al., 2022; **?**) (Harder): They provide specific target top-$K$ classes that should be in the top-$K$ predictions after the attack but no particular order of appearance is enforced as long as each target class is somewhere in the top-$K$ predictions. iii) *Ordered Top-$K$ Targeted Adversarial Attacks* (Zhang & Wu, 2020; Paniagua et al., 2023) (Hardest): They provide specific targeted top-$K$ classes in order and the top-$K$ predicted classes after attack must match this exact order. (Liu et al., 2016) has shown that transferable targeted top-1 attacks across images and/or models are much harder to learn. It remains unclear whether UAPs can achieve more aggressive attack objectives, e.g., ordered top-$K$ UAPs.

In this paper, we provide affirmative answers to the above two questions by learning **universal ordered top-$K$ perturbations that are doubly transferable across images and models consisting of disparate types of DNNs.** Our proposed method is dubbed as **AllAttac$K$**, as illustrated in Fig. 1. More specifically, we revisit and expand conventional UAPs (Moosavi-Dezfooli et al., 2017; Shafahi et al., 2020; Liu et al., 2016) simultaneously along three axes:

- *The model-axis:* How many different types of DNNs (e.g., convolutional neural networks, Transformer models and all-MLP models), and how many different models of each type can be attacked, simultaneously? Furthermore, can perturbations learned from an ensemble of training models generalize to unseen models that are of very different architectures than those in training?
- *The data-axis:* How many of training images (that are used in the optimization of learning the shared adversarial perturbation) can be attacked, and how many unseen images can the same perturbation transfer to, simultaneously?
- *The target-axis:* How many top-$K$ targets can be attacked, and can they be attacked with respect to any given orders? For example, we may want to see if a well-trained image classification DNN can be fooled to misclassify a dog image, not only just with cat as its top-1 prediction, but also, e.g., with [cat, car, fish] as its top-3 predictions with the exact given order.

Seeking quantitative analyses of AllAttac$K$ will facilitate us better understanding the adversarial vulnerability at the fundamental level and enable us to re-assess its severity considering that DNNs increasingly permeate various facets of daily life, from enhancing user experience on digital platforms to making critical decisions in autonomous vehicles. As we shall show, the severity is observed to be high. Fig. 2 shows qualitative examples of learned ordered top-$K$ UAPs by our proposed AllAttac$K$.

**Our Contributions.** This paper makes two main contributions to the field of learning white-box targeted adversarial attacks: (i) It presents, to our knowledge, the first large-scale study of learning UAPs that are both model-agnostic (up to 18 disparate DNNs in training) and image-agnostic (at the ImageNet-1k scale), with strong results obtained. (ii) It proposes two optimization formulations in learning AllAttacK, built on previous single-model-single-image ordered Top-$K$ attack work, with a proposed stochastic mini-data-batch and mini-model-batch optimization strategy for practicality and generalizability.

## 2 APPROACH

### 2.1 PROBLEM FORMULATION OF ALLATTACK

We consider image classification with the label set $\mathcal{Y}$ (e.g., 1000 classes in ImageNet (Russakovsky et al., 2015)). Let $F(\cdot)$ be a DNN (e.g., ResNet-50 (He et al., 2016)) trained on a dataset (e.g., the

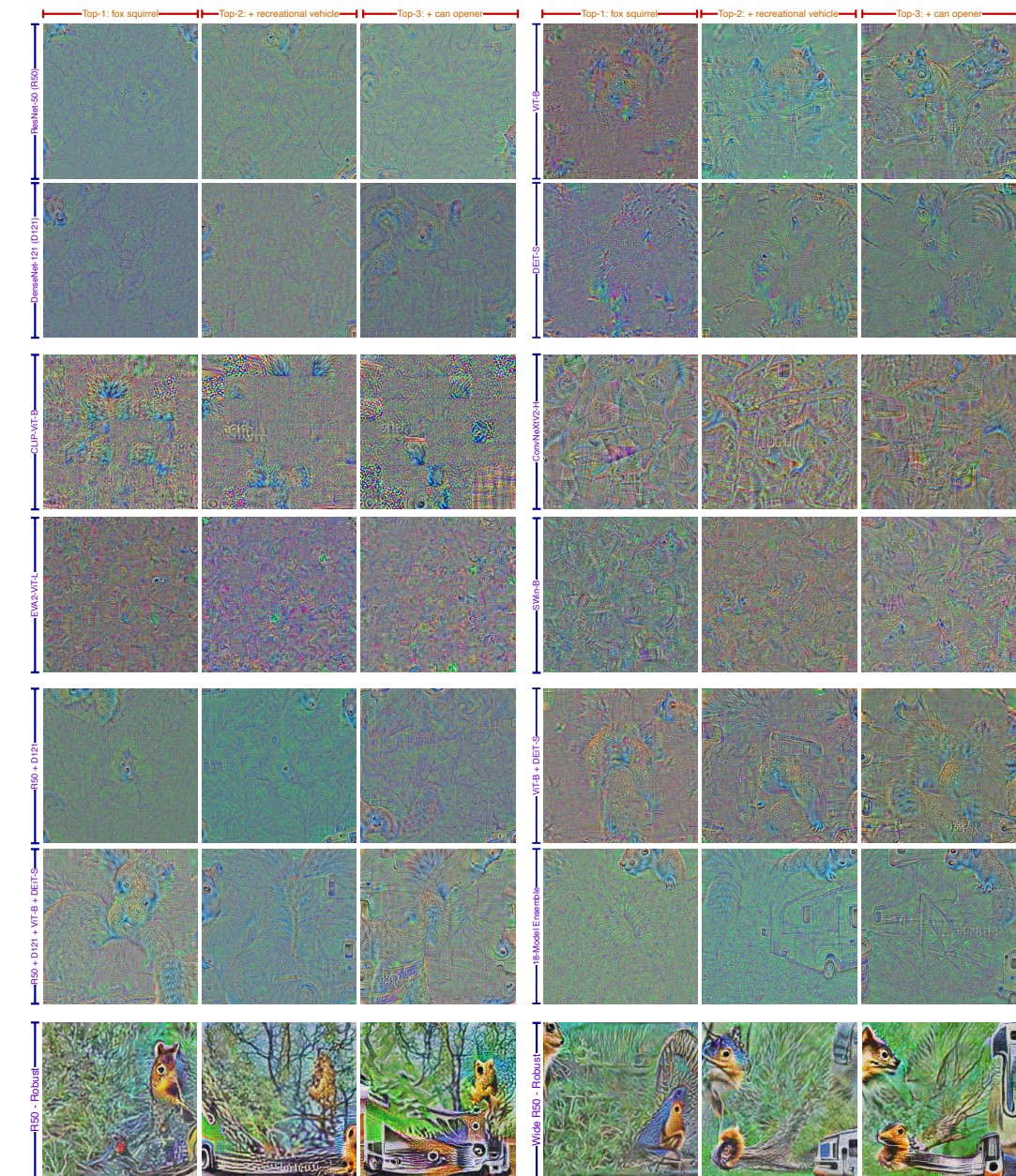

Figure 2: Examples of ordered top-$K$ UAPs leaned by our proposed AllAttacK using the quadratic programming formulation for $K = 1, 2, 3$ with randomly sampled targets [`fox squirrel`, `recreational vehicle`, `can opener`] sequentially added using 1000 ImageNet-1k `train` images as the training set and 1000 ImageNet-1k `val` images as the test set. The last row shows UAPs learned for two robust ResNet-50 models (Engstrom et al., 2019; Salman et al., 2020) sourced from the RobustBench (Croce et al., 2020). For both the multiple-model-multiple-image UAPs (e.g., the 18-model ensemble in the last 2nd row) and the single-robustified-model UAPs, we can observe emergent sense-making appearance in the learned UAPs. See text for details.

ImageNet), which consists of the feature backbone $f_\theta(\cdot) \in R^D$ transforming a raw data sample into a $D$-dim feature space (where $\theta$ collects all backbone parameters), and the linear head classifier $c(\cdot; W, b) \in R^{|\mathcal{Y}|}$ computing the logits (where $W \in R^{D \times |\mathcal{Y}|}$ and $b \in R^{|\mathcal{Y}|}$). For a data $x$, we have,

$$\text{Logits: } F_\Theta(x) = c(f(x; \theta); W, b) = f(x; \theta) \cdot W + b, \tag{1}$$

$$\text{Probabilities: } \hat{P}_\Theta(x) = \text{Softmax}(F_\Theta(x)), \tag{2}$$

$$\text{Sorted Label Indexes: } \hat{Y}_\Theta(x) = \arg \text{sort}(\hat{P}_\Theta(x)), \tag{3}$$

where $\Theta = (\theta, W, b)$, and $\hat{Y}_\Theta(x)$ are the predicted class indexes in the descending order of predicted probabilities (e.g., the top-1 prediction $\hat{Y}_\Theta(x)_1 = \arg\max_{y \in \mathcal{Y}} \hat{P}_\Theta(x)$). Our proposed AllAttacK aims to address the challenges along the three axes as follows,

- **The Model Axis.** Denote by $\mathcal{M}^{train} = \{F_{\Theta_1}(\cdot), \cdots, F_{\Theta_S}(\cdot)\}$ an ensemble of $S$ DNNs used in training AllAttacK where $\Theta_s = (\theta_s, W_s, b_s)$ are the parameters, and by $\mathcal{M}^{test} = \{F_{\Psi_1}(\cdot), \cdots, F_{\Psi_U}(\cdot)\}$ an ensemble of $U$ testing DNNs unseen in training, some of which have very different architectures than those in training (where we use $\Psi_u = (\theta_u, W_u, b_u)$ for notation clarity between training models and testing models). We have $\mathcal{M}^{train} \cap \mathcal{M}^{test} = \emptyset$.
- **The Data Axis.** Denote by $\mathcal{D}^{train}$ and $\mathcal{D}^{test}$ the training and testing sets for AllAttacK. We have $\mathcal{D}^{train} \cap \mathcal{D}^{test} = \emptyset$. In our experiments, $\mathcal{D}^{train}$ is sampled from the ImageNet-1k `train` set each sample of which can be correctly classified by all the DNNs in $\mathcal{M}^{train} \cup \mathcal{M}^{test}$. $\mathcal{D}^{test}$ is sampled from the ImageNet-1k `validation` set each sample of which is not only unseen in training AllAttacK, but also unseen by all the DNNs in their training stages if they are trained from scratch on ImageNet-1k. Similarly, we ensure images in $\mathcal{D}^{test}$ can be correctly classified by all DNNs. More specifically, we sample one image per category for both $\mathcal{D}^{train}$ and $\mathcal{D}^{test}$, resulting 1000 images for each in ImageNet-1k.
- **The Target Axis.** Denote by $\mathcal{T} = [t_1, \cdots, t_K]$ a list of ordered top-$K$ adversarial targets sampled from $\mathcal{Y}$, where $t_k \in \mathcal{Y}$. Consider the predictions of clean images, training and testing, by all DNNs, $\hat{Y}_\omega(x_i), \forall \omega \in [\Theta_1, \cdots, \Theta_S] \cup [\Psi_1, \cdots, \Psi_U]$ and $\forall x_i \in \mathcal{D}^{train} \cup \mathcal{D}^{test}$, we verify that the sampled $\mathcal{T}$ is not a segment of any of them, i.e., $\mathcal{T} \not\subset \hat{Y}_\omega(x_i)$, for $K \geq 3$.

**The objective of our AllAttacK** is, for a given list of ordered top-$K$ targets $\mathcal{T}$, to learn a single universal perturbation $\mathcal{P}(\mathcal{T}) \in [0, 1]^{3 \times H \times W}$ with as small as possible $\ell_2$ energy (to be visually imperceptible) using the ensemble of training DNNs $\mathcal{M}^{train}$ on the training dataset $\mathcal{D}^{train}$, such that we can not only attack as many as possible images $x_i \in \mathcal{D}^{train}$ for each DNN $\Theta_s \in \mathcal{M}^{train}$,

$$\hat{Y}_{\Theta_s}(\text{Clamp}(x_i + \mathcal{P}(\mathcal{T})))_{1:K} == \mathcal{T}, \tag{4}$$

but also generalize to attack images $\mathbf{x}_j \in \mathcal{D}^{test}$ and also for each DNN $\Psi_u \in \mathcal{M}^{test}$,

$$\hat{Y}_{\Theta_s}(\text{Clamp}(\mathbf{x}_j + \mathcal{P}(\mathcal{T})))_{1:K} == \mathcal{T}, \tag{5}$$

$$\hat{Y}_{\Psi_u}(\text{Clamp}(\mathbf{x}_j + \mathcal{P}(\mathcal{T})))_{1:K} == \mathcal{T}, \tag{6}$$

where $\text{Clamp}(z)$ is to clamp $z$ to $[0, 1]$ in an element-wise way. The accuracy of AllAttacK will be evaluated by the attack success rates (ASRs) on the training and testing datasets for each DNN, e.g.,

$$\text{ASR}(\Psi_u, \mathcal{D}^{test}) = |\{\mathbf{x}_j \in \mathcal{D}^{test} | \text{Eqn. 6 satisfied}\}| / |\mathcal{D}^{test}|. \tag{7}$$

Existing adversarial attack settings could thus be treated as special cases of our AllAttacK in a straightforward way, e.g., the most widely studied model- and instance-specific ordered Top-$K$ ($K \geq 1$) targeted attacks (Zhang & Wu, 2020; Paniagua et al., 2023), for which we have $\mathcal{M}^{train} = \{F_\Theta(\cdot)\}$ and $\mathcal{M}^{test} = \emptyset$, $\mathcal{D}^{train} = \{(x, y)\}$ and $\mathcal{D}^{test} = \emptyset$, and $\mathcal{T} = \{t_1, \cdots, t_K\}$. Typically, the learned perturbation will not be generalizable to other models and/or images.

**The Challenge of AllAttacK.** To put in other words, the goal of AllAttacK is to seek a stationary point $\mathcal{P}$ in the data space, which once added to an input clean image can "shut it off" and "steer it towards the adversarial targets $\mathcal{T}$" for any training or testing DNNs, no matter what the original top-$K$ predictions made by the DNNs are for the clean image. The existence of such universal perturbation clearly shows that those DNNs might still have "shallow and fragile understanding" of the structure of the data space. As we observed in experiments, the learned universal top-K perturbations alone indeed can fool most of DNNs (see examples in supplementary material).

## 2.2 LEARNING ALLATTACK

With the above definition of AllAttacK, learning a universal top-K perturbation $\mathcal{P}(\mathcal{T})$ can be cast as a vanilla constrained optimization,

$$\underset{\mathcal{P}}{\text{minimize}} \quad \|\mathcal{P}(\mathcal{T})\|_2, \tag{8}$$

$$\text{subject to} \quad t_k = \hat{Y}_{\Theta_s}(x_i')_k,$$

$$x_i' = \text{Clamp}(x_i + \mathcal{P}(\mathcal{T})), \quad \forall t_k \in \mathcal{T}, \forall x_i \in \mathcal{D}^{train}, \forall \Theta_s \in \mathcal{M}^{train},$$

which can not be optimized directly due to the highly-nonlinear DNNs in the constraints (the first one). We resort to re-formulate the constrained optimization problem in two ways.

### 2.2.1 ALLATTACK VIA MINIMIZING A SURROGATE LOSS FUNCTION

To reformulate Eqn. 8 as an unconstrained optimization problem, we seek some surrogate loss functions, $\mathcal{L}(x'_i; \Theta_s)$, such that the first constraint $t_k = \hat{Y}_{\Theta_s}(x'_i)_k$ is satisfied if and only if $\mathcal{L}(x'_i; \Theta_s) \leq 0$,

$$\underset{\mathcal{P}}{\text{minimize}} \quad \|\mathcal{P}(\mathcal{T})\|_2 + \lambda \cdot \frac{1}{S \cdot I} \sum_{i=1}^{I} \sum_{s=1}^{S} \mathcal{L}(x'_i; \Theta_s), \tag{9}$$

$$\text{subject to} \quad x'_i = \text{Clamp}(x_i + \mathcal{P}(\mathcal{T})), \quad \forall t_k \in \mathcal{T}, \forall x_i \in \mathcal{D}^{train}, \forall \Theta_s \in \mathcal{M}^{train},$$

where the constraints will be easily satisfied via the clamping operation, leading to an unconstrained optimization problem in practice. $\lambda$ is a trade-off parameter in optimization controlling the energy of the learned perturbation and the ASR. In this paper, we build on the adversarial distillation (AD) loss function proposed in (Zhang & Wu, 2020), which has shown state-of-the-art performance in learning model-/instance-specific top-$K$ targeted attacks under the unconstrained optimization formulation. The AD loss function is based on the Kullback-Leiber (KL) divergence between the predicted probability distribution $\hat{P}_{\Theta_s}(x'_i)$ and a top-down designed target distribution $P^{AD}(\mathcal{T})$,

$$\mathcal{L}(x'_i; \Theta_s) = \text{KL}(\hat{P}_{\Theta_s}(x'_i) \| P^{AD}(\mathcal{T})), \tag{10}$$

where $P^{AD}(\mathcal{T})$ maintains the ordered top-$K$ targets $\mathcal{T}$, $P^{AD}(\mathcal{T})_{t_k} > P^{AD}(\mathcal{T})_{t_l}, \forall k < l$, and is designed by accounting for the label distance between labels using the Glove embedding (Pennington et al., 2014). Please refer to (Zhang & Wu, 2020) for details. In Eqn. 10, $\text{KL}(\hat{P}_{\Theta_s}(x'_i) \| P^{AD}(\mathcal{T})) \geq 0$, and it equals 0 if and only if the two distributions exactly match $\hat{P}_{\Theta_s}(x'_i) == P^{AD}(\mathcal{T})$.

### 2.2.2 ALLATTACK VIA A QUADRATIC PROGRAMMING FORMULATION

In the surrogate KL-divergence loss function (Eqn. 10), the design of $P^{AD}(\mathcal{T})$ has more than needed information, that is the probability differences between different categories, in addition to maintaining the top-$K$ order of targets. As pointed by a recently proposed QuadAttacK (Paniagua et al., 2023) method, eliminating those unnecessary constraints and directly maintaining the order of the top-$K$ targets as linear constraints facilitate a Quadratic Programming (QP) solution with significantly better performance in learning model-/instance-specific top-$K$ targeted attacks. We also build on the QuadAttacK in solving our AllAttacK.

Consider the QuadAttacK (Paniagua et al., 2023) for a single model and a single instance, $F_\Theta(x) = f_\theta(x) \cdot W + b$, the key is to formulate the learning of perturbations in two steps. It first learns the perturbation in the feature embedding space,

$$\underset{z}{\text{minimize}} \|z - f_\theta(x')\|_2, \tag{11}$$

$$\text{subject to} \quad l_{t_k} > l_{t_{k+1}}, \quad \forall k \in [1, K-1], \quad t_k \in \mathcal{T}$$
$$l_{t_K} > l_j, \quad \forall j \in \mathcal{Y} \setminus \mathcal{T}, \quad t_K \in \mathcal{T},$$
$$l = z \cdot W + b,$$

where $x' = x + \mathcal{P}(\mathcal{T})$ is the current perturbed image. Eqn. 11 can be solved by a differentiable QP solver (Amos & Kolter, 2017). With the QP solution $z^*$ of Eqn. 11, we can compute the updated perturbation in the image space via an one-step back-propagation,

$$\mathcal{P}^* = \mathcal{P}(\mathcal{T}) - \gamma \cdot \frac{\partial}{\partial \mathcal{P}}(\lambda \cdot \|z^* - f_\theta(x')\|_2 + \|\mathcal{P}\|_2), \tag{12}$$

$$\mathcal{P}(\mathcal{T}) = \text{Clamp}(x + \mathcal{P}^*) - x, \tag{13}$$

where $\gamma$ is the learning rate. Please refer to (Paniagua et al., 2023) for more details.

Built on QuadAttacK (Paniagua et al., 2023), given the current perturbation $\mathcal{P}(\mathcal{T})$, our AllAttacK is learned by first solving,

$$\underset{z_{i,s}}{\text{minimize}} \frac{1}{S \cdot I} \sum_{i=1}^{I} \sum_{s=1}^{S} \frac{1}{\sqrt{D_s}} \cdot \|z_{i,s} - f_{\theta_s}(x'_i)\|_2, \tag{14}$$

$$\text{subject to} \quad l_{t_k}^{i,s} > l_{t_{k+1}}^{i,s}, \quad \forall k \in [1, K-1], \quad t_k \in \mathcal{T}$$
$$l_{t_K}^{i,s} > l_j^{i,s}, \quad \forall j \in \mathcal{Y} \setminus \mathcal{T}, \quad t_K \in \mathcal{T},$$
$$l^{i,s} = z_{i,s} \cdot W_s + b_s, \quad \forall(\theta_s, W_s, b_s) \in \mathcal{M}^{train},$$
$$x'_i = \text{Clamp}(x_i + \mathcal{P}(\mathcal{T})), \quad \forall x_i \in \mathcal{D}^{train},$$

where $D_s$ is the feature dimension of a DNN $f_{\theta_s}(\cdot)$, and $\frac{1}{\sqrt{D_s}}$ is introduced to normalize the $\ell_2$ distances which exhibit large variations among different DNNs.

Similarly, with the optimized $z_{i,s}^*$, we update the universal perturbation by,

$$\mathcal{P}(\mathcal{T}) \Leftarrow \mathcal{P}(\mathcal{T}) - \gamma \cdot \frac{\partial}{\partial \mathcal{P}} \left( \frac{\lambda}{S \cdot I} \sum_{i,s} \frac{||z_{i,s}^* - f_{\theta_s}(x_i + \mathcal{P}(\mathcal{T}))||_2}{\sqrt{D_s}} + ||\mathcal{P}||_2 \right), \tag{15}$$

which does not use the clamping as in Eqn. 13 due to a set of images and a set of models involved.

### 2.2.3 Learning via Stochastic Mini-Batch and Mini-Model

In practice, when the training dataset $\mathcal{D}^{train}$ and/or the training model ensemble $\mathcal{M}^{train}$ are large, we can not afford the full-batch optimization, even for a single large model, due to the GPU memory constraint. To handle this, we resort to stochastic mini-batch and mini-model learning. During each iteration in the optimization, we sample a mini-batch of training images with a predefined batch size (e.g., 64), and sample a number of models (e.g., 4) if all the models in $\mathcal{M}^{train}$ can not be loaded (due to GPU memory constraints).

At a first glance, since our goal is to learn a single perturbation for all images and all models, the practicality-enforced stochastic optimization strategy seems counter-intuitive. During the learning of AllAttacK, the single perturbation is the only "model parameters" to be estimated. Similar to how a randomly-initialized DNN can be successfully trained from scratch using mini-batch stochastic gradient descent, it actually makes sense to learn the single perturbation using stochastic optimization. The interesting aspect of AllAttacK is a learned single universal perturbation can obtain the model-agnosticity (across disparate training models and unseen testing models).

## 3 Experiments

In this section, we test our proposed AllAttacK in the ImageNet-1k benchmark (Russakovsky et al., 2015) with strong performance obtained. We randomly sample one image per class in the ImageNet `train` set and `val` set, as the training set $\mathcal{D}^{train}$ and the testing set $\mathcal{D}^{test}$ respectively. So, there are 1000 images sampled for both training and testing. See Appendix A.2 for optimization details. **Our PyTorch code will be released.**

**Metrics.** We evaluate a learned universal perturbation based on the ASR (e.g., Eqn. 7) and its energies in terms of $\ell_1, \ell_2$ and $\ell_\infty$ norms. For each given number of targets, $K$ (e.g., $K = 1, \cdots, 6$), we randomly sample 5 lists of ordered top-$K$ targets. For each given list of ordered top-$K$ targets $\mathcal{T}$, we learn the universal perturbation $\mathcal{P}(\mathcal{T})$ using the two optimization for-

Table 1: The `mean` ASRs and $\ell_2$ of learned AllAttacK perturbations across 5 runs under the single-model and image-agnostic setting. The four models are tested individually. The KL loss function (Eqn. 9) and the QP method (Eqn. 14) are tested and compared.

| Protocol | Dataset | Method | ResNet-50 | | DenseNet-121 | | DEiT-S | | ViT-B | |
|---|---|---|---|---|---|---|---|---|---|---|
| | | | ASR↑ | $\ell_2\downarrow$ | ASR↑ | $\ell_2\downarrow$ | ASR↑ | $\ell_2\downarrow$ | ASR↑ | $\ell_2\downarrow$ |
| Top-6 | $\mathcal{D}^{test}$ | $KL$ | 0.0008 | **25.88** | 0.0004 | **19.55** | 0.0110 | **27.78** | 0.0104 | **27.14** |
| | | $QP$ | **0.2360** | 56.28 | **0.3800** | 56.36 | **0.3986** | 62.38 | **0.5980** | 71.38 |
| | $\mathcal{D}^{train}$ | $KL$ | 0.0030 | **25.44** | 0.0010 | **19.50** | 0.0282 | **27.76** | 0.0256 | **27.13** |
| | | $QP$ | **0.5740** | 56.15 | **0.6444** | 56.49 | **0.6734** | 62.46 | **0.7764** | 71.82 |
| Top-5 | $\mathcal{D}^{test}$ | $KL$ | 0.0204 | **31.01** | 0.0226 | **29.20** | 0.0328 | 25.42 | 0.0324 | **26.60** |
| | | $QP$ | **0.3452** | 45.28 | **0.4652** | 45.89 | **0.5770** | 55.93 | **0.8000** | 61.81 |
| | $\mathcal{D}^{train}$ | $KL$ | 0.0592 | **30.76** | 0.0680 | **28.98** | 0.0714 | **25.48** | 0.0678 | **26.65** |
| | | $QP$ | **0.7416** | 45.17 | **0.7318** | 46.11 | **0.8600** | 56.15 | **0.9594** | 62.33 |
| Top-4 | $\mathcal{D}^{test}$ | $KL$ | 0.0544 | **30.40** | 0.0734 | **26.69** | 0.1446 | **26.76** | 0.1210 | **25.40** |
| | | $QP$ | **0.4050** | 34.43 | **0.5258** | 36.63 | **0.6712** | 40.98 | **0.8258** | 50.40 |
| | $\mathcal{D}^{train}$ | $KL$ | 0.1104 | **30.31** | 0.1616 | **26.82** | 0.2606 | **26.88** | 0.1988 | **25.48** |
| | | $QP$ | **0.7750** | 34.46 | **0.8442** | 36.84 | **0.9384** | 41.22 | **0.9844** | 50.81 |
| Top-3 | $\mathcal{D}^{test}$ | $KL$ | 0.1134 | **24.92** | 0.1906 | **26.24** | 0.3156 | **25.19** | 0.2392 | **22.53** |
| | | $QP$ | **0.5136** | 30.41 | **0.5762** | 29.11 | **0.7030** | 31.45 | **0.7958** | 36.10 |
| | $\mathcal{D}^{train}$ | $KL$ | 0.2160 | **24.93** | 0.3524 | **26.35** | 0.5194 | **25.33** | 0.3786 | **22.59** |
| | | $QP$ | **0.8850** | 30.49 | **0.8788** | 29.30 | **0.9526** | 31.61 | **0.9778** | 36.33 |
| Top-2 | $\mathcal{D}^{test}$ | $KL$ | 0.3838 | **24.26** | 0.4866 | 26.83 | 0.5480 | **21.44** | 0.5566 | **23.34** |
| | | $QP$ | **0.5608** | 25.23 | **0.6320** | **23.45** | **0.7882** | 27.24 | **0.8486** | 28.67 |
| | $\mathcal{D}^{train}$ | $KL$ | 0.5954 | **24.31** | 0.6566 | 27.00 | 0.7522 | **21.55** | 0.7038 | **23.47** |
| | | $QP$ | **0.9264** | 25.30 | **0.9254** | **23.59** | **0.9700** | 27.42 | **0.9802** | 28.88 |
| Top-1 | $\mathcal{D}^{test}$ | $KL$ | **0.8262** | 23.15 | **0.8526** | 21.46 | **0.9268** | 18.43 | 0.9470 | **19.88** |
| | | $QP$ | 0.7164 | **18.36** | 0.7532 | **19.67** | 0.8984 | **18.78** | **0.9508** | 20.02 |
| | $\mathcal{D}^{train}$ | $KL$ | **0.9990** | 23.30 | **0.9936** | 21.63 | **0.9954** | 18.56 | **0.9972** | **20.03** |
| | | $QP$ | 0.9722 | **18.45** | 0.9688 | **19.80** | 0.9890 | **18.91** | 0.9968 | 20.17 |

mulations (Sec. 2.2.1 and Sec. 2.2.2) respectively. We use different seeds in optimization in learning each of the universal perturbations. We compute the ASR with respect to the `Best, Worst` and `Mean` protocols. By `Best`, it means we call it a success attack if any of the 5 samplings does so for an image, training or testing. By `Worst`, it means we call it a failure if any of the 5 samplings does so for an image. By `Mean`, we use the mean success rate among the 5 samplings for an image. Then, the ASRs of a method are computed by the average over the set of data $\mathcal{D}^{train}$ or $\mathcal{D}^{test}$.

### 3.1 Qualitative Results

It is intriguing to visually check the learned universal perturbations (Fig. 2). **We also visualize all the learned 510 perturbations** for a comprehensively qualitative analyses using a HTML based interactive visualization tool in supplementary material (Appendix A.3).

## 3.2 Quantitative Results

We report results of our AllAttacK in terms of increasing universality across images and models. *We report the results using the* Mean *ASRs and $\ell_2$ norms, and provide full results in the Appendix A.4.*

### 3.2.1 AllAttacK: Single-Model and Image-Agnostic

We test $K = 1, 2, \cdots, 6$. We choose two widely recognized pretrained ConvNets: ResNet-50 (He et al., 2016) and DenseNet-121 (Huang et al., 2017), as well as two prominent pretrained Transformers: the vanilla ViT (Base) (Dosovitskiy et al., 2020) and the data-efficient variant DEiT (small) (Touvron et al., 2021). The pretrained checkpoints for these four networks are sourced from the mmpretrain package (Contributors, 2023). Table 1 shows the results. We have some observations as follows:

- **The Model Axis.** We can learn universal (image-agnostic) perturbations for all the four models individually. In terms of ASRs, we observe a *decreasing trend of attacking difficult* from ResNet-50, to DenseNet-121, to DEiT-S and to ViT-B, consistent across both the training set and the testing set and consistent across $K = 1, 2, \cdots, 6$. It is interesting to observe that among the four DNNs, ResNet-50 is the most difficult one to attack, while ViT-B is the easiest one. Our intuitive yet hypothetical explanation for this observation is that the more expressive DNNs are, the easier they might be to suffer from attacks, since the clear-box targeted adversarial attack can fully exploit their expressive power. This may provide some explanations for why aligned multi-modal large language models (which use variants of ViTs as their vision encoder) can be easily attacked as investigated in (Carlini et al., 2024).
- **The Data Axis.** On the unseen testing dataset, for the most difficulty to attack among the four DNNs, ResNet-50, our AllAttacK achieves ASRs greater than 0.5 when $K \leq 3$, and remains reasonably high up to $K = 6$, which shows the strong image-agnosticity of our AllAttacK. As expected, ASRs are consistently and significantly higher on the training dataset than those on the testing datasets. The gaps are roughly between 0.2 and 0.3.
- **The Target Axis.** As we expected, the shear complexity of learning AllAttacK perturbations is increased for larger $K$'s. On both the training and testing datasets, the ASRs decreases along $K = 1, \cdots, 6$. From Fig. 2 (viewed in magnification), it is interesting to observe that learned perturbations exhibit some "features" of the targets, e.g., the tail texture of 'fox squirrel', and the eye-ish shapes like 'can opener' and/or wheel of 'recreational vehicle'. It is also interesting to notice that perturbations learned for ViT-B remain the patchy style.
- **The Optimization Axis.** Overall, the QP optimization (Eqn 14) in the feature embedding space is much stronger than the KL surrogate loss function (Eqn. 9). For Top-1 perturbations, the KL formulation works better than the QP formulation. When $K \geq 2$, the QP formulation is significantly better. Especially for $K = 6$, the KL formulation almost fails to learn the AllAttacK perturbations, while the QP formulation can achieve reasonable high ASRs. For $K = 1$, the top-down designed target distribution $P^{AD}(\mathcal{T})$ (Eqn. 10) is very similar to the one-hot distribution, and the KL divergence objective function is similar to the cross-entropy objective, resulting in effective optimization in learning perturbations.
- **ASRs vs $\ell_2$ Norms.** We note that $\ell_2$ norms of the optimization methods are comparable only when their ASRs are comparable. For example, consider the perturbations for ResNet-50 on the testing dataset, it shows the QP method obtains the $\ell_2$ norm, 56.28, while the KL method has 25.88. The former is computed based on the universal perturbation that is learned to attack 57.4% images in training and to generalized to attack 23.6% images in testing, while the latter is based on the perturbation that can obtains ASRs, 0.3% and 0.08% in training and testing respectively. So, the KL method may achieve lower $\ell_2$ norms due to reaching a saturation point on the ASR, and the norms are only computed on the "easier" targets, which is also observed in (Paniagua et al., 2023). For the same protocol (e.g., Top-6), we have a single perturbation learned using either of the two methods. The slight difference between the $\ell_2$ norms in training and testing is due to the clamping operation, i.e., $\ell_2(\mathcal{P}(\mathcal{T})) = \frac{1}{|\mathcal{D}|} \sum_{x \in \mathcal{D}} ||\text{Clamp}(x + \mathcal{P}) - x||_2$, where $\mathcal{D}$ is the subset of images which can be attacked successfully.

### 3.2.2 AllAttacK: Training-Model-Agnostic and Image-Agnostic

For the same four models as in Sec. 3.2.1, we test three combinations of them: all-4-model, 2-ConvNet and 2-ViT, for $K = 1, \cdots, 6$. Table 2 shows the results. **The observations along the five axes in Sec. 3.2.1 largely remains.** The shear complexity of learning training-model-agnostic and image-agnostic AllAttacK perturbations is significantly increased, especially when $K > 3$. The 2-ConvNet

combination (⊕+⊙) is more difficult to attack than the 2-ViT combination (⊕+⊙). One interesting aspect is that the learned perturbations become more "perceptually meaningful", especially for $K = 1$ and the 4-model combination attack as shown in Fig. 2, which makes intuitive sense in terms of fooling a disparate ensemble of DNNs entailing "tricky yet meaningful" signals that respect and resemble the targets. For example, as pointed out in (Park & Kim, 2021), ConvNets tend to capture more high-frequency texture features, while ViTs tends to capture more low-frequency shape related features. So, fooling them all enforces the learned perturbations not only to respect those spectrum information in isolation, but also to "shut off" information of those images which can be successfully attacked. It is also interesting to observe the change between consecutive perturbations, e.g., we can roughly "perceive" a 'fox squirrel' for the two top-1 perturbations, which is then "mingled" with some vague vehicle part(s) looking regions in the two top-2 perturbations after the target 'recreational vehicle' is added.

Table 2: The mean ASRs and $\ell_2$ of learned AllAttacK perturbations across 5 runs under the training-model-agnostic and image-agnostic setting. Three combinations of four models are tested.

| Protocol | Dataset | Method | ASR↑ | $\ell_2$↓ | ASR↑ | $\ell_2$↓ | ASR↑ | $\ell_2$↓ |
|---|---|---|---|---|---|---|---|---|
| Top-6 | $\mathcal{D}^{test}$ | KL | 0.0004 | **32.97** | 0.0002 | **28.54** | 0.0000 | - |
| | | QP | **0.0928** | 65.07 | **0.1868** | 60.41 | **0.0014** | 59.83 |
| | $\mathcal{D}^{train}$ | KL | 0.0002 | **31.07** | 0.0008 | **28.15** | 0.0000 | - |
| | | QP | **0.2390** | 64.91 | **0.3732** | 60.50 | **0.0044** | 63.24 |
| Top-5 | $\mathcal{D}^{test}$ | KL | 0.0002 | **25.33** | 0.0004 | **21.38** | 0.0000 | - |
| | | QP | **0.1738** | 52.65 | **0.4414** | 64.31 | **0.0322** | 69.65 |
| | $\mathcal{D}^{train}$ | KL | 0.0018 | **25.44** | 0.0016 | **21.34** | 0.0000 | - |
| | | QP | **0.3912** | 52.58 | **0.7092** | 64.62 | **0.0604** | 69.63 |
| Top-4 | $\mathcal{D}^{test}$ | KL | 0.0002 | **21.30** | 0.0014 | **16.68** | 0.0002 | **29.23** |
| | | QP | **0.1610** | 33.78 | **0.6144** | 51.85 | **0.1012** | 56.58 |
| | $\mathcal{D}^{train}$ | KL | 0.0006 | **19.77** | 0.0022 | **16.78** | 0.0000 | - |
| | | QP | **0.3562** | 33.77 | **0.8900** | 52.11 | **0.2258** | 56.54 |
| Top-3 | $\mathcal{D}^{test}$ | KL | 0.0606 | **30.53** | 0.0998 | **25.81** | 0.0002 | **19.72** |
| | | QP | **0.3696** | 33.47 | **0.6748** | 38.93 | **0.2102** | 43.00 |
| | $\mathcal{D}^{train}$ | KL | 0.1152 | **30.58** | 0.1680 | **25.79** | 0.0010 | **19.64** |
| | | QP | **0.7006** | 33.54 | **0.9598** | 39.14 | **0.4316** | 43.03 |
| Top-2 | $\mathcal{D}^{test}$ | KL | 0.2190 | **23.76** | 0.3856 | **22.76** | 0.1030 | **28.05** |
| | | QP | **0.4458** | 23.87 | **0.7746** | 30.36 | **0.3282** | 29.60 |
| | $\mathcal{D}^{train}$ | KL | 0.4078 | **23.83** | 0.5308 | **22.87** | 0.1972 | **28.24** |
| | | QP | **0.7874** | 23.96 | **0.9680** | 30.56 | **0.5888** | 29.66 |
| Top-1 | $\mathcal{D}^{test}$ | KL | 0.7918 | 24.37 | 0.9026 | **19.29** | 0.7142 | **24.08** |
| | | QP | **0.7944** | **20.54** | **0.9374** | 22.19 | **0.7570** | 27.64 |
| | $\mathcal{D}^{train}$ | KL | **0.9918** | 24.55 | 0.9872 | **19.42** | 0.9494 | **24.21** |
| | | QP | 0.9770 | **20.68** | **0.9952** | 22.37 | **0.9538** | 27.83 |

### 3.2.3 ALLATTACK: MODEL- AND IMAGE-AGNOSTIC

We use 18 models in training: ResNets-(18, 34, 50, 101) (He et al., 2016) (with two differently trained checkpoints of ResNet-50 as to make the learning more challenging since our observation in Table 1 show 'ResNet-50' is more difficult to attack), DenseNets-(121, 161, 169, 201) (Huang et al., 2017), HRNet-W18 (Wang et al., 2020), ConvNeXt-(Tiny, Small, Base) (Liu et al., 2022), DEiT-Small (Touvron et al., 2021), DEiT3-(small, medium) (Touvron et al., 2022), ViT-Base (Dosovitskiy et al., 2020) and MLPMixer-Base (Tolstikhin et al., 2021). The pretrained checkpoints are souced from the `timm` package (Wightman, 2019). Due to the shear complexity of attacking a relatively large amount of DNNs simultaneously, we test $K = 1, 2, 3$.

Table 3: The mean ASRs and $\ell_2$ of learned AllAttacK perturbations across 5 runs under the model- and image-agnostic setting. There are 18 disparate models in training $\mathcal{M}^{train}$ (the top 18 rows), and 6 unseen testing models in $\mathcal{M}^{test}$ (the bottom 6 rows). See text for details.

| Model | Method | Top-3 $\mathcal{D}^{test}$ ASR↑ | $\ell_2$↓ | Top-3 $\mathcal{D}^{train}$ ASR↑ | $\ell_2$↓ | Top-2 $\mathcal{D}^{test}$ ASR↑ | $\ell_2$↓ | Top-2 $\mathcal{D}^{train}$ ASR↑ | $\ell_2$↓ | Top-1 $\mathcal{D}^{test}$ ASR↑ | $\ell_2$↓ | Top-1 $\mathcal{D}^{train}$ ASR↑ | $\ell_2$↓ |
|---|---|---|---|---|---|---|---|---|---|---|---|---|---|
| ResNet-18 | KL | 0.1076 | 55.21 | 0.1568 | 55.28 | 0.3987 | 53.03 | 0.4792 | 53.36 | 0.8520 | 31.59 | 0.9644 | 31.88 |
| | QP | **0.2525** | 51.42 | **0.3336** | 51.80 | **0.4531** | 46.36 | **0.5970** | 46.71 | 0.6592 | 25.07 | 0.8254 | 25.20 |
| ResNet-34 | KL | 0.0507 | 54.49 | 0.0792 | 54.71 | 0.3547 | 53.02 | 0.4884 | 53.38 | 0.8411 | 31.60 | 0.9670 | 31.87 |
| | QP | **0.2911** | 51.42 | **0.3936** | 51.76 | **0.4596** | 46.32 | **0.6236** | 46.69 | 0.6792 | 25.03 | 0.8582 | 25.19 |
| ResNet-50 | KL | 0.1096 | 54.70 | 0.1494 | 54.93 | 0.4277 | 53.07 | 0.5454 | 53.38 | 0.8295 | 31.61 | 0.9746 | 31.87 |
| | QP | **0.2797** | 51.54 | **0.3746** | 51.83 | **0.4844** | 46.34 | **0.6368** | 46.67 | 0.6408 | 25.07 | 0.8496 | 25.19 |
| ResNet-50₂ | KL | 0.1888 | 54.63 | 0.2400 | 54.91 | 0.3768 | 52.91 | 0.4476 | 53.25 | **0.8908** | 31.53 | 0.9700 | 31.86 |
| | QP | **0.2214** | 54.81 | **0.2752** | 55.35 | 0.3725 | 46.20 | **0.4882** | 46.56 | 0.5292 | 25.02 | 0.6464 | 25.18 |
| ResNet-101 | KL | 0.1489 | 54.84 | 0.2170 | 55.11 | 0.4337 | 53.08 | 0.5496 | 53.40 | 0.8232 | 31.61 | 0.9644 | 31.87 |
| | QP | **0.2839** | 51.36 | **0.3812** | 51.73 | **0.4580** | 46.36 | **0.6526** | 46.67 | 0.6435 | 25.07 | 0.8292 | 25.19 |
| DenseNet121 | KL | 0.1230 | 54.72 | 0.1854 | 55.11 | 0.3955 | 52.98 | 0.4980 | 53.38 | 0.8645 | 31.55 | 0.9752 | 31.87 |
| | QP | **0.3201** | 51.47 | **0.4320** | 51.78 | **0.5364** | 46.24 | **0.6940** | 46.62 | 0.6643 | 25.01 | 0.8392 | 25.18 |
| DenseNet161 | KL | 0.1464 | 54.54 | 0.1894 | 54.94 | 0.5056 | 53.03 | 0.6236 | 53.36 | 0.8672 | 31.55 | 0.9802 | 31.86 |
| | QP | **0.2938** | 51.32 | **0.4044** | 51.72 | **0.5107** | 46.25 | **0.6954** | 46.64 | 0.6368 | 25.04 | 0.8314 | 25.18 |
| DenseNet169 | KL | 0.2304 | 54.84 | 0.3268 | 55.10 | 0.4714 | 52.98 | 0.5842 | 53.37 | 0.8766 | 31.56 | 0.9786 | 31.87 |
| | QP | **0.3181** | 51.42 | **0.4198** | 51.73 | **0.5522** | 46.20 | **0.7430** | 46.61 | 0.7121 | 25.01 | 0.8614 | 25.18 |
| DenseNet201 | KL | 0.1272 | 54.80 | 0.1792 | 55.01 | 0.4092 | 52.97 | 0.5152 | 53.32 | 0.8692 | 31.56 | 0.9710 | 31.86 |
| | QP | **0.3123** | 51.33 | **0.4222** | 51.65 | **0.5359** | 46.25 | **0.7068** | 46.63 | 0.6609 | 25.04 | 0.8208 | 25.18 |
| HRNet-W18 | KL | 0.0009 | 35.31 | 0.0004 | 35.80 | 0.0806 | 33.22 | 0.1142 | 33.57 | **0.6556** | 31.64 | **0.8332** | 31.89 |
| | QP | **0.0040** | 35.04 | **0.0080** | 35.52 | **0.1371** | 32.45 | **0.2172** | 32.64 | 0.4464 | 25.06 | 0.6360 | 25.18 |
| ConvNeXt-S | KL | **0.3214** | 54.76 | **0.4028** | 54.96 | **0.5437** | 52.99 | **0.6292** | 53.38 | 0.9143 | 31.53 | 0.9732 | 31.87 |
| | QP | 0.2408 | 51.44 | 0.2884 | 51.74 | 0.4795 | 46.25 | 0.5496 | 46.71 | 0.4933 | 25.11 | 0.6142 | 25.26 |
| ConvNeXt-T | KL | **0.3404** | 54.73 | **0.4240** | 54.93 | **0.5768** | 52.96 | **0.6914** | 53.35 | 0.9192 | 31.52 | 0.9732 | 31.87 |
| | QP | 0.2703 | 51.25 | 0.3390 | 51.78 | 0.5109 | 46.27 | 0.6340 | 46.70 | 0.4759 | 25.18 | 0.5818 | 25.27 |
| ConvNeXt-B | KL | **0.3513** | 54.77 | **0.4272** | 54.95 | **0.5417** | 52.99 | **0.6558** | 53.40 | 0.9194 | 31.51 | 0.9714 | 31.87 |
| | QP | 0.2504 | 51.41 | 0.3000 | 51.75 | 0.4373 | 46.26 | 0.5028 | 46.68 | 0.4741 | 25.11 | 0.6064 | 25.28 |
| DEiT-S | KL | 0.2462 | 54.85 | 0.3372 | 55.00 | 0.5225 | 53.02 | 0.6484 | 53.42 | 0.7853 | 31.60 | 0.8970 | 31.89 |
| | QP | **0.4810** | 51.23 | **0.6634** | 51.65 | **0.6839** | 46.13 | **0.8332** | 46.62 | 0.6556 | 25.00 | 0.7850 | 25.19 |
| DEiT3-S | KL | 0.1545 | 59.67 | 0.1778 | 55.17 | 0.4857 | 53.10 | 0.5758 | 53.46 | **0.9103** | 31.54 | 0.9566 | 31.88 |
| | QP | **0.5886** | 51.19 | **0.7218** | 51.67 | **0.7417** | 46.15 | **0.8516** | 46.65 | 0.8145 | 24.98 | 0.8964 | 25.18 |
| DEiT3-M | KL | 0.0554 | 134.76 | 0.0552 | 84.10 | 0.1632 | 64.69 | 0.1528 | 57.43 | 0.7116 | 31.59 | 0.8448 | 31.88 |
| | QP | **0.2025** | 75.11 | **0.2046** | 75.78 | **0.4174** | 46.18 | **0.5188** | 46.78 | 0.4743 | 24.98 | 0.6268 | 25.19 |
| ViT-B | KL | 0.2192 | 54.71 | 0.2506 | 54.97 | 0.3790 | 53.02 | 0.4492 | 53.40 | 0.7183 | 31.62 | 0.8658 | 31.90 |
| | QP | **0.4991** | 51.23 | **0.6838** | 51.68 | **0.6732** | 46.13 | **0.8278** | 46.64 | 0.7154 | 25.01 | 0.8814 | 25.19 |
| MlpMixer-B | KL | **0.1033** | 68.15 | **0.1056** | 55.26 | 0.0879 | 33.55 | 0.1114 | 33.73 | **0.4614** | 31.84 | **0.5086** | 32.05 |
| | QP | 0.0179 | 36.09 | 0.0336 | 35.64 | **0.1547** | 45.23 | **0.2100** | 45.85 | 0.2127 | 25.23 | 0.2486 | 25.31 |
| ConvMixer-768 | KL | **0.0339** | 68.20 | **0.0510** | 67.99 | 0.2308 | 53.16 | 0.3328 | 53.33 | **0.6703** | 31.60 | **0.7852** | 31.89 |
| | QP | 0.0150 | 61.49 | 0.0300 | 62.27 | **0.2900** | 46.13 | 0.2738 | 46.63 | 0.3708 | 25.03 | 0.4524 | 25.19 |
| Swin-B | KL | 0.0228 | **54.78** | 0.0330 | 59.84 | 0.1667 | 53.27 | 0.1906 | 53.61 | **0.5935** | 31.74 | **0.7098** | 32.00 |
| | QP | **0.0679** | 55.21 | **0.0680** | 51.57 | **0.2589** | 46.39 | **0.2738** | 46.90 | 0.2243 | 25.25 | 0.2416 | 25.37 |
| HRNet-W30 | KL | **0.0016** | 34.55 | **0.0018** | 35.45 | 0.0654 | 32.90 | 0.0834 | 33.37 | **0.6362** | 31.67 | **0.8126** | 31.89 |
| | QP | 0.0004 | 34.10 | 0.0026 | 35.18 | **0.0942** | 32.59 | **0.1526** | 32.68 | 0.4109 | 25.07 | 0.5774 | 25.18 |
| ConvNeXtV2-H | KL | 0.0100 | 55.46 | 0.0154 | 34.89 | 0.1384 | 53.17 | 0.1692 | 58.48 | **0.2330** | 31.88 | **0.2359** | 32.26 |
| | QP | **0.0493** | 55.23 | **0.0667** | 72.25 | **0.1944** | 46.49 | **0.1846** | 49.87 | 0.1009 | 25.28 | 0.0974 | 25.23 |
| CLIP ViT-B | KL | **0.0018** | 34.86 | 0.0000 | - | 0.0210 | 52.98 | 0.0308 | 34.14 | **0.3100** | 31.72 | **0.4872** | 31.95 |
| | QP | **0.0018** | 75.63 | 0.0000 | - | **0.0922** | 46.49 | **0.1487** | 46.81 | 0.1587 | 25.11 | 0.2615 | 25.29 |
| EVA2 ViT-B | KL | 0.0000 | - | 0.0000 | - | 0.0000 | - | 0.0000 | - | 0.0560 | 31.41 | 0.0872 | 31.48 |
| | QP | 0.0000 | - | 0.0000 | - | 0.0000 | - | 0.0000 | - | **0.0592** | 25.36 | **0.0923** | 25.28 |

We test targeted attacks up to Top-3 on 6 unseen models including 3 ImageNet-1k trained HRNet-W30 (Wang et al., 2020), ConvMixer-768 (isotropic architecture) (Trockman & Kolter, 2023) and Swin-Base (Liu et al., 2021), where ConvMixer-768 represents the convolutional isotropic architecture which does not show in the training (which instead contains isotropic ViT architectures), and Swin-Base represents the hierarchical Transformer architecture which does not show in training (which instead includes hierarchical convolutional architectures). The remaining 3 unseen testing models are state-of-the-art DNNs pretrained either using Masked Image Modeling (MIM) with ImageNet-21k (e.g. ConvNeXtV2-H (Woo et al., 2023)), or using contrastive language image pretraining (CLIP) with a massive number ($\sim$4M) of proprietary image-caption pairs (e.g., OpenAI CLIP ViT-B (Radford et al., 2021)), or combining MIM and CLIP (e.g., EVA2 ViT-B (Fang et al., 2023)), before fine-tuned on the ImageNet-1k.

Table 3 shows the results. Fig. 2 show examples of the learned perturbations. For the 18 training models, we achieve very promising results overall across $K = 1, 2, 3$, except for the Top-3 attacks for HRNet-W18 and MlpMixer-B. The robustness achieved by HRNets may be due to the aggregation of high-resolution features in their backbones, while MlpMixer-B gains its robustness from the globally spatial MLP of token mixing. For the first 3 unseen models (in the red cells), HRNet-W30 retains its robustness similar to HRNet-W18 in training. For ConvMixer-768 and Swin-B, both Top-1 and Top-2 attacks have reasonable ASRs given that their architectures do not really show up in the training, while Top-3 attacks for them have a significant drop of ASRs. For the 3 unseen models (in the green cells) that have been pretrained using large-scale data, Top-3 attacks have overall low ASRs. EVA2 ViT-B that uses a sophisticated MIM and CLIP integrated pretraining strategy is the most robust one, for which even Top-1 attacks have low ASRs. Between ConvNeXtV2-H and OpenAI CLIP ViT-B, the former has promising ASRs for both Top-1 and Top2, while the latter has higher Top-1 ASRs, but lower Top-2 ASRs. **We note that the promising Top-1 and Top-2 testing ASRs on both ConvNeXtV2-H and CLIP ViT-B (unseen models) show the great potential of the proposed AllAttacK, which has not been made possible in the prior art, to the best of our knowledge.**

For the six testing DNNs in Table 3, we further run single-model and image-agnostic AllAttacK (as those in Sec. 3.2.1). Table 4 shows the results. We observe that ASRs for all the six DNNs are high at the expense of

Table 4: The `mean` ASRs and $\ell_2$ of learned AllAttacK perturbations across 5 runs under the single-model-image-agnostic setting for the six testing DNNs in Table 3. See text for details.

| Protocol | Dataset | Method | Swin-B | | HRNet-W30 | | ConvMixer-768 | | CLIP-ViT-B | | EVA2-ViT-B | | ConvNeXtV2-$H$ | |
|---|---|---|---|---|---|---|---|---|---|---|---|---|---|---|
| | | | ASR↑ | $\ell_2$↓ | ASR↑ | $\ell_2$↓ | ASR↑ | $\ell_2$↓ | ASR↑ | $\ell_2$↓ | ASR↑ | $\ell_2$↓ | ASR↑ | $\ell_2$↓ |
| Top-3 | $\mathcal{D}^{test}$ | QP | **0.9974** | **81.46** | **0.9810** | **69.36** | **0.9933** | **87.62** | **0.9997** | 115.42 | **0.9954** | 65.16 | **0.9932** | 88.36 |
| | | KL | 0.9350 | 104.13 | 0.6282 | 87.64 | 0.7522 | 94.18 | 0.4080 | 152.15 | 0.7714 | 96.12 | 0.9454 | 95.36 |
| | $\mathcal{D}^{train}$ | QP | **1.0000** | **81.45** | **0.9978** | **69.33** | **0.9990** | **87.61** | **0.9996** | 115.42 | **0.9986** | 65.12 | **0.9996** | 88.32 |
| | | KL | 0.9532 | 104.05 | 0.7888 | 87.69 | 0.8282 | 94.21 | 0.4068 | 151.42 | 0.8798 | 96.14 | 0.9898 | 95.24 |
| Top-2 | $\mathcal{D}^{test}$ | QP | **0.9972** | **76.01** | **0.9851** | **62.43** | **0.9947** | **78.38** | **0.9991** | 95.32 | **0.9942** | 54.84 | 0.5974 | 88.93 |
| | | KL | 0.9798 | 98.02 | 0.8694 | 84.02 | 0.9238 | 89.67 | 0.6426 | 141.67 | 0.9594 | 81.90 | **0.9548** | **88.80** |
| | $\mathcal{D}^{train}$ | QP | **0.9998** | **75.97** | **0.9996** | **62.38** | **0.9990** | **78.36** | **0.9994** | 95.35 | **0.9990** | 54.81 | 0.5998 | 88.88 |
| | | KL | 0.9858 | 97.98 | 0.8810 | 84.03 | 0.9144 | 89.71 | 0.6366 | 141.76 | 0.9732 | 81.89 | **0.9712** | **88.72** |
| Top-1 | $\mathcal{D}^{test}$ | QP | 0.9994 | 88.55 | 0.9953 | 60.32 | 0.9985 | 70.00 | 0.9992 | 86.93 | **0.9994** | 54.32 | **0.9982** | 101.80 |
| | | KL | **0.9996** | **72.41** | **0.9984** | **55.46** | **0.9994** | **56.55** | **1.0000** | **86.47** | 0.9940 | 35.79 | 0.8000 | **74.43** |
| | $\mathcal{D}^{train}$ | QP | **1.0000** | 88.56 | **1.0000** | 60.30 | **1.0000** | 69.98 | 0.9996 | 86.88 | **1.0000** | 54.30 | **1.0000** | 101.83 |
| | | KL | **1.0000** | **72.39** | **1.0000** | **55.45** | **1.0000** | **56.52** | **1.0000** | **86.45** | **1.0000** | 35.77 | 0.8000 | **74.37** |

increased $\ell_2$ energy of the learned UAPs, compared with DNNs in Table 1. However, it is interesting to observe that although trained with a sophisticated recipe with large-scale data, EVA2-ViT-B Fang et al. (2023) can be attacked with lower $\ell_2$ energy. The gaps between the ASRs by transferred UAPs and those learned with the six DNNs themselves show the shear complexity of transferring more aggressive ordered top-$K$ attacks.

### 3.2.4 ALLATTACK: ARE ADVERSARIALLY ROBUSTIFIED DNNS ACTUALLY ROBUST?

We test three models on ImageNet-1k, ResNet-50$^1_{robust}$ (Engstrom et al., 2019) (clean top-1 accuracy: 62.56%), ResNet-50$^2_{robust}$ (Salman et al., 2020) (clean top-1 accuracy: 64.02%) and WideResNet-50$_{robust}$ (Salman et al., 2020) (clean top-1 accuracy: 68.46%), sourced from the RobustBench (Croce et al., 2020). They have undergone different adversarial robustification training process with a trade-off significantly sacrificing top-1 accuracy on clean data. For example, the standard

Table 5: The `mean` ASRs and $\ell_2$ of learned AllAttacK perturbations across 5 runs under the single-model-image-agnostic setting. See text for details.

| Protocol | Dataset | Method | ResNet-50$^1_{Robust}$ | | ResNet-50$^2_{Robust}$ | | WideResNet-50$_{Robust}$ | |
|---|---|---|---|---|---|---|---|---|
| | | | ASR↑ | $\ell_2$↓ | ASR↑ | $\ell_2$↓ | ASR↑ | $\ell_2$↓ |
| Top-3 | $\mathcal{D}^{test}$ | QP | **0.9981** | **162.85** | **0.9977** | **173.32** | **0.9971** | **162.38** |
| | | KL | 0.7899 | 182.37 | 0.7739 | 192.88 | 0.7168 | 182.36 |
| | $\mathcal{D}^{train}$ | QP | **0.9998** | **162.69** | **0.9998** | **173.21** | **0.9998** | **162.19** |
| | | KL | 0.9847 | 182.65 | 0.9815 | 193.57 | 0.9765 | 182.72 |
| Top-2 | $\mathcal{D}^{test}$ | QP | **0.9917** | **140.47** | **0.9954** | **153.06** | **0.9942** | **141.12** |
| | | KL | 0.8739 | 168.22 | 0.8557 | 183.98 | 0.8735 | 174.14 |
| | $\mathcal{D}^{train}$ | QP | **0.9996** | **140.26** | **0.9998** | **152.90** | **1.0000** | **140.90** |
| | | KL | 0.9869 | 168.09 | 0.9805 | 184.13 | 0.9920 | 174.14 |
| Top-1 | $\mathcal{D}^{test}$ | QP | 0.9971 | **125.42** | 0.9981 | **137.32** | 0.9957 | **128.43** |
| | | KL | **0.9998** | 161.62 | **1.0000** | 176.38 | **1.0000** | 166.15 |
| | $\mathcal{D}^{train}$ | QP | **1.0000** | **125.21** | 0.9998 | **137.12** | 0.9998 | **128.15** |
| | | KL | **1.0000** | 161.33 | **1.0000** | 176.20 | **1.0000** | 165.85 |

ResNet-50 can obtain clean top-1 accuracy 76.52%. With the much worse clean top-1 accuracy, the

training set and testing set selected in training and evaluating AllAttacK are much more restricted. Table 5 shows the results. Our AllAttacK can achieve high ASRs at the expense of increased $\ell_2$ energy, similar to the observations for the six models in Table 4. Fig. 2 (the last row) shows examples of the learned UAPs, which "behave" significantly different from UAPs learned with the standard ResNet-50 in the left-top of Fig. 2. We can clearly see that adversarially robustified models can "enforce" the learned UAPs to be more sense-making, similar in spirit to those by a combination of standard models (e.g., the 18-model ensemble).

## 4 RELATED WORK

Since Szegedy et al. (2013) showed the brittleness of DNNs, many works have studied their vulnerabilities. We briefly overview adversarial attacks in thee categories that are important to understand the evolution to challenges this paper aims to address.

**Ordered Top-K Adversarial Attacks.** Ordered Top-K Adversarial Attacks aim to dictate the exact content and order of the first Top-K predicted classes. Adversarial Distillation (AD) Zhang & Wu (2020) addresses this by employing combining knowledge distillation and semantic word embeddings to craft a target distribution of class probabilities. Once AD computes a target class probabilities, it then uses a Kullback-Leiber divergence loss to solve for a perturbation that achieves its target. QuadAttack Paniagua et al. (2023) approaches this challenge through quadratic programming in the feature embedding space, directly finding a perturbation in embedding space. Once an embedding space perturbation has been found, an $\ell_2$ loss between the perturbed and current embedding space features can be used to solve for a perturbation in data-space.

**Universal Adversarial Perturbations (UAPs).** One remarkable discovery in Szegedy et al. (2013) was that adversarial attacks have the ability to transfer to models trained with different hyperparameters or training sets than those the adversarial attack was generated with. Liu et al. (2016) later investigates the transferability of adversarial examples among different neural network architectures, differentiating between targeted and non-targeted attacks. Liu et al. (2016) introduces an ensemble-based attack method similar in spirit to our own AllAttack, enabling successful targeted adversarial attacks. This further extended to Universal Adversarial Perturbations Hendrik Metzen et al. (2017); Moosavi-Dezfooli et al. (2017); Shafahi et al. (2020) . Moosavi-Dezfooli et al. (2017) and Shafahi et al. (2020) achieve a single perturbation that can be applied to a large number of images for a model and prevent correct classification, While Hendrik Metzen et al. (2017) extended the concept of universal attacks to the domain of segmentation. Zhang et al. (2020) first observes the possibility of common class-specific "features" across universal perturbations. Benz et al. (2020) extends the specificity of UAPs to only change predictions for one specified "source" class and change them to a prescribed "sink" class, leaving all other classes unchanged.

## 5 CONCLUSION AND DISCUSSION

This paper presents a method for learning universal ordered Top-K targeted adversarial perturbations that are both image-agnostic and model-agnostic under the white-box attack setting. The proposed method is dubbed as AllAttacK. It defines the problem of AllAttacK along three axes (model, data and targets). It presents two optimization methods in learning AllAttacK, built on previous single-model and instance-specific ordered Top-K attack methods, which enable training AllAttacK with a large number of disparate deep neural networks (up to 18 models). The proposed AllAttacK is thoroughly evaluated in experiments with 510 universal ordered top-K perturbations learned at three different levels of universality, and with strong or promising attack success rates obtained.

**Discussions: Towards Testing and Verifying the Interpretability-Robustness Conjecture.** The emerged sense-making appearance of the learned AllAttacK UAPs for a combination of models and adversarially trained models motivate us to make the interpretibility-robustness conjecture: **A DNN will be adversarially robust in a holistic way if its AllAttacK adversarial perturbations are semantically meaningful (i.e., in the close proximity to or even inside the real data manifold). From a quantitatively equivalent viewpoint, it means that the perturbations themselves in isolation will be classified by the DNN with the top-$K$ predictions equal to the ordered top-$K$ targets ($K \geq 1$). Ideally and ultimately (in the long run), a DNN is certified to be robust if its AllAttacK perturbations are confined to be high-fidelity synthesized images for the ordered top-$K$ targets, i.e., the closed-loop of AllAttacK-as-Generator.** We hope this conjecture can sheds light on addressing adversarial defense, and we leave it to be tested and verified in future work.

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

# A APPENDIX

## A.1 BROADER IMPACTS

The promising generalization capability of the learned universal top-K perturbations to unseen models, especially those at the so-called foundation model level (e.g., the CLIP ViT-B), might be exploited in a harmful way for applications built on those models. Powerful defense methods should be studied, which we will investigate in our future work based on the proposed interpretability-robustness conjecture. We will also release our source code to encourage more research on studying defense methods against the proposed AllAttacK.

## A.2 DETAILS OF OPTIMIZATION

We build on the released code of QuadAttacK Paniagua et al. (2023) [1]. For better understanding of our methodology, we provide details on the configuration used for optimizing our learned perturbations. In all configurations we use the *AdamW* optimizer with a learning rate of 0.002 from PyTorch to minimize our presented objectives (Eqn. 9 and Eqn. 14 in the main paper). We run all configurations for 50 epochs on the training images and models.

To learn a perturbation, both the QP and KL methods require choosing a hyperparameter $\lambda$ for the loss term focused on satisfying the Ordered Top-K constraint. There is no "optimal" value for this parameter, it is a trade-off parameter that selects a point on the ASR vs Energy tradeoff curve. While $\lambda$ just represents a tradeoff curve point, there also exists a minimum energy at which this curve yields an ASR greater than 0.

To facilitate finding successful attacks on more challenging cases (e.g., the 18-model ensemble attack), we perform our optimization for multiple $\lambda$ values and select the smallest energy that obtained a non-negligible ASR. For QP we search in $\lambda \in \{100, 150\}$ and for KL we search in $\lambda \in \{1000, 1500\}$, where we choose different magnitudes for QP and KL due to the different spaces that losses operate in.

We use 1 Nvidia A100 80G GPU in all our experiments. We run multiple configurations (e.g., different $K$'s and DNN combinations) in parallel across 4-8 GPUs on our server.

## A.3 ALL LEARNED 510 PERTURBATIONS

We visualize all learned perturbations for a comprehensively qualitative analyses. There are 5 runs in sampling the targets. For each run, we have 8 model variations: 4 individual models (in Table 1 of the main paper), 2-ConvNet combination and 2-ViT combination and 4-model combination (in Table 2 of the main paper), and 18-model combination (in Fig.4 of the main paper). For the first 7 model variations, we test $K = 1, \cdots, 6$, and for the last variation we test $K = 1, 2, 3$, all using two optimization methods functions, the KL formulation (Eqn. in the main paper) and the QP formulation (Eqn. in the main paper). The total number of perturbations are $5 \times 7 \times 6 \times 2 + 15 \times 3 \times 2 = 510$.

We develop a HTML based interactive visualization tool (Fig. 3). **Please check the** `index.html` **for browsing all the perturbations in this supplementary material.** Due to the file size limit of supplementary material (100M vs 140M), we remove some perturbations learned using the KL formulation. We will release all the 510 perturbations, together with the source code after the review process.

## A.4 DETAILED QUANTITATIVE RESULTS

In the main paper, due to space limit, we report results using `mean` ASRs and $\ell_2$ norms. In this section, we report the full results in terms of `Best`, `Mean` and `Worst` ASRs and $\ell_1$, $\ell_2$ and $\ell_\infty$ norms.

- The full results of Table 1 in the main paper are shown in Table 6 and Table 7.
- The full results of Table 2 in the main paper are shown in Table 8 and Table 9.

---

[1] `https://github.com/thomaspaniagua/quadattack`

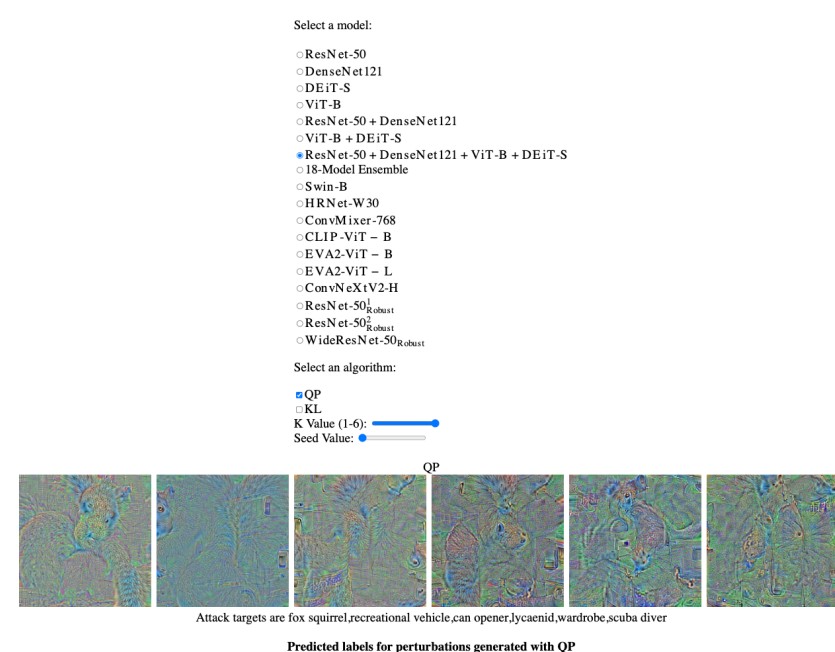

Figure 3: The HTML visualizer interface.

- The full results of the 18-model ensemble AllAttacK in Fig. 4 (which is a typo and should be Table 3 as aforementioned) in the main paper are shown in Tables from 10 to 16.

  - Table 10 shows the results for the 5 ResNets used in training.
  - Table 11 shows the results for the 4 DenseNets used in training.
  - Table 12 shows the results for the HRNet-W18 and 2 ConvNeXts used in training.
  - Table 13 shows the results for the 3 DEiTs used in training.
  - Table 14 shows the results for the ViT-B and MlpMixer-B used in training.
  - Table 15 shows the results for the three unseen testing DNNs (ConvMixer-768, SWin-B and HRNet-W30).
  - Table 16 shows the results for the three unseen testing DNNs at the foundation model level (ConvNeXtV2-H, CLIP-ViT-B and EVA2-ViT-B).
  - Table 17 shows the results for the six unseen testing DNNs (ConvMixer-768, SWin-B, HRNet-W30, ConvNeXtV2-H, CLIP-ViT-B and EVA2-ViT-B) under the single-model-image-agnostic AllAttacK.

Table 6: The ASRs and norms of learned AllAttacK perturbations across 5 runs under the single-model and image-agnostic setting: ResNet-50 (*top*) and DenseNet-121 (*bottom*), using the same 1000 training images $\mathcal{D}^{train}$ and 1000 testing images $\mathcal{D}^{test}$. The surrogate KL loss function (Eqn. 9) and the QP method (Eqn. 14) are tested and compared.

**ResNet-50**

| Protocol | Attack Method | Best | | | | Mean | | | | Worst | | | |
|---|---|---|---|---|---|---|---|---|---|---|---|---|---|
| | | ASR↑ | $\ell_1\downarrow$ | $\ell_2\downarrow$ | $\ell_\infty\downarrow$ | ASR↑ | $\ell_1\downarrow$ | $\ell_2\downarrow$ | $\ell_\infty\downarrow$ | ASR↑ | $\ell_1\downarrow$ | $\ell_2\downarrow$ | $\ell_\infty\downarrow$ |
| Top-6 | $KL_{test}$ | 0.0020 | **7423.81** | 25.01 | **0.6499** | 0.0008 | 7534.61 | 25.88 | 0.5851 | 0.0000 | - | - | - |
| | $KL_{train}$ | 0.0100 | **7129.05** | **24.65** | **0.5437** | 0.0030 | 7341.37 | 25.44 | 0.5781 | 0.0000 | - | - | - |
| | $QP_{test}$ | **0.2880** | 17380.21 | 56.73 | 0.7784 | **0.2360** | 17082.95 | 56.28 | 0.7882 | **0.1980** | 19745.88 | 64.20 | 0.8107 |
| | $QP_{train}$ | **0.6190** | 15870.16 | 52.67 | 0.7924 | **0.5740** | 17006.52 | 56.15 | 0.7928 | **0.5020** | 17019.15 | 56.37 | 0.8018 |
| Top-5 | $KL_{test}$ | 0.0430 | **9103.39** | 31.60 | **0.6410** | 0.0204 | 9036.74 | 31.01 | 0.6483 | 0.0060 | 8569.89 | 29.76 | 0.7003 |
| | $KL_{train}$ | 0.0830 | **8816.55** | **30.35** | 0.7230 | 0.0592 | 8937.48 | 30.76 | 0.6534 | 0.0220 | 9191.38 | 31.68 | 0.6515 |
| | $QP_{test}$ | **0.4020** | 12784.94 | 43.21 | 0.7235 | **0.3452** | 13484.07 | 45.28 | 0.7624 | **0.2950** | 12049.47 | 41.35 | 0.7621 |
| | $QP_{train}$ | **0.8030** | 13297.80 | 44.62 | 0.7875 | **0.7416** | 13420.23 | 45.17 | 0.7695 | **0.6740** | 15200.24 | 50.91 | 0.7959 |
| Top-4 | $KL_{test}$ | 0.1130 | **8609.84** | **30.10** | **0.6598** | 0.0544 | 8860.27 | 30.40 | 0.6863 | 0.0190 | 9231.51 | 31.40 | 0.6954 |
| | $KL_{train}$ | 0.1750 | **8498.26** | 29.82 | 0.6726 | 0.1104 | 8820.95 | 30.31 | 0.6995 | 0.0370 | 9195.29 | 31.28 | 0.7181 |
| | $QP_{test}$ | **0.4260** | 8754.73 | 31.12 | 0.7082 | **0.4050** | 9966.86 | 34.43 | 0.7224 | **0.3750** | 11121.31 | 37.48 | 0.7065 |
| | $QP_{train}$ | **0.8480** | 8734.67 | 31.09 | 0.7066 | **0.7750** | 9965.70 | 34.46 | 0.7257 | **0.7150** | 11122.73 | 37.53 | **0.7094** |
| Top-3 | $KL_{test}$ | 0.1380 | **7820.22** | 27.03 | **0.6499** | 0.1134 | 7189.22 | 24.92 | 0.5911 | 0.0810 | 6669.33 | 23.69 | 0.6076 |
| | $KL_{train}$ | 0.2710 | **7418.35** | 25.69 | 0.6184 | 0.2160 | 7180.67 | 24.93 | 0.5934 | 0.1320 | 6655.45 | 23.70 | 0.6125 |
| | $QP_{test}$ | **0.5300** | 9423.68 | 32.46 | 0.6738 | **0.5136** | 8798.63 | 30.41 | 0.6679 | **0.4780** | 8258.90 | 28.91 | 0.6242 |
| | $QP_{train}$ | **0.9090** | 8244.32 | 28.89 | 0.6281 | **0.8850** | 8811.83 | 30.49 | 0.6706 | **0.8640** | 9455.38 | 32.60 | 0.6750 |
| Top-2 | $KL_{test}$ | 0.4540 | **7050.01** | 24.58 | **0.5399** | 0.3838 | 6961.05 | 24.26 | 0.5593 | 0.3100 | 6742.45 | 23.83 | 0.5456 |
| | $KL_{train}$ | 0.7400 | 7080.44 | 24.70 | **0.5413** | 0.5954 | 6972.23 | 24.31 | 0.5599 | 0.4980 | 6727.38 | 23.80 | 0.5459 |
| | $QP_{test}$ | **0.6120** | 7598.11 | 26.47 | 0.5779 | **0.5608** | 7264.53 | 25.23 | 0.5983 | **0.5200** | 7599.71 | 26.09 | 0.6272 |
| | $QP_{train}$ | **0.9620** | **6744.20** | **23.78** | 0.5437 | **0.9264** | 7280.19 | 25.30 | 0.5976 | **0.9040** | 7612.42 | 26.15 | 0.6162 |
| Top-1 | $KL_{test}$ | 0.8490 | 7236.13 | 24.65 | 0.6471 | 0.8262 | 6724.54 | 23.15 | 0.5118 | 0.7810 | 6284.51 | 22.04 | **0.5044** |
| | $KL_{train}$ | **1.0000** | 6601.40 | 22.58 | **0.4444** | **0.9990** | 6772.62 | 23.30 | 0.5129 | **0.9980** | 7297.60 | 24.83 | 0.6539 |
| | $QP_{test}$ | 0.7930 | **4525.26** | **16.60** | 0.5165 | 0.7164 | **5220.64** | 18.36 | 0.5151 | 0.6460 | **5486.59** | 18.87 | 0.5320 |
| | $QP_{train}$ | 0.9830 | **4551.12** | **16.70** | 0.5169 | 0.9722 | **5245.27** | 18.45 | 0.5202 | 0.9640 | **5505.92** | 18.94 | 0.5392 |

**DenseNet121**

| Protocol | Attack Method | Best | | | | Mean | | | | Worst | | | |
|---|---|---|---|---|---|---|---|---|---|---|---|---|---|
| | | ASR↑ | $\ell_1\downarrow$ | $\ell_2\downarrow$ | $\ell_\infty\downarrow$ | ASR↑ | $\ell_1\downarrow$ | $\ell_2\downarrow$ | $\ell_\infty\downarrow$ | ASR↑ | $\ell_1\downarrow$ | $\ell_2\downarrow$ | $\ell_\infty\downarrow$ |
| Top-6 | $KL_{test}$ | 0.0010 | 5610.15 | 19.50 | 0.3955 | 0.0004 | 5560.38 | 19.55 | 0.4853 | 0.0000 | - | - | - |
| | $KL_{train}$ | 0.0040 | **5490.69** | 19.55 | 0.5415 | 0.0010 | 5542.69 | 19.50 | 0.4952 | 0.0000 | - | - | - |
| | $QP_{test}$ | **0.4130** | 15012.97 | 50.37 | 0.7441 | **0.3800** | 17065.69 | 56.36 | 0.7497 | **0.3260** | 19773.50 | 64.14 | 0.7576 |
| | $QP_{train}$ | **0.7040** | 14884.52 | 49.91 | 0.7066 | **0.6444** | 17093.06 | 56.49 | 0.7500 | **0.5300** | 17430.55 | 57.84 | 0.7887 |
| Top-5 | $KL_{test}$ | 0.0300 | **8175.26** | 28.92 | 0.6362 | 0.0226 | 8359.95 | 29.20 | 0.5907 | 0.0170 | 8400.00 | 29.52 | 0.5774 |
| | $KL_{train}$ | 0.1000 | **8007.07** | 28.47 | 0.6436 | 0.0680 | 8273.76 | 28.98 | 0.5960 | 0.0420 | 8601.66 | 29.80 | 0.5629 |
| | $QP_{test}$ | **0.5120** | 11333.09 | 39.52 | 0.6606 | **0.4652** | 13640.25 | 45.89 | 0.7054 | **0.4040** | 15879.71 | 52.21 | 0.7253 |
| | $QP_{train}$ | **0.8240** | 11425.21 | 39.78 | 0.6625 | **0.7318** | 13697.98 | 46.11 | 0.7055 | **0.6640** | 15673.12 | 51.92 | 0.7065 |
| Top-4 | $KL_{test}$ | 0.1010 | **7698.26** | 27.10 | 0.5438 | 0.0734 | 7652.10 | 26.69 | 0.5683 | 0.0440 | 7935.90 | 27.24 | 0.6076 |
| | $KL_{train}$ | 0.2820 | **7692.78** | 27.10 | 0.5445 | 0.1616 | 7673.86 | 26.82 | 0.5690 | 0.0810 | 8002.22 | 27.54 | 0.6011 |
| | $QP_{test}$ | **0.6130** | 12777.94 | 43.11 | 0.6665 | **0.5258** | 10638.53 | 36.63 | 0.6645 | **0.4040** | 9645.29 | 33.60 | 0.6178 |
| | $QP_{train}$ | **0.8790** | 9132.35 | 32.28 | 0.6909 | **0.8442** | 10704.41 | 36.84 | 0.6632 | **0.8040** | 9645.13 | 33.61 | 0.6188 |
| Top-3 | $KL_{test}$ | 0.2190 | **7419.83** | 25.96 | 0.5917 | 0.1906 | 7532.61 | 26.24 | 0.5883 | 0.1750 | 7703.96 | 26.57 | 0.5890 |
| | $KL_{train}$ | 0.4870 | **7422.59** | 25.99 | 0.5933 | 0.3524 | 7560.13 | 26.35 | 0.5909 | 0.2810 | 7708.60 | 26.66 | 0.5859 |
| | $QP_{test}$ | **0.6040** | 7830.06 | 27.29 | 0.6664 | **0.5762** | 8349.86 | 29.11 | 0.6174 | **0.5330** | 8505.32 | 29.35 | 0.6079 |
| | $QP_{train}$ | **0.9340** | 7169.80 | 25.63 | 0.5658 | **0.8788** | 8403.05 | 29.30 | 0.6171 | **0.8500** | 9492.00 | 33.09 | 0.6308 |
| Top-2 | $KL_{test}$ | 0.5390 | 7128.62 | 25.38 | 0.6042 | 0.4868 | 7699.19 | 26.83 | 0.6241 | 0.4420 | 8062.35 | 28.15 | 0.6419 |
| | $KL_{train}$ | 0.8030 | 7177.23 | 25.53 | 0.6035 | 0.6566 | 7747.83 | 27.00 | 0.6265 | 0.5650 | 8138.65 | 28.19 | 0.6343 |
| | $QP_{test}$ | **0.6990** | **6306.09** | **22.50** | 0.5974 | **0.6320** | 6650.82 | 23.45 | 0.5752 | **0.5790** | 6674.58 | 23.25 | 0.5708 |
| | $QP_{train}$ | **0.9470** | 6676.32 | 23.66 | 0.5703 | **0.9254** | 6690.53 | 23.59 | 0.5762 | **0.9060** | 7556.91 | 26.24 | 0.5922 |
| Top-1 | $KL_{test}$ | 0.8900 | 6641.33 | 23.32 | 0.5219 | 0.8526 | 6109.33 | 21.46 | 0.5047 | **0.7900** | 6218.27 | 21.50 | 0.4227 |
| | $KL_{train}$ | **0.9970** | 5403.39 | 19.48 | 0.5228 | **0.9936** | 6164.11 | 21.63 | 0.5036 | **0.9890** | 6274.64 | 21.68 | 0.4237 |
| | $QP_{test}$ | 0.8240 | **6154.92** | 20.90 | 0.4555 | 0.7532 | 5650.84 | 19.67 | 0.4818 | 0.7130 | 5598.51 | 19.37 | 0.3887 |
| | $QP_{train}$ | 0.9810 | **4980.71** | **17.74** | 0.5246 | 0.9688 | **5695.26** | 19.80 | 0.4816 | 0.9440 | 6359.25 | 21.77 | 0.4912 |

Table 7: The ASRs and norms of learned AllAttacK perturbations across 5 runs under the single-model and image-agnostic setting: DEiT-S (*top*) and ViT-B (*bottom*), using the same 1000 training images $\mathcal{D}^{train}$ and 1000 testing images $\mathcal{D}^{test}$. The surrogate KL loss function (Eqn. 9) and the QP method (Eqn. 14) are tested and compared.

| DEiT-S | | Best | | | | Mean | | | | Worst | | | |
|---|---|---|---|---|---|---|---|---|---|---|---|---|---|
| Protocol | Attack Method | ASR↑ | $\ell_1\downarrow$ | $\ell_2\downarrow$ | $\ell_\infty\downarrow$ | ASR↑ | $\ell_1\downarrow$ | $\ell_2\downarrow$ | $\ell_\infty\downarrow$ | ASR↑ | $\ell_1\downarrow$ | $\ell_2\downarrow$ | $\ell_\infty\downarrow$ |
| Top-6 | $KL_{test}$ | 0.0370 | **7554.69** | **26.38** | **0.4735** | 0.0110 | **7884.40** | 27.78 | 0.5097 | 0.0010 | 7937.90 | 28.28 | 0.4796 |
| | $KL_{train}$ | 0.0800 | **7682.20** | 26.70 | 0.4777 | 0.0282 | 7902.49 | 27.76 | 0.5139 | 0.0060 | 7946.10 | 28.14 | 0.4714 |
| | $QP_{test}$ | 0.4910 | 20282.67 | 66.02 | 0.8132 | 0.3986 | 19017.19 | 62.38 | 0.8147 | 0.3320 | 20806.12 | 67.77 | 0.8452 |
| | $QP_{train}$ | **0.8100** | 20412.34 | 66.36 | 0.8144 | **0.6734** | 19016.97 | 62.46 | 0.8180 | **0.5960** | 20666.03 | 67.65 | 0.8526 |
| Top-5 | $KL_{test}$ | 0.0550 | 7107.77 | 24.79 | 0.5279 | 0.0328 | 7245.84 | 25.42 | 0.5014 | 0.0090 | 7181.36 | 25.27 | 0.4964 |
| | $KL_{train}$ | 0.1280 | **7240.35** | 25.80 | 0.5283 | 0.0714 | 7245.06 | 25.48 | 0.5031 | 0.0140 | 7154.82 | 25.34 | 0.5033 |
| | $QP_{test}$ | 0.6380 | 16398.45 | 54.80 | 0.7661 | 0.5770 | 16873.89 | 55.93 | 0.7629 | 0.5490 | 16274.34 | 53.26 | 0.7298 |
| | $QP_{train}$ | **0.9000** | 16384.61 | 54.83 | 0.7658 | **0.8600** | 16937.20 | 56.15 | 0.7634 | **0.8060** | 17530.56 | 58.99 | 0.8019 |
| Top-4 | $KL_{test}$ | 0.2220 | **7709.48** | 26.37 | 0.4959 | 0.1446 | **7681.20** | 26.76 | 0.5082 | 0.0540 | 7333.33 | 25.90 | 0.4734 |
| | $KL_{train}$ | 0.4170 | 7793.41 | 26.63 | 0.4990 | 0.2606 | 7717.38 | 26.88 | 0.5105 | 0.1190 | 7766.74 | 27.17 | 0.5705 |
| | $QP_{test}$ | 0.6930 | 11896.16 | 39.47 | 0.6518 | 0.6712 | 12140.38 | 40.98 | 0.6472 | 0.6320 | 13606.76 | 45.91 | 0.6304 |
| | $QP_{train}$ | **0.9540** | 11263.74 | 38.46 | 0.6589 | **0.9384** | 12220.24 | 41.22 | 0.6489 | **0.9150** | 13697.70 | 46.20 | 0.6306 |
| Top-3 | $KL_{test}$ | 0.4150 | 7385.16 | 25.97 | 0.5270 | 0.3156 | 7202.23 | 25.19 | 0.5018 | 0.2330 | 7164.11 | 25.11 | 0.4955 |
| | $KL_{train}$ | 0.6900 | 7434.29 | 26.13 | 0.5300 | 0.5194 | 7251.29 | 25.33 | 0.5040 | 0.3270 | 7221.73 | 25.28 | 0.4971 |
| | $QP_{test}$ | 0.7320 | 9604.55 | 32.99 | 0.5252 | 0.7030 | 9123.45 | 31.45 | 0.5950 | 0.6760 | 9173.88 | 31.89 | 0.6579 |
| | $QP_{train}$ | **0.9830** | 8491.79 | 28.95 | 0.6332 | **0.9526** | 9173.53 | 31.61 | 0.5979 | **0.9190** | 10105.96 | 34.59 | 0.5834 |
| Top-2 | $KL_{test}$ | 0.6230 | **6063.17** | 21.83 | 0.4454 | 0.5480 | **6082.46** | 21.44 | 0.4553 | 0.4750 | **6062.32** | 21.31 | 0.5933 |
| | $KL_{train}$ | 0.7850 | 6473.64 | 22.79 | 0.4700 | 0.7522 | 6119.41 | 21.55 | 0.4566 | 0.6720 | 6115.10 | 21.46 | 0.5974 |
| | $QP_{test}$ | 0.8170 | 8180.33 | 28.01 | 0.5119 | 0.7882 | 7959.05 | 27.24 | 0.5070 | 0.7370 | 8235.29 | 28.09 | 0.5074 |
| | $QP_{train}$ | **0.9830** | 8470.84 | 28.62 | 0.6039 | **0.9700** | 8020.04 | 27.42 | 0.5081 | **0.9650** | 8037.13 | 27.93 | 0.4882 |
| Top-1 | $KL_{test}$ | 0.9410 | 5426.09 | 19.07 | 0.3806 | 0.9268 | 5250.73 | 18.43 | 0.3864 | 0.9160 | 4674.76 | 16.71 | 0.4135 |
| | $KL_{train}$ | **0.9960** | 5632.64 | 19.94 | 0.4101 | **0.9954** | 5298.05 | 18.56 | 0.3882 | **0.9940** | 5672.40 | 19.92 | 0.4023 |
| | $QP_{test}$ | 0.9220 | **5340.78** | **17.95** | **0.2677** | 0.8984 | 5426.08 | 18.78 | **0.3371** | 0.8740 | 5448.42 | 19.34 | **0.3336** |
| | $QP_{train}$ | 0.9930 | 5409.02 | 18.92 | 0.4047 | 0.9890 | 5474.60 | 18.91 | 0.3378 | 0.9870 | 5489.21 | 19.45 | 0.3345 |

| ViT-B | | Best | | | | Mean | | | | Worst | | | |
|---|---|---|---|---|---|---|---|---|---|---|---|---|---|
| Protocol | Attack Method | ASR↑ | $\ell_1\downarrow$ | $\ell_2\downarrow$ | $\ell_\infty\downarrow$ | ASR↑ | $\ell_1\downarrow$ | $\ell_2\downarrow$ | $\ell_\infty\downarrow$ | ASR↑ | $\ell_1\downarrow$ | $\ell_2\downarrow$ | $\ell_\infty\downarrow$ |
| Top-6 | $KL_{test}$ | 0.0200 | **7426.01** | 26.81 | 0.5382 | 0.0104 | **7715.54** | 27.14 | 0.5070 | 0.0020 | 8439.95 | 29.45 | 0.5522 |
| | $KL_{train}$ | 0.0490 | 7496.11 | 26.89 | 0.5470 | 0.0256 | 7691.11 | 27.13 | 0.5092 | 0.0060 | 8257.53 | 29.00 | 0.5546 |
| | $QP_{test}$ | 0.7920 | 22361.08 | 73.51 | 0.8747 | 0.5980 | 21834.92 | 71.38 | 0.8337 | 0.4380 | 21886.20 | 71.66 | 0.8199 |
| | $QP_{train}$ | **0.9530** | 22695.33 | 74.28 | 0.8664 | **0.7764** | 22003.51 | 71.82 | 0.8317 | **0.5840** | 21893.14 | 71.70 | 0.8211 |
| Top-5 | $KL_{test}$ | 0.0820 | **7765.80** | 27.17 | 0.5143 | 0.0324 | 7587.33 | 26.60 | 0.5008 | 0.0070 | 7685.13 | 26.69 | 0.4624 |
| | $KL_{train}$ | 0.1720 | 7758.33 | 27.20 | 0.5127 | 0.0678 | 7575.71 | 26.65 | 0.5154 | 0.0150 | 7666.16 | 26.89 | 0.5264 |
| | $QP_{test}$ | 0.8300 | 19601.43 | 64.32 | 0.7475 | 0.8000 | 18771.50 | 61.81 | 0.7630 | 0.7560 | 20701.62 | 68.08 | 0.7808 |
| | $QP_{train}$ | **0.9780** | 19887.48 | 64.98 | 0.7459 | **0.9594** | 18975.86 | 62.33 | 0.7608 | **0.9300** | 20918.99 | 68.66 | 0.7791 |
| Top-4 | $KL_{test}$ | 0.1910 | 7298.67 | 25.79 | 0.4530 | 0.1210 | **7186.62** | 25.40 | 0.5047 | 0.0610 | 7383.93 | 25.95 | 0.5470 |
| | $KL_{train}$ | 0.3120 | **7268.95** | 25.74 | 0.4520 | 0.1988 | 7209.97 | 25.48 | 0.5059 | 0.0840 | 7434.38 | 26.15 | 0.5557 |
| | $QP_{test}$ | 0.8540 | 14920.53 | 49.80 | 0.6701 | 0.8258 | 15222.82 | 50.40 | 0.6790 | 0.7930 | 16756.79 | 55.47 | 0.7011 |
| | $QP_{train}$ | **0.9960** | 15109.16 | 50.28 | 0.6698 | **0.9844** | 15378.26 | 50.81 | 0.6785 | **0.9600** | 16951.09 | 56.02 | 0.7053 |
| Top-3 | $KL_{test}$ | 0.3380 | 6830.77 | 23.87 | 0.4913 | 0.2392 | 6393.40 | 22.53 | 0.4995 | 0.1680 | 6184.27 | 22.15 | 0.4857 |
| | $KL_{train}$ | 0.5160 | 6821.05 | 23.90 | 0.4939 | 0.3786 | 6411.75 | 22.59 | 0.5029 | 0.2300 | 6213.00 | 22.27 | 0.4822 |
| | $QP_{test}$ | 0.8280 | 10761.71 | 35.86 | 0.5478 | 0.7958 | 10730.18 | 36.10 | 0.5687 | 0.7470 | 10393.48 | 34.96 | 0.5189 |
| | $QP_{train}$ | **0.9870** | 10382.41 | 34.61 | 0.5879 | **0.9778** | 10814.79 | 36.33 | 0.5702 | **0.9640** | 11479.07 | 38.85 | 0.5651 |
| Top-2 | $KL_{test}$ | 0.6310 | **6390.68** | 22.27 | 0.4495 | 0.5566 | 6707.49 | 23.34 | 0.4802 | 0.3640 | 6905.25 | 23.79 | 0.5055 |
| | $KL_{train}$ | 0.8090 | **6883.84** | 23.97 | 0.4651 | 0.7038 | 6749.49 | 23.47 | 0.4822 | 0.5570 | 6940.94 | 23.91 | 0.5112 |
| | $QP_{test}$ | 0.8700 | 8503.30 | 29.21 | 0.6450 | 0.8486 | 8423.92 | 28.67 | 0.5291 | 0.8250 | 8135.73 | 27.64 | 0.5062 |
| | $QP_{train}$ | **0.9920** | 8168.44 | 27.73 | 0.5069 | **0.9802** | 8501.22 | 28.88 | 0.5316 | **0.9530** | 8690.05 | 29.08 | 0.4654 |
| Top-1 | $KL_{test}$ | 0.9690 | 6275.22 | 21.36 | 0.3308 | 0.9470 | 5775.37 | 19.88 | 0.3598 | 0.9230 | 5264.73 | 18.41 | 0.3299 |
| | $KL_{train}$ | **0.9990** | 5307.63 | 18.54 | 0.3318 | **0.9972** | 5830.60 | 20.03 | 0.3609 | **0.9950** | 6040.47 | 20.75 | 0.3667 |
| | $QP_{test}$ | 0.9740 | 6202.25 | 21.27 | 0.3790 | 0.9508 | 5846.16 | 20.02 | 0.3751 | 0.9360 | 5355.73 | 18.78 | 0.3877 |
| | $QP_{train}$ | **0.9990** | 6240.86 | 21.19 | 0.4201 | **0.9968** | 5904.46 | 20.17 | 0.3771 | **0.9950** | 5403.55 | 18.92 | 0.3893 |

Table 8: The ASRs and norms of learned AllAttacK perturbations across 5 runs under the training-model-agnostic and image-agnostic setting: 2-ConvNet combination (ResNet-50 + DenseNet-121) (*top*), and 2-ViT combination (DEiT-S + ViT-B) (*bottom*) using the same 1000 training images $\mathcal{D}^{train}$ and 1000 testing images $\mathcal{D}^{test}$. The surrogate KL loss function (Eqn. 9) and the QP method (Eqn. 14) are tested and compared.

| | | ResNet-50 \| DenseNet121 | | | | | | | | | | | |
| --- | --- | --- | --- | --- | --- | --- | --- | --- | --- | --- | --- | --- | --- |
| | | Best | | | | Mean | | | | Worst | | | |
| Protocol | Attack Method | ASR↑ | $\ell_1\downarrow$ | $\ell_2\downarrow$ | $\ell_\infty\downarrow$ | ASR↑ | $\ell_1\downarrow$ | $\ell_2\downarrow$ | $\ell_\infty\downarrow$ | ASR↑ | $\ell_1\downarrow$ | $\ell_2\downarrow$ | $\ell_\infty\downarrow$ |
| Top-6 | $KL_{test}$ | 0.0010 | **9356.27** | **32.56** | **0.6355** | 0.0004 | **9461.32** | **32.97** | **0.6443** | 0.0000 | - | - | - |
| | $KL_{train}$ | 0.0010 | **9017.74** | **31.07** | **0.6355** | 0.0002 | **9017.74** | **31.07** | **0.6355** | 0.0000 | - | - | - |
| | $QP_{test}$ | **0.1210** | 17638.06 | 58.25 | 0.7922 | **0.0928** | 20118.39 | 65.07 | 0.7873 | **0.0650** | 23475.77 | 75.12 | 0.8147 |
| | $QP_{train}$ | **0.2940** | 17570.06 | 58.05 | 0.8000 | **0.2390** | 20018.87 | 64.91 | 0.7926 | **0.1830** | 23331.62 | 74.88 | 0.8219 |
| Top-5 | $KL_{test}$ | 0.0010 | **7125.80** | **25.33** | **0.7229** | 0.0002 | **7125.80** | **25.33** | **0.7229** | 0.0000 | - | - | - |
| | $KL_{train}$ | 0.0060 | **7030.86** | **24.92** | **0.6691** | 0.0018 | **7127.44** | **25.44** | **0.6366** | 0.0000 | - | - | - |
| | $QP_{test}$ | **0.2340** | 14378.07 | 48.17 | 0.7385 | **0.1738** | 15894.99 | 52.65 | 0.7600 | **0.1200** | 18130.94 | 59.36 | 0.7935 |
| | $QP_{train}$ | **0.4870** | 14256.11 | 48.10 | 0.7472 | **0.3912** | 15851.45 | 52.58 | 0.7597 | **0.3140** | 16845.02 | 55.31 | 0.7768 |
| Top-4 | $KL_{test}$ | 0.0010 | **5967.68** | **21.30** | **0.5511** | 0.0002 | **5967.68** | **21.30** | **0.5511** | 0.0000 | - | - | - |
| | $KL_{train}$ | 0.0010 | **5438.07** | **19.75** | **0.5655** | 0.0006 | **5582.88** | **19.77** | **0.5557** | 0.0000 | - | - | - |
| | $QP_{test}$ | **0.2240** | 8699.91 | 31.07 | 0.6478 | **0.1610** | 9689.51 | 33.78 | 0.6581 | **0.1230** | 11153.02 | 38.31 | 0.6791 |
| | $QP_{train}$ | **0.4620** | 8702.42 | 31.08 | 0.6505 | **0.3562** | 9675.40 | 33.77 | 0.6597 | **0.2530** | 11066.06 | 38.16 | 0.6812 |
| Top-3 | $KL_{test}$ | 0.1080 | **8405.48** | 28.78 | **0.6556** | 0.0606 | **8816.56** | 30.53 | 0.6789 | 0.0190 | **8798.46** | 30.67 | **0.6542** |
| | $KL_{train}$ | 0.2180 | 9253.28 | 32.00 | 0.6815 | 0.1152 | **8819.96** | 30.58 | 0.6754 | 0.0310 | **8705.31** | 30.52 | **0.6471** |
| | $QP_{test}$ | **0.4000** | 8553.18 | 30.55 | 0.6594 | **0.3696** | 9650.25 | 33.47 | **0.6676** | **0.3480** | 10869.45 | 37.58 | 0.7066 |
| | $QP_{train}$ | **0.7990** | 8593.43 | 30.69 | 0.6618 | **0.7006** | 9666.14 | 33.54 | **0.6702** | **0.5950** | 10919.67 | 37.77 | 0.7156 |
| Top-2 | $KL_{test}$ | 0.2490 | 6559.76 | 23.54 | 0.6312 | 0.2190 | 6753.34 | **23.76** | 0.6168 | 0.1700 | **6671.20** | **23.67** | 0.6138 |
| | $KL_{train}$ | 0.5020 | 6572.20 | 23.60 | **0.6262** | 0.4078 | 6764.51 | 23.83 | 0.6162 | 0.3520 | **6954.27** | 24.26 | 0.5883 |
| | $QP_{test}$ | **0.5220** | **6247.31** | 22.69 | **0.6281** | **0.4458** | 6692.53 | 23.87 | 0.6200 | **0.3570** | 7388.11 | 25.65 | **0.5609** |
| | $QP_{train}$ | **0.8800** | **6267.12** | 22.77 | 0.6287 | **0.7874** | 6714.57 | 23.96 | 0.6224 | **0.6850** | 7418.76 | 25.76 | **0.5606** |
| Top-1 | $KL_{test}$ | 0.8120 | 6290.36 | 22.15 | 0.6565 | 0.7918 | 7004.87 | 24.37 | 0.5542 | 0.7740 | 7426.65 | 26.04 | 0.6312 |
| | $KL_{train}$ | 0.9950 | 7282.12 | 24.80 | **0.4442** | 0.9918 | 7061.92 | 24.55 | 0.5554 | 0.9870 | 7487.19 | 26.25 | 0.6386 |
| | $QP_{test}$ | **0.8170** | 5148.40 | 18.89 | 0.5308 | 0.7944 | 5755.58 | 20.54 | 0.5232 | 0.7720 | **6271.25** | 22.25 | 0.5924 |
| | $QP_{train}$ | **0.9800** | **5580.75** | **20.44** | 0.5485 | **0.9770** | **5799.87** | **20.68** | 0.5282 | **0.9690** | **6314.22** | **22.41** | 0.6101 |

| | | ViT-B \| DEiT-S | | | | | | | | | | | |
| --- | --- | --- | --- | --- | --- | --- | --- | --- | --- | --- | --- | --- | --- |
| | | Best | | | | Mean | | | | Worst | | | |
| Protocol | Attack Method | ASR↑ | $\ell_1\downarrow$ | $\ell_2\downarrow$ | $\ell_\infty\downarrow$ | ASR↑ | $\ell_1\downarrow$ | $\ell_2\downarrow$ | $\ell_\infty\downarrow$ | ASR↑ | $\ell_1\downarrow$ | $\ell_2\downarrow$ | $\ell_\infty\downarrow$ |
| Top-6 | $KL_{test}$ | 0.0010 | **8290.64** | **28.54** | **0.5086** | 0.0002 | **8290.64** | **28.54** | **0.5086** | 0.0000 | - | - | - |
| | $KL_{train}$ | 0.0040 | **8058.18** | **28.15** | 0.6344 | 0.0008 | **8058.18** | **28.15** | 0.6344 | 0.0000 | - | - | - |
| | $QP_{test}$ | **0.2460** | 17062.54 | 55.65 | 0.7065 | **0.1868** | 18507.49 | 60.41 | 0.7371 | **0.1210** | 20619.11 | 67.31 | 0.7552 |
| | $QP_{train}$ | **0.4890** | 17157.28 | 55.90 | 0.7045 | **0.3732** | 18500.28 | 60.50 | 0.7383 | **0.2270** | 18118.13 | 59.71 | 0.7322 |
| Top-5 | $KL_{test}$ | 0.0010 | **6009.86** | 21.23 | 0.4327 | 0.0004 | **6039.43** | 21.38 | 0.4500 | 0.0000 | - | - | - |
| | $KL_{train}$ | 0.0050 | **6015.96** | 21.36 | 0.4315 | 0.0016 | **6026.06** | 21.34 | 0.4411 | 0.0000 | - | - | - |
| | $QP_{test}$ | **0.5350** | 20824.67 | 67.14 | 0.7966 | **0.4414** | 19738.61 | 64.31 | 0.7613 | **0.3070** | 22552.63 | 73.01 | 0.8024 |
| | $QP_{train}$ | **0.8080** | 20986.76 | 67.54 | 0.7942 | **0.7092** | 19845.15 | 64.62 | 0.7611 | **0.5460** | 22755.40 | 73.54 | 0.7993 |
| Top-4 | $KL_{test}$ | 0.0050 | **4648.75** | 17.03 | 0.4104 | 0.0014 | **4655.57** | 16.68 | 0.3594 | 0.0000 | - | - | - |
| | $KL_{train}$ | 0.0110 | **4525.51** | 16.78 | 0.4107 | 0.0022 | **4525.51** | 16.78 | 0.4107 | 0.0000 | - | - | - |
| | $QP_{test}$ | **0.6690** | 15054.55 | 49.36 | 0.6590 | **0.6144** | 15706.20 | 51.85 | 0.6878 | **0.5120** | 18329.80 | 60.00 | 0.7367 |
| | $QP_{train}$ | **0.9240** | 15182.73 | 49.71 | 0.6624 | **0.8900** | 15796.71 | 52.11 | 0.6889 | **0.7880** | 18372.43 | 60.20 | 0.7375 |
| Top-3 | $KL_{test}$ | 0.1430 | 7745.26 | 27.07 | 0.5694 | 0.0998 | 7399.86 | 25.81 | 0.5240 | 0.0640 | 7610.32 | 26.10 | 0.4659 |
| | $KL_{train}$ | 0.2580 | 7478.58 | 26.50 | 0.4787 | 0.1680 | 7377.14 | 25.79 | 0.5236 | 0.0970 | **6580.82** | 23.09 | 0.5219 |
| | $QP_{test}$ | **0.7130** | 10873.59 | 36.36 | 0.7110 | **0.6748** | 11564.26 | 38.93 | 0.6013 | **0.6280** | 11112.21 | 37.90 | 0.5776 |
| | $QP_{train}$ | **0.9780** | 10982.58 | 36.64 | 0.7116 | **0.9598** | 11632.23 | 39.14 | 0.6030 | **0.9400** | 13007.96 | 43.76 | 0.5835 |
| Top-2 | $KL_{test}$ | 0.4650 | **6871.62** | 23.75 | 0.4210 | 0.3856 | **6460.49** | 22.76 | 0.4715 | 0.2670 | **6423.73** | 22.16 | 0.4382 |
| | $KL_{train}$ | 0.6350 | **6884.71** | 23.82 | 0.4220 | 0.5308 | **6494.80** | 22.87 | 0.4759 | 0.3730 | **6464.91** | 22.29 | 0.4407 |
| | $QP_{test}$ | **0.8140** | 9282.50 | 31.73 | 0.5481 | **0.7746** | 8927.34 | 30.36 | 0.5374 | **0.7010** | 8824.44 | 29.74 | 0.5456 |
| | $QP_{train}$ | **0.9780** | 8881.52 | 30.04 | 0.5869 | **0.9680** | 8995.00 | 30.56 | 0.5403 | **0.9380** | 8866.49 | 29.88 | 0.5492 |
| Top-1 | $KL_{test}$ | 0.9130 | 6514.93 | 22.55 | 0.4603 | 0.9026 | **5556.07** | 19.29 | 0.3588 | 0.8850 | **5232.72** | 17.72 | 0.2970 |
| | $KL_{train}$ | 0.9910 | **5530.50** | 19.19 | 0.3271 | 0.9872 | **5605.21** | 19.42 | 0.3592 | 0.9790 | **5297.85** | 17.89 | 0.2976 |
| | $QP_{test}$ | **0.9550** | 5908.97 | 20.34 | 0.3486 | **0.9374** | 6498.39 | 22.19 | 0.3992 | **0.9100** | 6059.03 | 20.99 | 0.4103 |
| | $QP_{train}$ | **0.9990** | 6103.75 | 21.12 | 0.4159 | **0.9952** | 6565.26 | 22.37 | 0.4023 | **0.9900** | 6912.39 | 23.65 | 0.4211 |

Table 9: The ASRs and norms of learned AllAttacK perturbations across 5 runs under the training-model-agnostic and image-agnostic setting: 4-model combination (ResNet-50 + DenseNet-121 + DEiT-S + ViT-B), using the same 1000 training images $\mathcal{D}^{train}$ and 1000 testing images $\mathcal{D}^{test}$. The surrogate KL loss function (Eqn. 9) and the QP method (Eqn. 14) are tested and compared.

| | | ResNet-50 \| DenseNet121 \| ViT-B \| DEiT-S | | | | | | | | | | | |
| | | Best | | | | Mean | | | | Worst | | | |
| Protocol | Attack Method | ASR↑ | $\ell_1\downarrow$ | $\ell_2\downarrow$ | $\ell_\infty\downarrow$ | ASR↑ | $\ell_1\downarrow$ | $\ell_2\downarrow$ | $\ell_\infty\downarrow$ | ASR↑ | $\ell_1\downarrow$ | $\ell_2\downarrow$ | $\ell_\infty\downarrow$ |
|---|---|---|---|---|---|---|---|---|---|---|---|---|---|
| Top-6 | $KL_{test}$ | 0.0000 | - | - | - | 0.0000 | - | - | - | **0.0000** | - | - | - |
| | $KL_{train}$ | 0.0000 | - | - | - | 0.0000 | - | - | - | **0.0000** | - | - | - |
| | $QP_{test}$ | **0.0060** | **17336.61** | **56.03** | **0.7074** | **0.0014** | **18560.54** | **59.83** | **0.6914** | 0.0000 | - | - | - |
| | $QP_{train}$ | **0.0180** | **17009.50** | **55.21** | **0.7027** | **0.0044** | **19591.40** | **63.24** | **0.6923** | 0.0000 | - | - | - |
| Top-5 | $KL_{test}$ | 0.0000 | - | - | - | 0.0000 | - | - | - | 0.0000 | - | - | - |
| | $KL_{train}$ | 0.0000 | - | - | - | 0.0000 | - | - | - | 0.0000 | - | - | - |
| | $QP_{test}$ | **0.0520** | **20731.71** | **66.35** | **0.7726** | **0.0322** | **21769.04** | **69.65** | **0.7793** | **0.0050** | **22758.10** | **72.59** | **0.8032** |
| | $QP_{train}$ | **0.1060** | **19371.15** | **62.88** | **0.7337** | **0.0604** | **21708.92** | **69.63** | **0.7698** | **0.0080** | **23377.91** | **74.21** | **0.7608** |
| Top-4 | $KL_{test}$ | 0.0010 | **8221.32** | **29.23** | **0.6635** | 0.0002 | **8221.32** | **29.23** | **0.6635** | 0.0000 | - | - | - |
| | $KL_{train}$ | 0.0000 | - | - | - | 0.0000 | - | - | - | 0.0000 | - | - | - |
| | $QP_{test}$ | **0.1540** | 16601.63 | 54.34 | 0.6983 | **0.1012** | 17423.69 | 56.58 | 0.7051 | **0.0320** | 18550.70 | 59.86 | 0.6865 |
| | $QP_{train}$ | **0.3770** | 15173.79 | 49.93 | 0.6892 | **0.2258** | 17362.30 | 56.54 | 0.7063 | **0.0810** | 18634.20 | 60.05 | 0.6941 |
| Top-3 | $KL_{test}$ | 0.0010 | 5483.35 | 19.72 | 0.5098 | 0.0002 | 5483.35 | 19.72 | 0.5098 | 0.0000 | - | - | - |
| | $KL_{train}$ | 0.0040 | **5452.43** | **19.61** | **0.4976** | 0.0010 | **5500.95** | **19.64** | **0.4636** | 0.0000 | - | - | - |
| | $QP_{test}$ | **0.2610** | 12237.89 | 41.09 | 0.6227 | **0.2102** | 12886.82 | 43.00 | 0.6612 | **0.1650** | 13393.16 | 44.74 | 0.6865 |
| | $QP_{train}$ | **0.5390** | 11825.12 | 39.82 | 0.5864 | **0.4316** | 12886.71 | 43.03 | 0.6617 | **0.3320** | 14420.31 | 48.00 | 0.6750 |
| Top-2 | $KL_{test}$ | 0.1470 | **8023.93** | 27.98 | 0.6367 | 0.1030 | **8000.37** | 28.05 | 0.6309 | 0.0610 | **7856.85** | 27.70 | **0.5808** |
| | $KL_{train}$ | 0.3120 | 8036.94 | 28.14 | 0.6382 | 0.1972 | **8053.00** | **28.24** | 0.6354 | 0.0820 | **7962.50** | **27.94** | **0.5875** |
| | $QP_{test}$ | **0.3580** | 8215.26 | 28.22 | **0.6251** | **0.3282** | 8567.16 | 29.60 | **0.5876** | **0.2760** | 8940.55 | 30.70 | 0.6406 |
| | $QP_{train}$ | **0.6600** | 7910.85 | 27.69 | 0.5018 | **0.5888** | 8584.08 | 29.66 | **0.5871** | **0.5210** | 8973.04 | 30.79 | 0.6443 |
| Top-1 | $KL_{test}$ | 0.7430 | 6616.65 | 23.57 | 0.5230 | 0.7142 | **6941.49** | **24.08** | 0.5202 | 0.6810 | 7340.90 | **25.55** | 0.5562 |
| | $KL_{train}$ | 0.9690 | **6664.76** | 23.73 | 0.5238 | 0.9494 | **6982.48** | 24.21 | 0.5223 | 0.9210 | **6886.12** | 23.29 | 0.5242 |
| | $QP_{test}$ | **0.8200** | 8509.04 | 28.54 | **0.5084** | **0.7570** | 8160.65 | 27.64 | **0.4988** | **0.6980** | 7872.09 | 26.91 | **0.5107** |
| | $QP_{train}$ | **0.9730** | 8609.30 | 28.82 | **0.5122** | **0.9538** | 8223.69 | 27.83 | **0.4996** | **0.9310** | 7930.71 | 27.07 | **0.5099** |

Table 10: Results of the 5 ResNets in the 18-model ensemble AllAttacK.

**ResNet-18**

| Protocol | Attack Method | Best | | | | Mean | | | | Worst | | | |
|---|---|---|---|---|---|---|---|---|---|---|---|---|---|
| | | ASR↑ | $\ell_1\downarrow$ | $\ell_2\downarrow$ | $\ell_\infty\downarrow$ | ASR↑ | $\ell_1\downarrow$ | $\ell_2\downarrow$ | $\ell_\infty\downarrow$ | ASR↑ | $\ell_1\downarrow$ | $\ell_2\downarrow$ | $\ell_\infty\downarrow$ |
| Top-3 | $KL_{test}$ | 0.4118 | 43424.02 | 136.44 | 0.9870 | 0.1076 | 16861.33 | 55.21 | 0.7161 | 0.0089 | **10485.64** | **35.72** | 0.6911 |
| | $KL_{train}$ | 0.5580 | 43999.30 | 136.90 | 0.9860 | 0.1568 | 16940.45 | 55.28 | 0.7134 | 0.0190 | **10436.65** | **35.81** | 0.6738 |
| | $QP_{test}$ | **0.9107** | **35913.60** | **114.89** | **0.9733** | 0.2525 | **15421.16** | **51.42** | **0.6866** | **0.0357** | 10679.11 | 36.95 | **0.6222** |
| | $QP_{train}$ | **0.9940** | **36669.07** | **116.07** | **0.9709** | **0.3336** | **15604.25** | **51.80** | **0.6851** | **0.0660** | 10635.25 | 36.96 | **0.6205** |
| Top-2 | $KL_{test}$ | **0.9944** | 41464.60 | 132.13 | 0.9886 | 0.3987 | 15964.81 | 53.03 | 0.7342 | 0.1763 | 9539.21 | 33.11 | **0.5806** |
| | $KL_{train}$ | **0.9970** | 42338.41 | 132.99 | 0.9856 | 0.4792 | 16181.23 | 53.36 | 0.7335 | 0.2450 | 9671.39 | 33.53 | 0.5805 |
| | $QP_{test}$ | 0.9375 | **31589.33** | **101.97** | **0.9664** | 0.4531 | **13808.98** | **46.36** | **0.6936** | 0.2467 | 9433.12 | 32.68 | 0.6909 |
| | $QP_{train}$ | 0.9960 | **32282.84** | **103.16** | **0.9627** | **0.5970** | **13975.49** | **46.71** | **0.6912** | **0.3830** | 9252.20 | 32.41 | **0.5658** |
| Top-1 | $KL_{test}$ | **0.8783** | 9570.48 | 33.29 | 0.5938 | **0.8520** | 9096.83 | 31.59 | 0.6240 | **0.8170** | 8962.19 | 31.31 | 0.6716 |
| | $KL_{train}$ | **0.9840** | 9683.44 | 33.59 | 0.5908 | **0.9644** | 9193.77 | 31.88 | 0.6258 | **0.9530** | 9038.70 | 31.57 | 0.6771 |
| | $QP_{test}$ | 0.7299 | **7066.08** | **24.73** | **0.4559** | 0.6592 | 7245.04 | 25.07 | 0.4931 | 0.5480 | 7549.05 | 25.81 | **0.4553** |
| | $QP_{train}$ | 0.8850 | **7123.41** | **24.88** | **0.4541** | 0.8254 | 7287.84 | 25.20 | 0.4931 | 0.7850 | 7281.04 | 25.23 | 0.4871 |

**ResNet-34**

| Protocol | Attack Method | Best | | | | Mean | | | | Worst | | | |
|---|---|---|---|---|---|---|---|---|---|---|---|---|---|
| | | ASR↑ | $\ell_1\downarrow$ | $\ell_2\downarrow$ | $\ell_\infty\downarrow$ | ASR↑ | $\ell_1\downarrow$ | $\ell_2\downarrow$ | $\ell_\infty\downarrow$ | ASR↑ | $\ell_1\downarrow$ | $\ell_2\downarrow$ | $\ell_\infty\downarrow$ |
| Top-3 | $KL_{test}$ | 0.0837 | **9628.84** | **32.71** | **0.6444** | 0.0507 | 16166.52 | 54.49 | 0.7231 | 0.0156 | **10261.54** | **35.12** | 0.6478 |
| | $KL_{train}$ | 0.1260 | **9719.70** | **33.05** | **0.6425** | 0.0792 | 16217.00 | 54.71 | 0.7216 | 0.0270 | **10273.97** | **35.13** | 0.6472 |
| | $QP_{test}$ | **0.9431** | 35866.28 | 114.83 | 0.9739 | 0.2911 | 15413.17 | 51.42 | 0.6879 | 0.0580 | 10499.24 | 35.72 | **0.6123** |
| | $QP_{train}$ | **0.9950** | 36657.31 | 116.05 | 0.9709 | **0.3936** | 15580.64 | 51.76 | 0.6865 | **0.1270** | 10365.32 | 35.54 | **0.6068** |
| Top-2 | $KL_{test}$ | 0.5391 | 40957.37 | 131.74 | 0.9916 | 0.3547 | 15888.77 | 53.02 | 0.7347 | 0.2310 | 9709.71 | 33.54 | 0.5817 |
| | $KL_{train}$ | 0.5670 | 41949.08 | 132.70 | 0.9872 | 0.4884 | 16135.37 | 53.38 | 0.7340 | 0.4130 | 9728.55 | 33.65 | 0.5817 |
| | $QP_{test}$ | **0.9431** | **31574.64** | **101.96** | **0.9664** | 0.4596 | **13787.04** | **46.32** | **0.6936** | 0.3181 | 9197.24 | 32.21 | **0.5644** |
| | $QP_{train}$ | **0.9980** | **32273.97** | **103.15** | **0.9627** | **0.6236** | **13968.65** | **46.69** | **0.6923** | **0.4500** | 9243.07 | 32.38 | 0.5669 |
| Top-1 | $KL_{test}$ | **0.8683** | 8676.72 | 30.18 | 0.6375 | **0.8411** | 9098.94 | 31.60 | 0.6243 | **0.8203** | 8967.47 | 31.32 | 0.6727 |
| | $KL_{train}$ | **0.9850** | 8426.15 | 29.66 | 0.6242 | **0.9670** | 9189.54 | 31.87 | 0.6261 | **0.9550** | 10008.52 | 34.03 | 0.5974 |
| | $QP_{test}$ | 0.7478 | **7074.05** | **24.84** | **0.5439** | 0.6792 | 7230.03 | 25.03 | 0.4933 | 0.5357 | 7517.93 | 25.73 | **0.4553** |
| | $QP_{train}$ | 0.9430 | **7123.41** | **25.00** | **0.5397** | 0.8582 | 7284.29 | 25.19 | 0.4933 | 0.7880 | 7566.40 | 25.90 | **0.4553** |

**ResNet-50**

| Protocol | Attack Method | Best | | | | Mean | | | | Worst | | | |
|---|---|---|---|---|---|---|---|---|---|---|---|---|---|
| | | ASR↑ | $\ell_1\downarrow$ | $\ell_2\downarrow$ | $\ell_\infty\downarrow$ | ASR↑ | $\ell_1\downarrow$ | $\ell_2\downarrow$ | $\ell_\infty\downarrow$ | ASR↑ | $\ell_1\downarrow$ | $\ell_2\downarrow$ | $\ell_\infty\downarrow$ |
| Top-3 | $KL_{test}$ | 0.2478 | 42058.93 | 135.47 | 0.9914 | 0.1096 | 16468.79 | 54.70 | 0.7204 | 0.0335 | **10177.76** | **34.90** | 0.6401 |
| | $KL_{train}$ | 0.2490 | 42807.82 | 136.05 | 0.9907 | 0.1494 | 16632.95 | 54.93 | 0.7202 | 0.0390 | **10249.42** | **35.27** | 0.6418 |
| | $QP_{test}$ | **0.9152** | **35910.64** | **114.88** | **0.9735** | 0.2797 | 15450.61 | 51.54 | **0.6847** | 0.0592 | 10459.88 | 35.72 | **0.6056** |
| | $QP_{train}$ | **0.9940** | 36664.97 | 116.05 | 0.9709 | **0.3746** | 15605.31 | 51.83 | **0.6856** | **0.1370** | 10631.63 | 37.04 | 0.6232 |
| Top-2 | $KL_{test}$ | 0.6942 | 41508.02 | 132.15 | 0.9878 | 0.4277 | 15982.01 | 53.07 | 0.7327 | 0.2422 | 9677.96 | 33.47 | 0.6934 |
| | $KL_{train}$ | 0.6730 | 42553.44 | 133.16 | 0.9846 | 0.5454 | 16216.55 | 53.38 | 0.7329 | 0.4120 | 9729.21 | 33.67 | 0.6941 |
| | $QP_{test}$ | **0.9230** | **31618.51** | **102.03** | **0.9663** | 0.4844 | **13803.22** | **46.34** | **0.6926** | 0.2835 | 9540.52 | 32.88 | **0.6575** |
| | $QP_{train}$ | **0.9920** | **32287.04** | **103.18** | **0.9627** | **0.6368** | **13965.67** | **46.67** | **0.6914** | **0.4500** | 9562.55 | 32.93 | 0.6553 |
| Top-1 | $KL_{test}$ | **0.8806** | 9898.85 | 33.71 | 0.5919 | **0.8295** | 9102.70 | 31.61 | 0.6242 | **0.7411** | 8377.73 | 29.50 | 0.6247 |
| | $KL_{train}$ | **0.9910** | 9682.78 | 33.59 | 0.5909 | **0.9746** | 9189.40 | 31.87 | 0.6260 | **0.9610** | 8429.89 | 29.67 | 0.6240 |
| | $QP_{test}$ | 0.7176 | **7215.88** | **25.04** | **0.4861** | 0.6408 | 7241.52 | 25.07 | 0.4930 | 0.5257 | 7537.87 | 25.78 | **0.4548** |
| | $QP_{train}$ | 0.8820 | **7121.04** | **25.00** | **0.5390** | 0.8496 | 7281.86 | 25.19 | 0.4932 | 0.8170 | 7557.57 | 25.87 | 0.4549 |

**ResNet-50$_2$**

| Protocol | Attack Method | Best | | | | Mean | | | | Worst | | | |
|---|---|---|---|---|---|---|---|---|---|---|---|---|---|
| | | ASR↑ | $\ell_1\downarrow$ | $\ell_2\downarrow$ | $\ell_\infty\downarrow$ | ASR↑ | $\ell_1\downarrow$ | $\ell_2\downarrow$ | $\ell_\infty\downarrow$ | ASR↑ | $\ell_1\downarrow$ | $\ell_2\downarrow$ | $\ell_\infty\downarrow$ |
| Top-3 | $KL_{test}$ | 0.4922 | 43314.78 | 136.38 | 0.9865 | 0.1888 | 16622.63 | **54.63** | 0.7257 | 0.0223 | **10241.99** | **35.11** | 0.7067 |
| | $KL_{train}$ | 0.6340 | 43989.06 | 136.86 | 0.9852 | 0.2400 | 16792.10 | 54.91 | 0.7219 | 0.0540 | **10224.57** | **35.28** | 0.6919 |
| | $QP_{test}$ | **0.8482** | 36015.86 | 115.02 | 0.9733 | 0.2214 | 16501.96 | 54.81 | **0.7077** | 0.0000 | - | - | - |
| | $QP_{train}$ | **0.9330** | 36728.70 | 116.16 | 0.9704 | **0.2752** | 16783.98 | 55.35 | **0.7016** | 0.0000 | - | - | - |
| Top-2 | $KL_{test}$ | 0.5737 | **9654.96** | **33.43** | **0.6958** | 0.3768 | 15881.10 | 52.91 | 0.7350 | 0.1373 | 9799.31 | 34.27 | 0.6715 |
| | $KL_{train}$ | 0.6230 | **9752.39** | **33.69** | **0.6926** | 0.4476 | 16112.29 | 53.25 | 0.7337 | 0.2230 | 9811.92 | 34.37 | 0.6717 |
| | $QP_{test}$ | **0.8248** | 31640.58 | 102.09 | 0.9662 | 0.3725 | **13746.04** | **46.20** | 0.6955 | 0.1283 | 9229.26 | 31.88 | **0.5870** |
| | $QP_{train}$ | **0.9180** | 32338.13 | 103.27 | 0.9620 | **0.4882** | 13928.84 | 46.56 | 0.6930 | **0.2090** | 9262.76 | 32.01 | 0.5883 |
| Top-1 | $KL_{test}$ | **0.9330** | 8326.29 | 29.36 | 0.6256 | **0.8908** | 9069.21 | 31.53 | 0.6240 | **0.8304** | 8936.87 | 31.25 | 0.6714 |
| | $KL_{train}$ | **0.9910** | 9678.14 | 33.58 | 0.5913 | **0.9700** | 9186.05 | 31.86 | 0.6259 | **0.9320** | 9025.93 | 31.53 | 0.6765 |
| | $QP_{test}$ | 0.6551 | **7044.36** | **24.77** | **0.5433** | 0.5292 | 7221.52 | 25.02 | 0.4928 | 0.3873 | 7491.44 | 25.66 | **0.4547** |
| | $QP_{train}$ | 0.7800 | **7117.37** | **24.97** | **0.5372** | 0.6464 | 7277.07 | 25.18 | 0.4923 | 0.5230 | 7541.82 | 25.84 | **0.4541** |

**ResNet-101**

| Protocol | Attack Method | Best | | | | Mean | | | | Worst | | | |
|---|---|---|---|---|---|---|---|---|---|---|---|---|---|
| | | ASR↑ | $\ell_1\downarrow$ | $\ell_2\downarrow$ | $\ell_\infty\downarrow$ | ASR↑ | $\ell_1\downarrow$ | $\ell_2\downarrow$ | $\ell_\infty\downarrow$ | ASR↑ | $\ell_1\downarrow$ | $\ell_2\downarrow$ | $\ell_\infty\downarrow$ |
| Top-3 | $KL_{test}$ | 0.4699 | 42901.18 | 135.98 | 0.9878 | 0.1489 | 16658.74 | 54.84 | 0.7196 | 0.0435 | **10249.30** | **35.07** | 0.6395 |
| | $KL_{train}$ | 0.6370 | 43734.60 | 136.65 | 0.9855 | 0.2170 | 16848.79 | 55.11 | 0.7202 | 0.0740 | **10257.94** | **35.25** | 0.6420 |
| | $QP_{test}$ | **0.9129** | 35967.69 | 114.97 | 0.9736 | 0.2839 | 15386.84 | 51.36 | 0.6869 | 0.0670 | 10394.04 | 36.29 | **0.6253** |
| | $QP_{train}$ | **0.9960** | 36650.70 | 116.04 | 0.9709 | **0.3812** | 15574.71 | 51.73 | 0.6867 | **0.1200** | 10616.62 | 36.89 | **0.6216** |
| Top-2 | $KL_{test}$ | 0.7288 | 41469.52 | 132.16 | 0.9887 | 0.4337 | 15985.99 | 53.08 | 0.7316 | 0.2779 | 9626.07 | 33.33 | 0.6935 |
| | $KL_{train}$ | 0.7680 | 42310.90 | 133.02 | 0.9865 | 0.5496 | 16192.13 | 53.40 | 0.7328 | 0.4230 | **9268.07** | **32.06** | 0.7335 |
| | $QP_{test}$ | **0.9096** | **31595.47** | **102.00** | **0.9662** | 0.4580 | **13808.78** | **46.36** | **0.6934** | 0.3080 | 9400.61 | 32.62 | 0.6910 |
| | $QP_{train}$ | **0.9980** | **32274.16** | **103.15** | **0.9627** | **0.6526** | **13961.57** | **46.67** | **0.6919** | **0.5200** | 9382.35 | 32.64 | 0.6841 |
| Top-1 | $KL_{test}$ | **0.8873** | 9546.63 | 33.23 | 0.5941 | **0.8232** | 9104.84 | 31.61 | 0.6238 | **0.7333** | 8383.35 | 29.51 | 0.6247 |
| | $KL_{train}$ | **0.9920** | 9680.22 | 33.58 | 0.5911 | **0.9644** | 9188.99 | 31.87 | 0.6261 | **0.9420** | 8791.88 | 30.51 | 0.6405 |
| | $QP_{test}$ | 0.7310 | **7221.36** | **25.05** | **0.4859** | 0.6435 | 7243.10 | 25.07 | 0.4934 | 0.5792 | 7305.36 | 24.89 | 0.5267 |
| | $QP_{train}$ | 0.8880 | **7123.64** | **25.00** | **0.5380** | 0.8292 | 7282.06 | 25.19 | 0.4933 | 0.7400 | 7332.77 | 24.97 | 0.5324 |

Table 11: Results of the 4 DenseNets in the 18-model ensemble AllAttacK.

**DenseNet121**

| Protocol | Attack Method | Best | | | | Mean | | | | Worst | | | |
|---|---|---|---|---|---|---|---|---|---|---|---|---|---|
| | | ASR↑ | $\ell_1\downarrow$ | $\ell_2\downarrow$ | $\ell_\infty\downarrow$ | ASR↑ | $\ell_1\downarrow$ | $\ell_2\downarrow$ | $\ell_\infty\downarrow$ | ASR↑ | $\ell_1\downarrow$ | $\ell_2\downarrow$ | $\ell_\infty\downarrow$ |
| Top-3 | $KL_{test}$ | 0.4241 | 42980.81 | 136.04 | 0.9872 | 0.1230 | 16614.73 | 54.72 | 0.7185 | 0.0335 | **9732.79** | **33.83** | 0.6468 |
| | $KL_{train}$ | 0.5770 | 43675.11 | 136.65 | 0.9852 | 0.1854 | 16842.51 | 55.11 | 0.7179 | 0.0450 | **9980.72** | **34.57** | 0.6487 |
| | $QP_{test}$ | **0.9364** | **35888.01** | **114.86** | **0.9738** | **0.3201** | **15433.60** | **51.47** | **0.6858** | 0.0625 | 10515.34 | 35.75 | **0.6149** |
| | $QP_{train}$ | **0.9970** | 36651.15 | 116.04 | 0.9709 | **0.4320** | 15593.72 | 51.78 | **0.6870** | **0.1150** | 10617.15 | 36.92 | **0.6263** |
| Top-2 | $KL_{test}$ | 0.6518 | 41516.16 | 132.27 | 0.9891 | 0.3955 | 15944.04 | 52.98 | 0.7353 | 0.2321 | 9631.21 | 33.37 | 0.6953 |
| | $KL_{train}$ | 0.7620 | 42395.59 | 133.11 | 0.9859 | 0.4980 | 16190.08 | 53.38 | 0.7338 | 0.2650 | 9724.76 | 33.63 | 0.6936 |
| | $QP_{test}$ | **0.9118** | **31606.91** | **102.02** | **0.9660** | 0.5364 | **13760.54** | **46.24** | **0.6941** | 0.2935 | 9188.19 | 32.21 | 0.5643 |
| | $QP_{train}$ | **0.9920** | 32301.29 | 103.20 | 0.9626 | **0.6940** | 13946.59 | 46.62 | 0.6919 | **0.4400** | 9230.05 | 32.34 | 0.5643 |
| Top-1 | $KL_{test}$ | 0.8873 | 8676.47 | 30.18 | 0.6369 | 0.8645 | 9076.95 | 31.55 | 0.6244 | 0.8080 | 9897.05 | 33.71 | 0.5922 |
| | $KL_{train}$ | 0.9900 | 8425.99 | 29.66 | 0.6243 | **0.9752** | 9188.67 | 31.87 | 0.6261 | **0.9400** | 10016.17 | 34.05 | 0.5973 |
| | $QP_{test}$ | 0.7467 | 7052.40 | 24.80 | **0.5438** | 0.6643 | 7221.68 | 25.01 | **0.4932** | 0.6127 | 7053.45 | 24.71 | 0.4563 |
| | $QP_{train}$ | 0.9150 | 7118.62 | 24.99 | 0.5394 | 0.8392 | 7278.52 | 25.18 | 0.4931 | 0.7850 | 7278.90 | 25.23 | 0.4873 |

**DenseNet161**

| Protocol | Attack Method | Best | | | | Mean | | | | Worst | | | |
|---|---|---|---|---|---|---|---|---|---|---|---|---|---|
| | | ASR↑ | $\ell_1\downarrow$ | $\ell_2\downarrow$ | $\ell_\infty\downarrow$ | ASR↑ | $\ell_1\downarrow$ | $\ell_2\downarrow$ | $\ell_\infty\downarrow$ | ASR↑ | $\ell_1\downarrow$ | $\ell_2\downarrow$ | $\ell_\infty\downarrow$ |
| Top-3 | $KL_{test}$ | 0.4487 | 42839.08 | 135.92 | 0.9880 | 0.1464 | 16530.50 | 54.54 | 0.7226 | 0.0525 | **10154.03** | **34.91** | 0.6920 |
| | $KL_{train}$ | 0.4880 | 43558.41 | 136.50 | 0.9862 | 0.1894 | 16758.92 | 54.94 | 0.7198 | 0.1040 | **10240.53** | **35.23** | 0.6406 |
| | $QP_{test}$ | **0.9297** | **35838.69** | **114.81** | **0.9739** | **0.2938** | **15362.39** | **51.32** | **0.6870** | 0.0692 | 10549.87 | 36.71 | **0.6204** |
| | $QP_{train}$ | **1.0000** | 36646.21 | 116.03 | 0.9710 | **0.4044** | 15568.96 | 51.72 | **0.6855** | **0.1590** | 10582.79 | 36.90 | **0.6250** |
| Top-2 | $KL_{test}$ | **0.9654** | 41516.79 | 132.17 | 0.9884 | 0.5056 | 15969.56 | 53.03 | 0.7331 | **0.3326** | 9843.51 | 34.37 | 0.6680 |
| | $KL_{train}$ | 0.9880 | 42350.34 | 133.01 | 0.9856 | 0.6282 | 16185.79 | 53.36 | 0.7329 | **0.5040** | 9247.14 | **32.00** | 0.7364 |
| | $QP_{test}$ | 0.9475 | **31535.84** | **101.90** | **0.9666** | 0.5107 | **13763.33** | **46.25** | **0.6937** | 0.2835 | 9175.11 | 32.15 | 0.5623 |
| | $QP_{train}$ | **0.9960** | 32277.96 | 103.16 | 0.9627 | **0.6954** | 13950.96 | 46.64 | 0.6917 | 0.4640 | **9239.06** | 32.37 | 0.5666 |
| Top-1 | $KL_{test}$ | **0.8817** | 8683.09 | 30.19 | 0.6370 | **0.8672** | 9079.22 | 31.55 | 0.6243 | **0.8571** | 8940.17 | 31.26 | 0.6728 |
| | $KL_{train}$ | **0.9910** | 9028.33 | 31.54 | 0.6779 | **0.9802** | 9186.75 | 31.86 | 0.6261 | **0.9690** | 8792.63 | 30.51 | 0.6400 |
| | $QP_{test}$ | 0.7210 | **7217.14** | 25.05 | **0.4861** | 0.6368 | **7233.16** | 25.04 | **0.4930** | 0.5714 | 7513.88 | 25.72 | **0.4552** |
| | $QP_{train}$ | 0.9000 | 7126.82 | 25.00 | 0.5389 | 0.8314 | 7281.12 | 25.18 | 0.4931 | 0.8050 | 7119.53 | 24.87 | 0.4539 |

**DenseNet169**

| Protocol | Attack Method | Best | | | | Mean | | | | Worst | | | |
|---|---|---|---|---|---|---|---|---|---|---|---|---|---|
| | | ASR↑ | $\ell_1\downarrow$ | $\ell_2\downarrow$ | $\ell_\infty\downarrow$ | ASR↑ | $\ell_1\downarrow$ | $\ell_2\downarrow$ | $\ell_\infty\downarrow$ | ASR↑ | $\ell_1\downarrow$ | $\ell_2\downarrow$ | $\ell_\infty\downarrow$ |
| Top-3 | $KL_{test}$ | 0.6663 | 43221.14 | 136.28 | 0.9872 | 0.2304 | 16693.91 | 54.84 | 0.7211 | 0.0658 | 10254.43 | 35.10 | **0.6454** |
| | $KL_{train}$ | 0.8320 | 43823.25 | 136.75 | 0.9856 | 0.3268 | 16851.44 | 55.10 | 0.7189 | 0.0920 | **10278.86** | **35.30** | 0.6489 |
| | $QP_{test}$ | **0.9330** | **35842.76** | **114.79** | **0.9737** | 0.3181 | **15405.76** | **51.42** | **0.6853** | 0.1295 | 10088.45 | 34.99 | 0.6481 |
| | $QP_{train}$ | **0.9960** | 36655.30 | 116.05 | 0.9709 | **0.4198** | 15573.00 | 51.73 | **0.6866** | **0.2310** | 10524.42 | 36.70 | **0.6287** |
| Top-2 | $KL_{test}$ | 0.8315 | 41623.82 | 132.28 | 0.9875 | 0.4714 | 15965.24 | 52.98 | 0.7342 | 0.2634 | 9590.63 | 33.22 | 0.6957 |
| | $KL_{train}$ | 0.9310 | 42402.13 | 133.07 | 0.9849 | 0.5842 | 16199.60 | 53.37 | 0.7347 | 0.3510 | 9768.37 | 33.70 | 0.6938 |
| | $QP_{test}$ | **0.9219** | **31609.59** | **102.01** | **0.9661** | 0.5522 | **13753.34** | **46.20** | **0.6948** | 0.4118 | 9472.02 | 32.69 | 0.6623 |
| | $QP_{train}$ | **0.9960** | 32283.11 | 103.16 | 0.9627 | **0.7430** | 13940.92 | 46.61 | 0.6926 | **0.6110** | 9553.49 | 32.93 | 0.6583 |
| Top-1 | $KL_{test}$ | **0.8984** | 9544.75 | 33.23 | 0.5943 | **0.8766** | 9082.49 | 31.56 | 0.6243 | **0.8460** | 8347.16 | 29.41 | 0.6257 |
| | $KL_{train}$ | **0.9860** | 9030.65 | 31.55 | 0.6777 | **0.9786** | 9187.60 | 31.87 | 0.6261 | **0.9650** | 10010.82 | 34.04 | 0.5973 |
| | $QP_{test}$ | 0.7522 | **7048.35** | 24.69 | **0.4560** | 0.7121 | **7220.10** | 25.01 | **0.4930** | 0.6529 | 7283.95 | 24.83 | 0.5240 |
| | $QP_{train}$ | 0.8920 | 7114.45 | 24.86 | 0.4543 | 0.8614 | 7278.10 | 25.18 | 0.4933 | 0.8110 | 7273.19 | 25.22 | 0.4871 |

**DenseNet201**

| Protocol | Attack Method | Best | | | | Mean | | | | Worst | | | |
|---|---|---|---|---|---|---|---|---|---|---|---|---|---|
| | | ASR↑ | $\ell_1\downarrow$ | $\ell_2\downarrow$ | $\ell_\infty\downarrow$ | ASR↑ | $\ell_1\downarrow$ | $\ell_2\downarrow$ | $\ell_\infty\downarrow$ | ASR↑ | $\ell_1\downarrow$ | $\ell_2\downarrow$ | $\ell_\infty\downarrow$ |
| Top-3 | $KL_{test}$ | 0.3717 | 43063.55 | 136.00 | 0.9863 | 0.1272 | 16662.74 | 54.80 | 0.7225 | 0.0391 | **10035.16** | **34.65** | 0.6429 |
| | $KL_{train}$ | 0.5150 | 43756.76 | 136.62 | 0.9853 | 0.1792 | 16804.21 | 55.01 | 0.7188 | 0.0460 | **10260.34** | **35.36** | 0.6421 |
| | $QP_{test}$ | **0.9364** | **35869.21** | **114.84** | **0.9739** | 0.3123 | **15372.93** | **51.33** | **0.6850** | 0.0714 | 10531.07 | 36.58 | **0.6184** |
| | $QP_{train}$ | **0.9950** | 36649.87 | 116.04 | 0.9710 | **0.4222** | 15542.45 | 51.65 | **0.6855** | **0.1350** | 10485.59 | 36.59 | **0.6242** |
| Top-2 | $KL_{test}$ | 0.7768 | 41568.42 | 132.19 | 0.9885 | 0.4092 | 15955.73 | 52.97 | 0.7368 | 0.2266 | 9193.70 | **31.82** | 0.7453 |
| | $KL_{train}$ | 0.8300 | 42358.44 | 133.03 | 0.9854 | 0.5152 | 16168.42 | 53.32 | 0.7364 | 0.2720 | **9234.37** | 31.98 | 0.7501 |
| | $QP_{test}$ | **0.9308** | **31575.42** | **101.96** | **0.9663** | 0.5359 | **13765.89** | **46.25** | **0.6938** | 0.3940 | 9165.40 | 32.15 | 0.5633 |
| | $QP_{train}$ | **0.9980** | 32279.66 | 103.16 | 0.9627 | **0.7068** | 13943.57 | 46.63 | 0.6912 | **0.5740** | 9560.68 | 32.96 | 0.6539 |
| Top-1 | $KL_{test}$ | **0.9062** | 9541.95 | 33.22 | 0.5942 | **0.8692** | 9080.10 | 31.56 | 0.6243 | **0.8304** | 9900.18 | 33.72 | 0.5920 |
| | $KL_{train}$ | **0.9910** | 9679.35 | 33.58 | 0.5910 | **0.9710** | 9186.99 | 31.86 | 0.6259 | **0.9440** | 10009.61 | 34.04 | 0.5972 |
| | $QP_{test}$ | 0.6920 | **7062.60** | 24.83 | **0.5430** | 0.6609 | **7231.54** | 25.04 | **0.4933** | 0.5714 | 7313.70 | 24.90 | 0.5266 |
| | $QP_{train}$ | 0.9040 | 7117.67 | 24.98 | 0.5391 | 0.8208 | 7279.18 | 25.18 | 0.4933 | 0.7400 | 7277.26 | 25.22 | 0.4871 |

Table 12: Results of the HRNet-W18 and 3 ConvNeXts in the 18-model ensemble AllAttacK.

**HRNet-W18**

| Protocol | Attack Method | Best ASR↑ | $\ell_1\downarrow$ | $\ell_2\downarrow$ | $\ell_\infty\downarrow$ | Mean ASR↑ | $\ell_1\downarrow$ | $\ell_2\downarrow$ | $\ell_\infty\downarrow$ | Worst ASR↑ | $\ell_1\downarrow$ | $\ell_2\downarrow$ | $\ell_\infty\downarrow$ |
|---|---|---|---|---|---|---|---|---|---|---|---|---|---|
| Top-3 | $KL_{test}$ | 0.0033 | 10420.89 | 35.56 | 0.6851 | 0.0009 | 10324.34 | 35.31 | 0.6759 | **0.0000** | - | - | - |
| | $KL_{train}$ | 0.0020 | 10514.89 | 35.80 | 0.6103 | 0.0004 | 10514.89 | 35.80 | 0.6103 | **0.0000** | - | - | - |
| | $QP_{test}$ | **0.0201** | **10109.50** | **35.04** | **0.5743** | 0.0040 | 10109.50 | 35.04 | 0.5743 | **0.0000** | - | - | - |
| | $QP_{train}$ | **0.0400** | 10308.71 | 35.52 | 0.5728 | 0.0080 | 10308.71 | 35.52 | 0.5728 | **0.0000** | - | - | - |
| Top-2 | $KL_{test}$ | 0.1138 | 9888.54 | 34.40 | 0.6686 | 0.0806 | 9584.90 | 33.22 | 0.6636 | **0.0000** | - | - | - |
| | $KL_{train}$ | 0.1840 | 9943.49 | 34.68 | 0.6675 | 0.1142 | 9700.18 | 33.57 | 0.6660 | **0.0000** | - | - | - |
| | $QP_{test}$ | **0.2946** | 9276.55 | 32.06 | 0.5867 | 0.1371 | 9367.11 | 32.45 | 0.6256 | **0.0000** | - | - | - |
| | $QP_{train}$ | **0.4330** | 9337.91 | 32.22 | 0.5878 | 0.2172 | 9417.57 | 32.64 | 0.6220 | **0.0000** | - | - | - |
| Top-1 | $KL_{test}$ | **0.8047** | 8701.76 | 30.23 | 0.6378 | **0.6556** | 9117.55 | 31.64 | 0.6227 | 0.4375 | 8985.29 | 31.34 | 0.6655 |
| | $KL_{train}$ | 0.9350 | 8429.00 | 29.66 | 0.6240 | 0.8332 | 9196.55 | 31.89 | 0.6245 | 0.6580 | 9032.92 | 31.56 | 0.6721 |
| | $QP_{test}$ | 0.5290 | **7210.77** | **25.04** | **0.4855** | 0.4464 | **7239.52** | 25.06 | 0.4920 | 0.2690 | 7507.79 | 25.71 | 0.4544 |
| | $QP_{train}$ | 0.7540 | **7135.19** | **25.02** | **0.5360** | 0.6360 | 7279.95 | 25.18 | 0.4920 | 0.4780 | 7527.42 | 25.81 | 0.4536 |

**ConvNeXt-T**

| Protocol | Attack Method | Best ASR↑ | $\ell_1\downarrow$ | $\ell_2\downarrow$ | $\ell_\infty\downarrow$ | Mean ASR↑ | $\ell_1\downarrow$ | $\ell_2\downarrow$ | $\ell_\infty\downarrow$ | Worst ASR↑ | $\ell_1\downarrow$ | $\ell_2\downarrow$ | $\ell_\infty\downarrow$ |
|---|---|---|---|---|---|---|---|---|---|---|---|---|---|
| Top-3 | $KL_{test}$ | 0.8471 | 42835.37 | 135.97 | 0.9884 | 0.3404 | 16593.09 | 54.73 | 0.7216 | 0.1228 | 10150.04 | 34.86 | 0.6412 |
| | $KL_{train}$ | 0.8860 | 43641.31 | 136.59 | 0.9862 | 0.4240 | 16762.49 | 54.93 | 0.7227 | 0.1830 | 10155.68 | 35.02 | 0.6448 |
| | $QP_{test}$ | **0.9944** | **35710.10** | **114.64** | **0.9748** | 0.2703 | 15350.56 | 51.25 | 0.6871 | 0.0123 | 10509.20 | 36.18 | **0.6276** |
| | $QP_{train}$ | **1.0000** | 36646.22 | 116.03 | 0.9710 | 0.3390 | 15595.78 | 51.78 | 0.6870 | 0.0220 | 10583.16 | 36.75 | 0.6345 |
| Top-2 | $KL_{test}$ | 0.8940 | 41559.50 | 132.24 | 0.9882 | 0.5768 | 15945.11 | 52.96 | 0.7336 | 0.4342 | 9770.34 | 34.17 | 0.6712 |
| | $KL_{train}$ | 0.9590 | 42359.72 | 133.03 | 0.9853 | 0.6914 | 16182.04 | 53.35 | 0.7330 | 0.4720 | 9752.06 | 33.66 | 0.6927 |
| | $QP_{test}$ | **0.9967** | 31435.69 | 101.75 | 0.9676 | 0.5109 | **13758.97** | **46.27** | **0.6936** | 0.3192 | 9226.28 | 32.30 | **0.5629** |
| | $QP_{train}$ | **1.0000** | 32269.97 | 103.14 | 0.9628 | 0.6340 | **13966.56** | 46.70 | 0.6912 | 0.4680 | 9273.99 | 32.50 | 0.5653 |
| Top-1 | $KL_{test}$ | **0.9498** | 8648.65 | 30.11 | 0.6373 | **0.9192** | 9067.07 | 31.52 | 0.6245 | **0.8661** | 9899.85 | 33.72 | 0.5923 |
| | $KL_{train}$ | **0.9930** | 8789.17 | 30.50 | 0.6403 | **0.9732** | 9188.63 | 31.87 | 0.6261 | **0.9370** | 10011.70 | 34.04 | 0.5974 |
| | $QP_{test}$ | 0.5915 | **7490.26** | **25.67** | **0.4552** | 0.4759 | **7253.81** | 25.10 | 0.4934 | 0.3058 | 7278.80 | 25.22 | 0.4871 |
| | $QP_{train}$ | 0.7640 | **7550.62** | **25.85** | **0.4547** | 0.5818 | 7305.54 | 25.27 | 0.4940 | 0.3570 | 7310.52 | 25.33 | 0.4875 |

**ConvNeXt-S**

| Protocol | Attack Method | Best ASR↑ | $\ell_1\downarrow$ | $\ell_2\downarrow$ | $\ell_\infty\downarrow$ | Mean ASR↑ | $\ell_1\downarrow$ | $\ell_2\downarrow$ | $\ell_\infty\downarrow$ | Worst ASR↑ | $\ell_1\downarrow$ | $\ell_2\downarrow$ | $\ell_\infty\downarrow$ |
|---|---|---|---|---|---|---|---|---|---|---|---|---|---|
| Top-3 | $KL_{test}$ | 0.8672 | 42974.20 | 136.06 | 0.9883 | 0.3214 | 16622.99 | 54.76 | 0.7200 | 0.1406 | 10168.16 | 34.92 | 0.6392 |
| | $KL_{train}$ | 0.9430 | 43641.98 | 136.60 | 0.9863 | 0.4028 | 16772.54 | 54.96 | 0.7206 | 0.2030 | 9677.32 | 32.93 | 0.6414 |
| | $QP_{test}$ | **0.9933** | **35720.41** | **114.65** | **0.9748** | 0.2408 | 15389.55 | 51.44 | 0.6849 | 0.0067 | 10758.47 | 37.26 | **0.6232** |
| | $QP_{train}$ | **1.0000** | 36646.21 | 116.03 | 0.9710 | 0.2884 | 15585.74 | 51.74 | 0.6881 | 0.0220 | 10526.73 | 36.58 | 0.6302 |
| Top-2 | $KL_{test}$ | 0.8996 | 41620.68 | 132.23 | 0.9880 | 0.5437 | 15970.53 | 52.99 | 0.7336 | 0.3973 | 9185.46 | 31.79 | 0.7333 |
| | $KL_{train}$ | 0.9330 | 42394.31 | 133.03 | 0.9856 | 0.6292 | 16198.86 | 53.38 | 0.7335 | 0.4090 | 9702.29 | 33.55 | 0.6914 |
| | $QP_{test}$ | **0.9978** | 31431.94 | 101.74 | 0.9676 | 0.4795 | **13748.97** | **46.25** | **0.6941** | 0.2589 | 9245.51 | 31.96 | **0.5854** |
| | $QP_{train}$ | **1.0000** | 32269.98 | 103.14 | 0.9628 | 0.5496 | 13970.45 | 46.71 | 0.6905 | 0.2430 | 9314.50 | 32.22 | 0.5875 |
| Top-1 | $KL_{test}$ | **0.9375** | 8918.89 | 31.21 | 0.6738 | **0.9143** | 9072.65 | 31.53 | 0.6247 | **0.8828** | 8342.73 | 29.41 | 0.6256 |
| | $KL_{train}$ | **0.9880** | 8789.17 | 30.50 | 0.6402 | **0.9732** | 9187.32 | 31.87 | 0.6260 | **0.9460** | 8430.27 | 29.67 | 0.6240 |
| | $QP_{test}$ | 0.6272 | **7075.12** | **24.77** | **0.4561** | 0.4933 | **7256.15** | 25.11 | 0.4938 | 0.3627 | 7303.93 | 25.28 | 0.4865 |
| | $QP_{train}$ | 0.7720 | **7124.11** | **24.89** | **0.4548** | 0.6142 | 7304.81 | 25.26 | 0.4941 | 0.4930 | 7337.11 | 25.00 | 0.5326 |

**ConvNeXt-B**

| Protocol | Attack Method | Best ASR↑ | $\ell_1\downarrow$ | $\ell_2\downarrow$ | $\ell_\infty\downarrow$ | Mean ASR↑ | $\ell_1\downarrow$ | $\ell_2\downarrow$ | $\ell_\infty\downarrow$ | Worst ASR↑ | $\ell_1\downarrow$ | $\ell_2\downarrow$ | $\ell_\infty\downarrow$ |
|---|---|---|---|---|---|---|---|---|---|---|---|---|---|
| Top-3 | $KL_{test}$ | 0.8438 | 43061.05 | 136.12 | 0.9879 | 0.3513 | 16628.36 | 54.77 | 0.7195 | **0.1641** | 10112.01 | 34.83 | 0.6391 |
| | $KL_{train}$ | 0.9350 | 43657.10 | 136.61 | 0.9859 | 0.4272 | 16762.22 | 54.95 | 0.7206 | **0.2250** | 10194.74 | 35.11 | 0.6465 |
| | $QP_{test}$ | **0.9866** | **35730.24** | **114.66** | **0.9747** | 0.2504 | 15376.51 | 51.41 | 0.6868 | 0.0335 | 10600.57 | 36.91 | **0.6233** |
| | $QP_{train}$ | **1.0000** | 36646.22 | 116.03 | 0.9710 | 0.3000 | 15568.98 | 51.75 | 0.6839 | 0.0720 | 10527.04 | 36.70 | 0.6246 |
| Top-2 | $KL_{test}$ | 0.9085 | 41744.18 | 132.31 | 0.9878 | 0.5417 | 15988.36 | 52.99 | 0.7344 | **0.3728** | 9637.31 | 33.37 | 0.6956 |
| | $KL_{train}$ | 0.9510 | 42486.12 | 133.10 | 0.9851 | 0.6558 | 16216.38 | 53.40 | 0.7326 | **0.3970** | 9746.37 | 33.67 | 0.6916 |
| | $QP_{test}$ | **0.9933** | 31442.47 | 101.76 | 0.9676 | 0.4373 | **13752.34** | **46.26** | **0.6940** | 0.2600 | 9267.04 | 32.00 | **0.5862** |
| | $QP_{train}$ | **1.0000** | 32269.98 | 103.14 | 0.9628 | 0.5028 | 13959.06 | 46.68 | 0.6910 | 0.2920 | 9297.12 | 32.16 | 0.5875 |
| Top-1 | $KL_{test}$ | **0.9520** | 9518.96 | 33.16 | 0.5942 | **0.9194** | 9062.52 | 31.51 | 0.6245 | **0.8683** | 8336.12 | 29.38 | 0.6252 |
| | $KL_{train}$ | **0.9930** | 9676.56 | 33.57 | 0.5910 | **0.9714** | 9187.14 | 31.87 | 0.6260 | **0.9300** | 8430.28 | 29.67 | 0.6241 |
| | $QP_{test}$ | 0.5346 | 7535.08 | 25.79 | **0.4551** | 0.4741 | **7253.90** | 25.11 | 0.4941 | 0.3750 | 7287.91 | 24.86 | 0.5272 |
| | $QP_{train}$ | 0.6730 | **7140.85** | **24.95** | **0.4552** | 0.6064 | 7308.35 | 25.28 | 0.4943 | 0.5170 | 7355.06 | 25.04 | 0.5341 |

Table 13: Results of the 3 DEiTs in the 18-model ensemble AllAttacK.

**DEiT-S**

| Protocol | Attack Method | Best | | | | Mean | | | | Worst | | | |
|---|---|---|---|---|---|---|---|---|---|---|---|---|---|
| | | ASR↑ | $\ell_1\downarrow$ | $\ell_2\downarrow$ | $\ell_\infty\downarrow$ | ASR↑ | $\ell_1\downarrow$ | $\ell_2\downarrow$ | $\ell_\infty\downarrow$ | ASR↑ | $\ell_1\downarrow$ | $\ell_2\downarrow$ | $\ell_\infty\downarrow$ |
| Top-3 | $KL_{test}$ | 0.7489 | 43018.77 | 136.10 | 0.9882 | 0.2462 | 16668.20 | 54.85 | 0.7170 | 0.0413 | 10287.89 | 35.19 | 0.6308 |
| | $KL_{train}$ | 0.8160 | 43682.62 | 136.66 | 0.9854 | 0.3372 | 16786.58 | 55.00 | 0.7168 | 0.0820 | 10180.55 | 35.06 | 0.6283 |
| | $QP_{test}$ | 0.9888 | 35696.10 | 114.62 | 0.9748 | 0.4810 | 15310.89 | 51.23 | 0.6869 | 0.2824 | 10121.96 | 35.12 | 0.5763 |
| | $QP_{train}$ | 0.9990 | 36647.35 | 116.04 | 0.9710 | 0.6634 | 15538.81 | 51.65 | 0.6867 | 0.4810 | 10292.76 | 35.33 | 0.6083 |
| Top-2 | $KL_{test}$ | 0.9554 | 41545.30 | 132.20 | 0.9883 | 0.5225 | 15964.92 | 53.02 | 0.7336 | 0.2991 | 9163.59 | 31.76 | 0.7330 |
| | $KL_{train}$ | 0.9930 | 42348.47 | 133.00 | 0.9855 | 0.6484 | 16199.13 | 53.42 | 0.7327 | 0.3330 | 9290.64 | 32.14 | 0.7368 |
| | $QP_{test}$ | 0.9955 | 31434.24 | 101.74 | 0.9676 | 0.6839 | 13705.50 | 46.13 | 0.6948 | 0.5045 | 9199.50 | 31.86 | 0.5862 |
| | $QP_{train}$ | 1.0000 | 32269.98 | 103.14 | 0.9628 | 0.8332 | 13942.33 | 46.62 | 0.6926 | 0.6390 | 9297.81 | 32.13 | 0.5878 |
| Top-1 | $KL_{test}$ | 0.8281 | 8927.29 | 31.23 | 0.6718 | 0.7853 | 9100.77 | 31.60 | 0.6245 | 0.7254 | 9608.39 | 33.37 | 0.5957 |
| | $KL_{train}$ | 0.9530 | 8436.33 | 29.68 | 0.6242 | 0.8970 | 9197.78 | 31.89 | 0.6254 | 0.8010 | 9717.01 | 33.68 | 0.5898 |
| | $QP_{test}$ | 0.7121 | 7088.34 | 24.88 | 0.5429 | 0.6556 | 7215.05 | 25.00 | 0.4922 | 0.6094 | 7235.95 | 24.72 | 0.5202 |
| | $QP_{train}$ | 0.9390 | 7121.76 | 25.00 | 0.5392 | 0.7850 | 7282.78 | 25.19 | 0.4928 | 0.6740 | 7340.99 | 24.98 | 0.5286 |

**DeiT3-S**

| Protocol | Attack Method | Best | | | | Mean | | | | Worst | | | |
|---|---|---|---|---|---|---|---|---|---|---|---|---|---|
| | | ASR↑ | $\ell_1\downarrow$ | $\ell_2\downarrow$ | $\ell_\infty\downarrow$ | ASR↑ | $\ell_1\downarrow$ | $\ell_2\downarrow$ | $\ell_\infty\downarrow$ | ASR↑ | $\ell_1\downarrow$ | $\ell_2\downarrow$ | $\ell_\infty\downarrow$ |
| Top-3 | $KL_{test}$ | 0.4509 | 42491.13 | 135.70 | 0.9901 | 0.1545 | 18144.89 | 59.67 | 0.7276 | 0.0000 | - | - | - |
| | $KL_{train}$ | 0.4590 | 43138.54 | 136.21 | 0.9880 | 0.1778 | 16776.99 | 55.17 | 0.6950 | 0.0010 | 10512.81 | 36.04 | 0.5725 |
| | $QP_{test}$ | 0.9978 | 35689.85 | 114.61 | 0.9749 | 0.5886 | 15301.01 | 51.19 | 0.6861 | 0.4040 | 10262.21 | 35.17 | 0.6080 |
| | $QP_{train}$ | 1.0000 | 36646.22 | 116.03 | 0.9710 | 0.7218 | 15555.13 | 51.67 | 0.6864 | 0.5520 | 10333.47 | 35.37 | 0.6076 |
| Top-2 | $KL_{test}$ | 0.9565 | 41531.36 | 132.18 | 0.9883 | 0.4857 | 16002.11 | 53.10 | 0.7324 | 0.2277 | 9963.28 | 34.69 | 0.6665 |
| | $KL_{train}$ | 0.9660 | 42394.58 | 133.05 | 0.9853 | 0.5758 | 16224.20 | 53.46 | 0.7323 | 0.3110 | 9977.45 | 34.84 | 0.6664 |
| | $QP_{test}$ | 1.0000 | 31426.25 | 101.73 | 0.9677 | 0.7417 | 13717.81 | 46.15 | 0.6943 | 0.5134 | 9368.94 | 32.50 | 0.6930 |
| | $QP_{train}$ | 1.0000 | 32269.98 | 103.14 | 0.9628 | 0.8516 | 13955.15 | 46.65 | 0.6923 | 0.6610 | 9406.71 | 32.68 | 0.6876 |
| Top-1 | $KL_{test}$ | 0.9475 | 8331.28 | 29.37 | 0.6255 | 0.9103 | 9072.65 | 31.54 | 0.6247 | 0.8761 | 8931.07 | 31.24 | 0.6738 |
| | $KL_{train}$ | 0.9710 | 10017.37 | 34.06 | 0.5974 | 0.9566 | 9193.09 | 31.88 | 0.6259 | 0.9390 | 8790.45 | 30.50 | 0.6399 |
| | $QP_{test}$ | 0.8895 | 7191.41 | 24.98 | 0.4860 | 0.8145 | 7207.28 | 24.98 | 0.4938 | 0.7243 | 7261.30 | 24.78 | 0.5262 |
| | $QP_{train}$ | 0.9390 | 7266.79 | 25.21 | 0.4873 | 0.8964 | 7279.57 | 25.18 | 0.4934 | 0.8320 | 7333.50 | 24.96 | 0.5315 |

**DeiT3-M**

| Protocol | Attack Method | Best | | | | Mean | | | | Worst | | | |
|---|---|---|---|---|---|---|---|---|---|---|---|---|---|
| | | ASR↑ | $\ell_1\downarrow$ | $\ell_2\downarrow$ | $\ell_\infty\downarrow$ | ASR↑ | $\ell_1\downarrow$ | $\ell_2\downarrow$ | $\ell_\infty\downarrow$ | ASR↑ | $\ell_1\downarrow$ | $\ell_2\downarrow$ | $\ell_\infty\downarrow$ |
| Top-3 | $KL_{test}$ | 0.2768 | 41202.53 | 134.76 | 0.9921 | 0.0554 | 41202.53 | 134.76 | 0.9921 | 0.0000 | - | - | - |
| | $KL_{train}$ | 0.2730 | 42455.26 | 135.74 | 0.9918 | 0.0552 | 26018.27 | 84.10 | 0.8208 | 0.0000 | - | - | - |
| | $QP_{test}$ | 1.0000 | 35685.10 | 114.60 | 0.9749 | 0.2025 | 22984.78 | 75.11 | 0.7744 | 0.0000 | - | - | - |
| | $QP_{train}$ | 1.0000 | 36646.22 | 116.03 | 0.9710 | 0.2046 | 23465.52 | 75.78 | 0.7744 | 0.0000 | - | - | - |
| Top-2 | $KL_{test}$ | 0.6283 | 41371.44 | 131.94 | 0.9885 | 0.1632 | 19397.14 | 64.69 | 0.7870 | 0.0000 | - | - | - |
| | $KL_{train}$ | 0.5060 | 42469.34 | 133.03 | 0.9861 | 0.1528 | 17582.40 | 57.43 | 0.7808 | 0.0000 | - | - | - |
| | $QP_{test}$ | 1.0000 | 31426.25 | 101.73 | 0.9677 | 0.4174 | 13725.73 | 46.18 | 0.6937 | 0.1094 | 9120.65 | 32.01 | 0.5593 |
| | $QP_{train}$ | 1.0000 | 32269.98 | 103.14 | 0.9628 | 0.5188 | 14001.64 | 46.78 | 0.6913 | 0.0780 | 9406.47 | 32.82 | 0.5629 |
| Top-1 | $KL_{test}$ | 0.9275 | 8641.64 | 30.10 | 0.6375 | 0.7116 | 9090.80 | 31.59 | 0.6220 | 0.3783 | 9004.43 | 31.41 | 0.6623 |
| | $KL_{train}$ | 0.9720 | 8790.32 | 30.50 | 0.6400 | 0.8448 | 9192.51 | 31.88 | 0.6246 | 0.6360 | 9031.65 | 31.55 | 0.6726 |
| | $QP_{test}$ | 0.7087 | 7057.89 | 24.82 | 0.5449 | 0.4743 | 7204.25 | 24.98 | 0.4929 | 0.1775 | 7493.28 | 25.67 | 0.4539 |
| | $QP_{train}$ | 0.8830 | 7123.51 | 25.00 | 0.5387 | 0.6268 | 7281.08 | 25.19 | 0.4927 | 0.3610 | 7552.86 | 25.87 | 0.4532 |

Table 14: Results of the ViT-B and MlpMixer-B in the 18-model ensemble AllAttacK.

**ViT-B**

| Protocol | Attack Method | Best | | | | Mean | | | | Worst | | | |
|---|---|---|---|---|---|---|---|---|---|---|---|---|---|
| | | ASR↑ | $\ell_1\downarrow$ | $\ell_2\downarrow$ | $\ell_\infty\downarrow$ | ASR↑ | $\ell_1\downarrow$ | $\ell_2\downarrow$ | $\ell_\infty\downarrow$ | ASR↑ | $\ell_1\downarrow$ | $\ell_2\downarrow$ | $\ell_\infty\downarrow$ |
| Top-3 | $KL_{test}$ | 0.8482 | 42903.41 | 136.04 | 0.9884 | 0.2192 | 16577.01 | 54.71 | 0.7212 | 0.0335 | 10017.73 | 34.68 | 0.6433 |
| | $KL_{train}$ | 0.9110 | 43652.71 | 136.62 | 0.9858 | 0.2506 | 16760.20 | 54.97 | 0.7226 | 0.0400 | 9990.42 | 34.76 | 0.6436 |
| | $QP_{test}$ | 0.9933 | 35699.80 | 114.62 | 0.9748 | 0.4991 | 15315.13 | 51.23 | 0.6870 | 0.3158 | 10496.03 | 36.54 | 0.6228 |
| | $QP_{train}$ | 0.9990 | 36645.57 | 116.03 | 0.9709 | 0.6838 | 15552.83 | 51.68 | 0.6868 | 0.5120 | 10579.38 | 36.84 | 0.6267 |
| Top-2 | $KL_{test}$ | 0.9654 | 41535.71 | 132.19 | 0.9884 | 0.3790 | 15964.40 | 53.02 | 0.7333 | 0.1562 | 9187.78 | 31.83 | 0.7315 |
| | $KL_{train}$ | 0.9910 | 42349.49 | 133.00 | 0.9855 | 0.4492 | 16196.67 | 53.40 | 0.7322 | 0.2470 | 9689.99 | 33.55 | 0.5822 |
| | $QP_{test}$ | 0.9955 | 31440.41 | 101.75 | 0.9676 | 0.6732 | 13708.89 | 46.13 | 0.6951 | 0.5536 | 9169.76 | 31.78 | 0.5866 |
| | $QP_{train}$ | 1.0000 | 32269.98 | 103.14 | 0.9628 | 0.8278 | 13949.34 | 46.64 | 0.6926 | 0.6470 | 9307.80 | 32.16 | 0.5882 |
| Top-1 | $KL_{test}$ | 0.7891 | 9578.80 | 33.30 | 0.5950 | 0.7183 | 9103.36 | 31.62 | 0.6238 | 0.5569 | 8385.41 | 29.53 | 0.6260 |
| | $KL_{train}$ | 0.9110 | 10018.33 | 34.06 | 0.5972 | 0.8658 | 9199.93 | 31.90 | 0.6251 | 0.8000 | 8802.26 | 30.55 | 0.6390 |
| | $QP_{test}$ | 0.7400 | 7219.84 | 25.05 | 0.4853 | 0.7154 | 7219.63 | 25.01 | 0.4932 | 0.6786 | 7075.21 | 24.86 | 0.5450 |
| | $QP_{train}$ | 0.9170 | 7265.84 | 25.20 | 0.4872 | 0.8814 | 7280.67 | 25.19 | 0.4931 | 0.8620 | 7115.35 | 24.86 | 0.4544 |

**MlpMixer-B**

| Protocol | Attack Method | Best | | | | Mean | | | | Worst | | | |
|---|---|---|---|---|---|---|---|---|---|---|---|---|---|
| | | ASR↑ | $\ell_1\downarrow$ | $\ell_2\downarrow$ | $\ell_\infty\downarrow$ | ASR↑ | $\ell_1\downarrow$ | $\ell_2\downarrow$ | $\ell_\infty\downarrow$ | ASR↑ | $\ell_1\downarrow$ | $\ell_2\downarrow$ | $\ell_\infty\downarrow$ |
| Top-3 | $KL_{test}$ | 0.4643 | 43219.71 | 136.19 | 0.9875 | 0.1033 | 21074.36 | 68.15 | 0.7545 | 0.0000 | - | - | - |
| | $KL_{train}$ | 0.4520 | 44003.84 | 136.81 | 0.9851 | 0.1056 | 16910.44 | 55.26 | 0.7034 | 0.0010 | 10680.91 | 36.49 | 0.6510 |
| | $QP_{test}$ | 0.0435 | 10039.03 | 34.88 | 0.6521 | 0.0179 | 10453.45 | 36.09 | 0.6002 | 0.0000 | - | - | - |
| | $QP_{train}$ | 0.0840 | 10215.04 | 35.35 | 0.6465 | 0.0336 | 10360.78 | 35.64 | 0.6147 | 0.0000 | - | - | - |
| Top-2 | $KL_{test}$ | 0.1496 | 9983.38 | 34.78 | 0.6642 | 0.0879 | 9682.25 | 33.55 | 0.6624 | 0.0000 | - | - | - |
| | $KL_{train}$ | 0.2000 | 9925.96 | 34.75 | 0.6638 | 0.1114 | 9723.57 | 33.73 | 0.6636 | 0.0000 | - | - | - |
| | $QP_{test}$ | 0.2667 | 9607.19 | 33.05 | 0.6563 | 0.1547 | 12993.21 | 45.23 | 0.6955 | 0.0335 | 27291.52 | 95.71 | 0.9880 |
| | $QP_{train}$ | 0.3860 | 9638.89 | 33.18 | 0.6544 | 0.2100 | 13411.17 | 45.85 | 0.6931 | 0.0330 | 29372.07 | 98.47 | 0.9813 |
| Top-1 | $KL_{test}$ | 0.5658 | 9030.95 | 31.49 | 0.6698 | 0.4614 | 9182.98 | 31.84 | 0.6229 | 0.2824 | 10006.06 | 34.02 | 0.5919 |
| | $KL_{train}$ | 0.6760 | 8494.16 | 29.86 | 0.6240 | 0.5086 | 9245.34 | 32.05 | 0.6240 | 0.3640 | 9793.77 | 33.92 | 0.5873 |
| | $QP_{test}$ | 0.2768 | 7281.86 | 25.22 | 0.4848 | 0.2127 | 7299.70 | 25.23 | 0.4932 | 0.1384 | 7347.16 | 24.98 | 0.5221 |
| | $QP_{train}$ | 0.3360 | 7177.63 | 25.20 | 0.5368 | 0.2486 | 7308.03 | 25.31 | 0.4923 | 0.1040 | 7342.19 | 25.09 | 0.5304 |

Table 15: Results of the three **unseen** testing DNNs (ConvMixer-768, SWin-B and HRNet-30) in the 18-model ensemble AllAttacK.

**ConvMixer-768**

| Protocol | Attack Method | Best | | | | Mean | | | | Worst | | | |
|---|---|---|---|---|---|---|---|---|---|---|---|---|---|
| | | ASR↑ | $\ell_1\downarrow$ | $\ell_2\downarrow$ | $\ell_\infty\downarrow$ | ASR↑ | $\ell_1\downarrow$ | $\ell_2\downarrow$ | $\ell_\infty\downarrow$ | ASR↑ | $\ell_1\downarrow$ | $\ell_2\downarrow$ | $\ell_\infty\downarrow$ |
| Top-3 | $KL_{test}$ | **0.0982** | 10142.63 | 34.94 | 0.6343 | **0.0339** | 21069.72 | 68.20 | 0.7551 | **0.0000** | - | - | - |
| | $KL_{train}$ | **0.1710** | **10106.77** | **34.87** | 0.6432 | **0.0510** | 20969.48 | 67.99 | 0.7570 | **0.0000** | - | - | - |
| | $QP_{test}$ | 0.0636 | **9989.84** | **34.76** | **0.5766** | 0.0150 | **18538.55** | **61.49** | 0.7428 | **0.0000** | - | - | - |
| | $QP_{train}$ | 0.1180 | 10138.39 | 35.13 | **0.5769** | 0.0300 | **19172.00** | **62.27** | 0.7386 | **0.0000** | - | - | - |
| Top-2 | $KL_{test}$ | 0.5558 | 41975.76 | 132.67 | 0.9870 | 0.2308 | 16077.85 | 53.16 | 0.7328 | **0.0960** | 9625.95 | 33.27 | **0.6898** |
| | $KL_{train}$ | 0.6660 | 42651.81 | 133.34 | 0.9834 | 0.3328 | 16202.02 | 53.33 | 0.7333 | **0.1660** | 9575.57 | 33.25 | 0.6934 |
| | $QP_{test}$ | **0.7522** | **31651.29** | **102.06** | 0.9646 | 0.2900 | **13737.18** | **46.13** | 0.6945 | 0.0625 | 9297.55 | 32.23 | 0.7035 |
| | $QP_{train}$ | **0.8310** | 32373.43 | 103.34 | 0.9613 | 0.3730 | 13958.04 | 46.63 | 0.6924 | 0.1460 | 9296.54 | 32.34 | 0.6980 |
| Top-1 | $KL_{test}$ | 0.8125 | 8333.09 | 29.38 | 0.6251 | **0.6703** | 9098.49 | 31.60 | 0.6236 | **0.4955** | 8942.69 | 31.28 | 0.6709 |
| | $KL_{train}$ | **0.9270** | 8428.17 | 29.67 | 0.6240 | **0.7852** | 9194.93 | 31.89 | 0.6244 | **0.5760** | 9026.39 | 31.56 | 0.6717 |
| | $QP_{test}$ | 0.6373 | **7037.36** | **24.75** | **0.5447** | 0.3708 | **7224.56** | **25.03** | **0.4926** | 0.2567 | **7303.90** | **24.90** | 0.5218 |
| | $QP_{train}$ | 0.7340 | **7124.40** | **25.00** | **0.5383** | 0.4524 | **7279.79** | **25.19** | **0.4919** | 0.3280 | **7525.14** | **25.82** | **0.4533** |

**SWin-B**

| Protocol | Attack Method | Best | | | | Mean | | | | Worst | | | |
|---|---|---|---|---|---|---|---|---|---|---|---|---|---|
| | | ASR↑ | $\ell_1\downarrow$ | $\ell_2\downarrow$ | $\ell_\infty\downarrow$ | ASR↑ | $\ell_1\downarrow$ | $\ell_2\downarrow$ | $\ell_\infty\downarrow$ | ASR↑ | $\ell_1\downarrow$ | $\ell_2\downarrow$ | $\ell_\infty\downarrow$ |
| Top-3 | $KL_{test}$ | 0.0502 | 43172.79 | 136.17 | 0.9901 | 0.0228 | **16662.27** | 54.78 | 0.7137 | 0.0022 | 10015.45 | 34.49 | 0.5918 |
| | $KL_{train}$ | 0.0850 | **10115.34** | **35.01** | **0.6469** | 0.0330 | 18422.92 | 59.84 | 0.7481 | 0.0000 | - | - | - |
| | $QP_{test}$ | **0.3002** | **35995.68** | **114.79** | 0.9740 | 0.0679 | 16671.67 | 55.21 | **0.7018** | 0.0000 | - | - | - |
| | $QP_{train}$ | **0.2740** | 36935.15 | 116.20 | 0.9702 | 0.0680 | **15551.76** | **51.54** | **0.6936** | 0.0010 | 10317.88 | 35.73 | 0.6444 |
| Top-2 | $KL_{test}$ | 0.3225 | 42166.83 | 132.72 | 0.9866 | 0.1667 | 16143.42 | 53.27 | 0.7371 | 0.0748 | 9859.18 | 34.43 | 0.6736 |
| | $KL_{train}$ | 0.3350 | 42639.94 | 133.26 | 0.9852 | 0.1906 | 16308.45 | 53.61 | 0.7322 | **0.0940** | 9999.30 | 34.89 | 0.6715 |
| | $QP_{test}$ | **0.9196** | **31568.16** | **101.93** | 0.9662 | 0.2589 | **13809.34** | **46.39** | 0.6945 | 0.0826 | 9596.37 | 33.03 | 0.6614 |
| | $QP_{train}$ | **0.8830** | 32450.70 | 103.41 | 0.9611 | 0.2738 | 14038.73 | 46.90 | 0.6908 | 0.0920 | 9332.18 | 32.71 | 0.5666 |
| Top-1 | $KL_{test}$ | 0.7824 | 9930.26 | 33.79 | 0.5909 | 0.5935 | 9148.25 | 31.74 | 0.6226 | 0.3940 | 8408.84 | 29.61 | 0.6239 |
| | $KL_{train}$ | **0.8760** | 10037.36 | 34.10 | 0.5970 | **0.7098** | 9228.72 | 32.00 | 0.6246 | **0.5980** | 8463.16 | 29.78 | 0.6234 |
| | $QP_{test}$ | 0.3203 | **7280.77** | **25.25** | **0.4851** | 0.2243 | **7292.25** | **25.25** | **0.4933** | 0.1574 | **7361.81** | **25.07** | 0.5260 |
| | $QP_{train}$ | 0.3840 | **7325.08** | **25.38** | **0.4870** | 0.2416 | **7327.70** | **25.37** | **0.4931** | 0.1360 | **7565.33** | **25.97** | **0.4523** |

**HRNet-W30**

| Protocol | Attack Method | Best | | | | Mean | | | | Worst | | | |
|---|---|---|---|---|---|---|---|---|---|---|---|---|---|
| | | ASR↑ | $\ell_1\downarrow$ | $\ell_2\downarrow$ | $\ell_\infty\downarrow$ | ASR↑ | $\ell_1\downarrow$ | $\ell_2\downarrow$ | $\ell_\infty\downarrow$ | ASR↑ | $\ell_1\downarrow$ | $\ell_2\downarrow$ | $\ell_\infty\downarrow$ |
| Top-3 | $KL_{test}$ | **0.0045** | 10276.44 | 35.28 | 0.6617 | **0.0016** | 10128.63 | 34.55 | 0.6938 | **0.0000** | - | - | - |
| | $KL_{train}$ | 0.0050 | 10607.65 | 36.15 | 0.6607 | 0.0018 | 10322.33 | 35.45 | 0.6612 | **0.0000** | - | - | - |
| | $QP_{test}$ | 0.0022 | **9844.63** | **34.10** | **0.5799** | 0.0004 | **9844.63** | **34.10** | **0.5799** | **0.0000** | - | - | - |
| | $QP_{train}$ | 0.0130 | 10204.23 | 35.18 | **0.5773** | 0.0026 | 10204.23 | 35.18 | **0.5773** | **0.0000** | - | - | - |
| Top-2 | $KL_{test}$ | **0.1931** | 9725.59 | 33.59 | **0.5821** | 0.0654 | 9487.07 | 32.90 | 0.6591 | **0.0000** | - | - | - |
| | $KL_{train}$ | **0.2400** | 9821.72 | 33.90 | **0.5811** | 0.0834 | 9616.60 | 33.37 | 0.6671 | **0.0000** | - | - | - |
| | $QP_{test}$ | 0.1518 | **9370.52** | **32.28** | 0.5868 | 0.0942 | **9409.03** | **32.59** | 0.6227 | **0.0000** | - | - | - |
| | $QP_{train}$ | 0.2270 | **9401.19** | **32.36** | 0.5875 | 0.1526 | 9428.82 | 32.68 | 0.6223 | **0.0000** | - | - | - |
| Top-1 | $KL_{test}$ | **0.8359** | 8678.02 | 30.18 | 0.6372 | **0.6362** | 9124.14 | 31.67 | 0.6228 | **0.4319** | 9639.00 | 33.48 | 0.5927 |
| | $KL_{train}$ | **0.9210** | 8791.17 | 30.50 | 0.6400 | **0.8126** | 9195.47 | 31.89 | 0.6245 | **0.6770** | 9026.14 | 31.55 | 0.6731 |
| | $QP_{test}$ | 0.6183 | **7085.78** | **24.88** | **0.5452** | 0.4109 | **7241.31** | **25.07** | **0.4934** | 0.2444 | **7511.37** | **25.72** | **0.4540** |
| | $QP_{train}$ | 0.7750 | **7129.44** | **25.02** | **0.5376** | 0.5774 | **7278.13** | **25.18** | **0.4923** | 0.4090 | **7525.87** | **25.81** | **0.4528** |

Table 16: Results of the three **unseen** testing DNNs *at the foundation model level* (ConvNeXtV2-H, CLIP-ViT-B and EVA2-ViT-B) in the 18-model ensemble AllAttacK.

**ConvNeXtV2-$H$**

| Protocol | Attack Method | Best | | | | Mean | | | | Worst | | | |
|---|---|---|---|---|---|---|---|---|---|---|---|---|---|
| | | ASR↑ | $\ell_1\downarrow$ | $\ell_2\downarrow$ | $\ell_\infty\downarrow$ | ASR↑ | $\ell_1\downarrow$ | $\ell_2\downarrow$ | $\ell_\infty\downarrow$ | ASR↑ | $\ell_1\downarrow$ | $\ell_2\downarrow$ | $\ell_\infty\downarrow$ |
| Top-3 | $KL_{test}$ | 0.0424 | **10103.21** | **34.90** | **0.6415** | 0.0100 | 17285.70 | 55.46 | 0.7209 | **0.0011** | 10372.17 | 35.24 | 0.6745 |
| | $KL_{train}$ | 0.0769 | **10209.38** | **34.89** | **0.6547** | 0.0154 | **10209.38** | **34.89** | **0.6547** | 0.0000 | - | - | - |
| | $QP_{test}$ | **0.2176** | 35394.05 | 114.45 | 0.9804 | **0.0493** | 16623.10 | 55.23 | **0.7119** | 0.0000 | - | - | - |
| | $QP_{train}$ | **0.3077** | 36324.36 | 115.92 | 0.9759 | **0.0667** | 23240.93 | 75.25 | 0.8227 | 0.0000 | - | - | - |
| Top-2 | $KL_{test}$ | 0.4464 | 41899.93 | 132.42 | 0.9865 | 0.1384 | 16086.00 | 53.17 | 0.7335 | **0.0279** | 9119.14 | **31.63** | 0.7333 |
| | $KL_{train}$ | 0.4615 | 43572.84 | 133.61 | 0.9702 | 0.1692 | 18115.51 | 58.48 | 0.7302 | 0.0000 | - | - | - |
| | $QP_{test}$ | **0.7545** | **31744.83** | **102.21** | **0.9645** | **0.1944** | **13875.24** | **46.49** | **0.6900** | 0.0045 | 9476.35 | 32.54 | **0.5723** |
| | $QP_{train}$ | 0.6154 | **31929.99** | **102.41** | 0.9563 | 0.1846 | 14904.23 | 49.87 | 0.7209 | 0.0000 | - | - | - |
| Top-1 | $KL_{test}$ | **0.2991** | 9680.39 | 33.58 | 0.5944 | **0.2330** | 9201.79 | 31.88 | 0.6194 | **0.1853** | 9038.93 | 31.48 | 0.6589 |
| | $KL_{train}$ | **0.3077** | 8578.32 | 29.97 | 0.6120 | **0.2359** | 9350.13 | 32.26 | 0.6050 | 0.1282 | 9119.61 | 31.87 | 0.5994 |
| | $QP_{test}$ | 0.1496 | **7121.98** | **24.92** | **0.4596** | 0.1009 | **7319.13** | **25.28** | **0.4935** | 0.0580 | **7390.53** | **25.10** | 0.5143 |
| | $QP_{train}$ | 0.1282 | **7467.81** | **25.73** | **0.4898** | 0.0974 | **7393.86** | **25.53** | 0.5018 | 0.0256 | **7780.77** | **26.49** | **0.4625** |

**CLIP-ViT-B**

| Protocol | Attack Method | Best | | | | Mean | | | | Worst | | | |
|---|---|---|---|---|---|---|---|---|---|---|---|---|---|
| | | ASR↑ | $\ell_1\downarrow$ | $\ell_2\downarrow$ | $\ell_\infty\downarrow$ | ASR↑ | $\ell_1\downarrow$ | $\ell_2\downarrow$ | $\ell_\infty\downarrow$ | ASR↑ | $\ell_1\downarrow$ | $\ell_2\downarrow$ | $\ell_\infty\downarrow$ |
| Top-3 | $KL_{test}$ | **0.0067** | **9927.98** | **33.63** | **0.6158** | **0.0018** | **10174.50** | **34.86** | **0.6276** | **0.0000** | - | - | - |
| | $KL_{train}$ | **0.0000** | - | - | - | **0.0000** | - | - | - | **0.0000** | - | - | - |
| | $QP_{test}$ | 0.0056 | 36719.51 | 116.23 | 0.9765 | **0.0018** | 23385.06 | 75.63 | 0.7780 | **0.0000** | - | - | - |
| | $QP_{train}$ | **0.0000** | - | - | - | **0.0000** | - | - | - | **0.0000** | - | - | - |
| Top-2 | $KL_{test}$ | 0.0379 | **9988.84** | **34.70** | **0.6646** | 0.0210 | 15886.55 | 52.98 | 0.7312 | 0.0100 | 9988.20 | 34.28 | 0.5806 |
| | $KL_{train}$ | 0.0513 | **10015.69** | **34.37** | **0.5898** | 0.0308 | **9909.85** | **34.14** | 0.6777 | 0.0000 | - | - | - |
| | $QP_{test}$ | 0.1886 | 32026.21 | 102.55 | 0.9609 | 0.0922 | **13896.60** | **46.49** | **0.6904** | 0.0145 | **9158.52** | **32.13** | **0.5633** |
| | $QP_{train}$ | 0.2564 | 32339.93 | 102.92 | 0.9484 | **0.1487** | 14002.58 | 46.81 | 0.6792 | 0.0513 | 9527.82 | 33.27 | 0.5441 |
| Top-1 | $KL_{test}$ | **0.5100** | 8724.35 | 30.30 | 0.6337 | **0.3100** | 9137.74 | 31.72 | 0.6186 | **0.1842** | 9660.73 | 33.57 | 0.5878 |
| | $KL_{train}$ | **0.7692** | 8791.55 | 30.39 | 0.6357 | **0.4872** | 9215.21 | 31.95 | 0.6184 | **0.2821** | 9754.42 | 33.84 | 0.5745 |
| | $QP_{test}$ | 0.2254 | **7100.00** | 24.93 | **0.5458** | 0.1587 | **7256.02** | 25.11 | **0.4925** | 0.0982 | **7325.37** | 24.94 | 0.5199 |
| | $QP_{train}$ | 0.3846 | **7460.55** | **25.72** | **0.4512** | 0.2615 | **7321.85** | 25.29 | **0.4943** | 0.1282 | **7491.41** | 25.86 | **0.4962** |

**EVA2-ViT-B**

| Protocol | Attack Method | Best | | | | Mean | | | | Worst | | | |
|---|---|---|---|---|---|---|---|---|---|---|---|---|---|
| | | ASR↑ | $\ell_1\downarrow$ | $\ell_2\downarrow$ | $\ell_\infty\downarrow$ | ASR↑ | $\ell_1\downarrow$ | $\ell_2\downarrow$ | $\ell_\infty\downarrow$ | ASR↑ | $\ell_1\downarrow$ | $\ell_2\downarrow$ | $\ell_\infty\downarrow$ |
| Top-3 | $KL_{test}$ | **0.0000** | - | - | - | **0.0000** | - | - | - | **0.0000** | - | - | - |
| | $KL_{train}$ | **0.0000** | - | - | - | **0.0000** | - | - | - | **0.0000** | - | - | - |
| | $QP_{test}$ | **0.0000** | - | - | - | **0.0000** | - | - | - | **0.0000** | - | - | - |
| | $QP_{train}$ | **0.0000** | - | - | - | **0.0000** | - | - | - | **0.0000** | - | - | - |
| Top-2 | $KL_{test}$ | **0.0000** | - | - | - | **0.0000** | - | - | - | **0.0000** | - | - | - |
| | $KL_{train}$ | **0.0000** | - | - | - | **0.0000** | - | - | - | **0.0000** | - | - | - |
| | $QP_{test}$ | **0.0000** | - | - | - | **0.0000** | - | - | - | **0.0000** | - | - | - |
| | $QP_{train}$ | **0.0000** | - | - | - | **0.0000** | - | - | - | **0.0000** | - | - | - |
| Top-1 | $KL_{test}$ | **0.1998** | 8762.19 | 30.40 | 0.6340 | 0.0560 | 9039.48 | 31.41 | 0.6145 | **0.0000** | - | - | - |
| | $KL_{train}$ | **0.2051** | 8883.08 | 30.57 | 0.6376 | 0.0872 | 9023.45 | 31.48 | 0.6242 | **0.0000** | - | - | - |
| | $QP_{test}$ | 0.1752 | **7279.96** | **24.85** | **0.5244** | 0.0592 | **7358.06** | **25.36** | **0.5035** | **0.0000** | - | - | - |
| | $QP_{train}$ | **0.2051** | 7388.33 | 25.56 | 0.4440 | **0.0923** | 7318.77 | 25.28 | 0.5000 | **0.0000** | - | - | - |

Table 17: Results of previously unseen model set when **seen** during training.

**Swin-B**

| Protocol | Attack Method | Best | | | | Mean | | | | Worst | | | |
|---|---|---|---|---|---|---|---|---|---|---|---|---|---|
| | | ASR↑ | $\ell_1$↓ | $\ell_2$↓ | $\ell_\infty$↓ | ASR↑ | $\ell_1$↓ | $\ell_2$↓ | $\ell_\infty$↓ | ASR↑ | $\ell_1$↓ | $\ell_2$↓ | $\ell_\infty$↓ |
| Top-3 | $QP_{test}$ | 0.9990 | 24540.19 | 79.65 | 0.8952 | 0.9974 | 25110.71 | 81.46 | 0.9118 | 0.9950 | 23007.00 | 75.03 | 0.8986 |
| | $QP_{train}$ | 1.0000 | 24536.30 | 79.64 | 0.8974 | 1.0000 | 25110.62 | 81.45 | 0.9130 | 1.0000 | 24536.30 | 79.64 | 0.8974 |
| | $KL_{test}$ | 0.9720 | 29787.00 | 96.13 | 0.9542 | 0.9350 | 32452.50 | 104.13 | 0.9700 | 0.8980 | 32640.93 | 104.66 | 0.9732 |
| | $KL_{train}$ | 0.9870 | 29820.40 | 96.16 | 0.9541 | 0.9532 | 32411.51 | 104.05 | 0.9702 | 0.9190 | 32579.29 | 104.54 | 0.9728 |
| Top-2 | $QP_{test}$ | 0.9980 | 23136.34 | 76.05 | 0.9286 | 0.9972 | 23314.75 | 76.01 | 0.9084 | 0.9960 | 22281.68 | 72.39 | 0.8869 |
| | $QP_{train}$ | 1.0000 | 22275.55 | 72.31 | 0.8848 | 0.9998 | 23296.73 | 75.97 | 0.9074 | 0.9990 | 23009.64 | 75.82 | 0.9302 |
| | $KL_{test}$ | 0.9960 | 29431.99 | 94.62 | 0.9541 | 0.9798 | 30468.20 | 98.02 | 0.9616 | 0.9350 | 32060.72 | 102.97 | 0.9681 |
| | $KL_{train}$ | 1.0000 | 31233.87 | 100.59 | 0.9666 | 0.9858 | 30435.74 | 97.98 | 0.9615 | 0.9450 | 31977.27 | 102.83 | 0.9678 |
| Top-1 | $QP_{test}$ | 1.0000 | 26403.61 | 85.39 | 0.9125 | 0.9994 | 27434.66 | 88.55 | 0.9256 | 0.9971 | 28208.06 | 91.11 | 0.9412 |
| | $QP_{train}$ | 1.0000 | 26340.53 | 85.33 | 0.9138 | 1.0000 | 27416.17 | 88.56 | 0.9255 | 1.0000 | 26340.53 | 85.33 | 0.9138 |
| | $KL_{test}$ | 1.0000 | 23104.45 | 75.70 | 0.8816 | 0.9996 | 22164.69 | 72.41 | 0.8749 | 0.9990 | 19679.64 | 65.03 | 0.8373 |
| | $KL_{train}$ | 1.0000 | 19694.63 | 65.03 | 0.8362 | 1.0000 | 22156.36 | 72.39 | 0.8744 | 1.0000 | 19694.63 | 65.03 | 0.8362 |

**HRNet-W30**

| Protocol | Attack Method | Best | | | | Mean | | | | Worst | | | |
|---|---|---|---|---|---|---|---|---|---|---|---|---|---|
| | | ASR↑ | $\ell_1$↓ | $\ell_2$↓ | $\ell_\infty$↓ | ASR↑ | $\ell_1$↓ | $\ell_2$↓ | $\ell_\infty$↓ | ASR↑ | $\ell_1$↓ | $\ell_2$↓ | $\ell_\infty$↓ |
| Top-3 | $QP_{test}$ | 0.9880 | 23168.02 | 75.45 | 0.9293 | 0.9810 | 21138.58 | 69.36 | 0.9316 | 0.9660 | 21865.56 | 71.53 | 0.9362 |
| | $QP_{train}$ | 0.9990 | 21950.48 | 72.15 | 0.9240 | 0.9978 | 21116.67 | 69.33 | 0.9324 | 0.9970 | 19429.58 | 64.21 | 0.9302 |
| | $KL_{test}$ | 0.7380 | 28115.32 | 90.56 | 0.9676 | 0.6282 | 27281.20 | 87.64 | 0.9562 | 0.5220 | 26862.44 | 86.68 | 0.9547 |
| | $KL_{train}$ | 0.8300 | 28205.96 | 90.70 | 0.9675 | 0.7888 | 27313.62 | 87.69 | 0.9580 | 0.7180 | 27603.06 | 88.17 | 0.9603 |
| Top-2 | $QP_{test}$ | 0.9912 | 19796.01 | 64.54 | 0.9136 | 0.9851 | 19159.13 | 62.43 | 0.8903 | 0.9722 | 15574.23 | 51.32 | 0.8651 |
| | $QP_{train}$ | 1.0000 | 15481.31 | 51.16 | 0.8700 | 0.9996 | 19112.94 | 62.38 | 0.8916 | 0.9979 | 21513.55 | 69.81 | 0.8845 |
| | $KL_{test}$ | 0.9210 | 24272.99 | 78.93 | 0.9472 | 0.8694 | 26127.12 | 84.02 | 0.9507 | 0.7830 | 25212.21 | 80.87 | 0.9401 |
| | $KL_{train}$ | 0.9280 | 24268.70 | 78.89 | 0.9481 | 0.8810 | 26133.35 | 84.03 | 0.9505 | 0.8210 | 25273.17 | 80.97 | 0.9393 |
| Top-1 | $QP_{test}$ | 0.9985 | 18562.94 | 60.49 | 0.8946 | 0.9953 | 18590.65 | 60.32 | 0.8541 | 0.9927 | 21124.19 | 68.02 | 0.8749 |
| | $QP_{train}$ | 1.0000 | 15454.53 | 51.04 | 0.8367 | 1.0000 | 18558.31 | 60.30 | 0.8567 | 1.0000 | 15454.53 | 51.04 | 0.8367 |
| | $KL_{test}$ | 0.9990 | 16755.41 | 55.35 | 0.8885 | 0.9984 | 16784.70 | 55.46 | 0.8765 | 0.9970 | 15941.66 | 53.22 | 0.8801 |
| | $KL_{train}$ | 1.0000 | 15956.82 | 53.27 | 0.8753 | 1.0000 | 16772.27 | 55.45 | 0.8764 | 1.0000 | 15956.82 | 53.27 | 0.8753 |

**ConvMixer-768**

| Protocol | Attack Method | Best | | | | Mean | | | | Worst | | | |
|---|---|---|---|---|---|---|---|---|---|---|---|---|---|
| | | ASR↑ | $\ell_1$↓ | $\ell_2$↓ | $\ell_\infty$↓ | ASR↑ | $\ell_1$↓ | $\ell_2$↓ | $\ell_\infty$↓ | ASR↑ | $\ell_1$↓ | $\ell_2$↓ | $\ell_\infty$↓ |
| Top-3 | $QP_{test}$ | 0.9956 | 27666.59 | 90.09 | 0.9533 | 0.9933 | 26984.55 | 87.62 | 0.9427 | 0.9912 | 25219.01 | 82.74 | 0.9415 |
| | $QP_{train}$ | 1.0000 | 28116.82 | 90.43 | 0.9447 | 0.9990 | 26952.49 | 87.61 | 0.9431 | 0.9959 | 25199.53 | 82.72 | 0.9380 |
| | $KL_{test}$ | 0.8620 | 28086.21 | 90.09 | 0.9237 | 0.7522 | 29261.82 | 94.18 | 0.9511 | 0.6490 | 29377.40 | 94.58 | 0.9531 |
| | $KL_{train}$ | 0.9120 | 28138.44 | 90.16 | 0.9223 | 0.8282 | 29283.29 | 94.21 | 0.9513 | 0.7540 | 29345.09 | 94.49 | 0.9551 |
| Top-2 | $QP_{test}$ | 0.9971 | 21134.29 | 69.68 | 0.8719 | 0.9947 | 24096.42 | 78.38 | 0.9073 | 0.9912 | 28750.18 | 92.31 | 0.9367 |
| | $QP_{train}$ | 1.0000 | 20292.20 | 66.82 | 0.8808 | 0.9990 | 24069.73 | 78.36 | 0.9066 | 0.9959 | 26091.54 | 84.71 | 0.9406 |
| | $KL_{test}$ | 0.9540 | 26778.85 | 86.45 | 0.9436 | 0.9238 | 27857.19 | 89.67 | 0.9437 | 0.9060 | 27665.11 | 89.29 | 0.9508 |
| | $KL_{train}$ | 0.9470 | 26759.37 | 86.39 | 0.9459 | 0.9144 | 27883.61 | 89.71 | 0.9435 | 0.8870 | 30007.15 | 96.24 | 0.9579 |
| Top-1 | $QP_{test}$ | 1.0000 | 22449.09 | 72.89 | 0.8889 | 0.9985 | 21496.57 | 70.00 | 0.8644 | 0.9971 | 20309.03 | 66.53 | 0.8643 |
| | $QP_{train}$ | 1.0000 | 20157.37 | 65.77 | 0.8436 | 1.0000 | 21467.20 | 69.68 | 0.8651 | 1.0000 | 20157.37 | 65.77 | 0.8436 |
| | $KL_{test}$ | 1.0000 | 18248.97 | 59.50 | 0.8337 | 0.9994 | 17103.97 | 56.55 | 0.8173 | 0.9980 | 14218.59 | 47.78 | 0.7766 |
| | $KL_{train}$ | 1.0000 | 14223.16 | 47.77 | 0.7754 | 1.0000 | 17089.81 | 56.52 | 0.8157 | 1.0000 | 14223.16 | 47.77 | 0.7754 |

**CLIP-ViT-B**

| Protocol | Attack Method | Best | | | | Mean | | | | Worst | | | |
|---|---|---|---|---|---|---|---|---|---|---|---|---|---|
| | | ASR↑ | $\ell_1$↓ | $\ell_2$↓ | $\ell_\infty$↓ | ASR↑ | $\ell_1$↓ | $\ell_2$↓ | $\ell_\infty$↓ | ASR↑ | $\ell_1$↓ | $\ell_2$↓ | $\ell_\infty$↓ |
| Top-3 | $QP_{test}$ | 1.0000 | 34654.59 | 114.18 | 0.9842 | 0.9997 | 35112.00 | 115.42 | 0.9842 | 0.9985 | 34728.69 | 115.56 | 0.9857 |
| | $QP_{train}$ | 1.0000 | 34578.89 | 114.16 | 0.9858 | 0.9996 | 35090.07 | 115.42 | 0.9843 | 0.9990 | 34769.59 | 115.62 | 0.9860 |
| | $KL_{test}$ | 0.9020 | 49293.40 | 154.16 | 0.9946 | 0.4080 | 46355.60 | 152.15 | 0.9967 | 0.0010 | 49052.12 | 154.53 | 1.0000 |
| | $KL_{train}$ | 0.9110 | 49259.36 | 154.20 | 0.9947 | 0.4068 | 46254.19 | 151.42 | 0.9936 | 0.0000 | - | - | - |
| Top-2 | $QP_{test}$ | 1.0000 | 26357.61 | 89.12 | 0.9614 | 0.9991 | 28446.28 | 95.32 | 0.9719 | 0.9985 | 27916.95 | 93.95 | 0.9686 |
| | $QP_{train}$ | 1.0000 | 26284.29 | 89.06 | 0.9624 | 0.9994 | 28422.88 | 95.35 | 0.9721 | 0.9990 | 27947.58 | 94.10 | 0.9701 |
| | $KL_{test}$ | 0.9980 | 44208.61 | 140.39 | 0.9918 | 0.6426 | 44164.83 | 141.67 | 0.9951 | 0.0180 | 43464.04 | 140.91 | 0.9989 |
| | $KL_{train}$ | 1.0000 | 44281.64 | 140.47 | 0.9917 | 0.6366 | 44018.60 | 141.76 | 0.9950 | 0.0090 | 42664.03 | 141.20 | 0.9991 |
| Top-1 | $QP_{test}$ | 1.0000 | 27269.24 | 88.93 | 0.9514 | 0.9992 | 26705.81 | 86.93 | 0.9342 | 0.9980 | 22323.44 | 72.22 | 0.8475 |
| | $QP_{train}$ | 1.0000 | 27282.61 | 88.96 | 0.9496 | 0.9996 | 26681.01 | 86.88 | 0.9335 | 0.9980 | 22368.83 | 72.29 | 0.8443 |
| | $KL_{test}$ | 1.0000 | 25907.21 | 85.20 | 0.9533 | 1.0000 | 26343.30 | 86.47 | 0.9446 | 1.0000 | 25907.21 | 85.20 | 0.9533 |
| | $KL_{train}$ | 1.0000 | 25945.80 | 85.28 | 0.9513 | 1.0000 | 26335.81 | 86.45 | 0.9436 | 1.0000 | 25945.80 | 85.28 | 0.9513 |

**EVA2-ViT-B**

| Protocol | Attack Method | Best | | | | Mean | | | | Worst | | | |
|---|---|---|---|---|---|---|---|---|---|---|---|---|---|
| | | ASR↑ | $\ell_1$↓ | $\ell_2$↓ | $\ell_\infty$↓ | ASR↑ | $\ell_1$↓ | $\ell_2$↓ | $\ell_\infty$↓ | ASR↑ | $\ell_1$↓ | $\ell_2$↓ | $\ell_\infty$↓ |
| Top-3 | $QP_{test}$ | 0.9990 | 21314.92 | 69.67 | 0.9225 | 0.9954 | 19896.05 | 65.16 | 0.8793 | 0.9920 | 18538.02 | 61.01 | 0.8512 |
| | $QP_{train}$ | 1.0000 | 21309.14 | 69.66 | 0.9218 | 0.9986 | 19878.14 | 65.12 | 0.8798 | 0.9950 | 19260.00 | 62.87 | 0.8457 |
| | $KL_{test}$ | 0.8170 | 27769.76 | 88.80 | 0.9440 | 0.7714 | 30104.63 | 96.12 | 0.9593 | 0.7350 | 27555.40 | 88.37 | 0.9419 |
| | $KL_{train}$ | 0.9300 | 27676.84 | 88.59 | 0.9439 | 0.8798 | 30139.58 | 96.14 | 0.9581 | 0.8490 | 27771.35 | 88.66 | 0.9382 |
| Top-2 | $QP_{test}$ | 0.9956 | 16364.13 | 53.88 | 0.7936 | 0.9942 | 16712.75 | 54.84 | 0.8048 | 0.9927 | 16992.77 | 55.60 | 0.8109 |
| | $QP_{train}$ | 1.0000 | 16917.59 | 55.49 | 0.8146 | 0.9990 | 16678.32 | 54.81 | 0.8052 | 0.9959 | 16347.58 | 53.88 | 0.7964 |
| | $KL_{test}$ | 0.9780 | 27044.71 | 86.89 | 0.9488 | 0.9594 | 25446.25 | 81.90 | 0.9236 | 0.9370 | 24641.59 | 79.20 | 0.9185 |
| | $KL_{train}$ | 0.9910 | 24015.74 | 77.20 | 0.8814 | 0.9732 | 25446.83 | 79.07 | 0.9196 | 0.9650 | 24598.83 | 79.07 | 0.9196 |
| Top-1 | $QP_{test}$ | 1.0000 | 18853.34 | 62.10 | 0.8375 | 0.9994 | 16412.60 | 54.32 | 0.7796 | 0.9985 | 13582.94 | 44.77 | 0.6327 |
| | $QP_{train}$ | 1.0000 | 18792.10 | 62.01 | 0.8329 | 1.0000 | 16382.95 | 54.30 | 0.7805 | 1.0000 | 18792.10 | 62.01 | 0.8329 |
| | $KL_{test}$ | 0.9970 | 13449.05 | 44.75 | 0.6954 | 0.9940 | 10806.82 | 35.79 | 0.5983 | 0.9890 | 9571.74 | 31.47 | 0.4424 |
| | $KL_{train}$ | 1.0000 | 11484.26 | 37.94 | 0.5829 | 1.0000 | 10797.96 | 35.77 | 0.5969 | 1.0000 | 11484.26 | 37.94 | 0.5829 |

**ConvNeXtV2-H**

| Protocol | Attack Method | Best | | | | Mean | | | | Worst | | | |
|---|---|---|---|---|---|---|---|---|---|---|---|---|---|
| | | ASR↑ | $\ell_1$↓ | $\ell_2$↓ | $\ell_\infty$↓ | ASR↑ | $\ell_1$↓ | $\ell_2$↓ | $\ell_\infty$↓ | ASR↑ | $\ell_1$↓ | $\ell_2$↓ | $\ell_\infty$↓ |
| Top-3 | $QP_{test}$ | 0.9980 | 24945.29 | 81.07 | 0.9354 | 0.9932 | 27163.90 | 88.36 | 0.9520 | 0.9770 | 22306.51 | 74.64 | 0.9473 |
| | $QP_{train}$ | 1.0000 | 22155.59 | 74.44 | 0.9519 | 0.9996 | 27137.74 | 88.32 | 0.9528 | 0.9990 | 24936.93 | 81.06 | 0.9366 |
| | $KL_{test}$ | 0.9670 | 31722.42 | 101.31 | 0.9534 | 0.9454 | 29654.27 | 95.36 | 0.9540 | 0.9110 | 31060.00 | 99.65 | 0.9638 |
| | $KL_{train}$ | 0.9980 | 31715.49 | 101.27 | 0.9511 | 0.9898 | 29587.92 | 95.24 | 0.9540 | 0.9740 | 31056.11 | 99.58 | 0.9621 |
| Top-2 | $QP_{test}$ | 0.9990 | 29030.98 | 93.98 | 0.9649 | 0.5974 | 27490.60 | 88.93 | 0.9452 | 0.0000 | - | - | - |
| | $QP_{train}$ | 1.0000 | 28284.06 | 90.78 | 0.9484 | 0.5998 | 27446.39 | 88.88 | 0.9456 | 0.0000 | - | - | - |
| | $KL_{test}$ | 0.9920 | 34999.71 | 110.47 | 0.9734 | 0.9548 | 27633.93 | 88.80 | 0.9306 | 0.8530 | 27379.13 | 89.63 | 0.9418 |
| | $KL_{train}$ | 1.0000 | 34976.64 | 110.32 | 0.9727 | 0.9712 | 27608.80 | 88.72 | 0.9305 | 0.8670 | 27445.30 | 89.72 | 0.9421 |
| Top-1 | $QP_{test}$ | 1.0000 | 35972.03 | 115.01 | 0.9846 | 0.9982 | 31676.12 | 101.80 | 0.9694 | 0.9927 | 27356.92 | 88.33 | 0.9538 |
| | $QP_{train}$ | 1.0000 | 27287.05 | 88.28 | 0.9556 | 1.0000 | 31669.78 | 101.83 | 0.9696 | 1.0000 | 27287.05 | 88.28 | 0.9556 |
| | $KL_{test}$ | 1.0000 | 26136.14 | 84.45 | 0.9252 | 0.8000 | 23033.74 | 74.43 | 0.8774 | 0.0000 | - | - | - |
| | $KL_{train}$ | 1.0000 | 26154.14 | 84.44 | 0.9246 | 0.8000 | 23008.02 | 74.37 | 0.8766 | 0.0000 | - | - | - |

