# OpenReview forum: "Revisiting and Expanding Targeted Universal Adversarial Perturbations"
_ICLR.cc/2025/Conference — ICLR 2025 Conference Withdrawn Submission_

### Official Review · Reviewer_o9Xu · 2024-11-03

**Soundness:** 3
**Presentation:** 3
**Contribution:** 3
**Rating:** 6
**Confidence:** 4

**Summary:**

The paper proposes white-box target UAP attacks that are transferable across models and target images. They explore the challenging attack setting to fool the attacked network into predicting a pre-determined sequence of targeted classes. They formulate two optimization objectives to achieve this, one minimizing KL divergence of model predictions with the target distribution, and the other utilizing quadratic programming to learn the UAP. The approaches are extensively evaluated on various models and configurations.

**Strengths:**

1) The paper is fairly well written, the problem setup is explained in a sufficient level of detail.

2) The experiments are exhaustive and evaluate a variety of model architectures. Several interesting insights are highlighted in the paper, that are useful for the research community.

**Weaknesses:**

1) The paper does not compare their proposed attack against any existing methods. Although the existing methods for creating targeted UAPs are listed under references, there is no actual reference to them in the problem formulation or experiments. Please explain what differentiates the design choices of AllAttacK from these approaches. It is also acceptable to modify the existing methods to fairly evaluate against AllAttacK.

2) What is meant by the 'best', 'worst', and 'mean' cases in the experiments? The description in the text does not make sense ('best' and 'worst' have the same word-to-word explanation, and the sentence doesn't make sense).

3) There are several typos in the paper, including grammatical errors. Some instances include:

     a) The last line of the abstract.

     b) The repeated mention of the "shear complexity of ...". What does this mean? Assuming the intended word is "sheer"?

     c) Line 176 "images for each in ImageNet1k". For each what?

     d) Line 164, "axies".

**Questions:**

Refer to weaknesses. Mainly, existing works should be included in experimental evaluations (at least under the key experimental setups), and the grammatical and typographical errors need correction to improve readability.

---

### Official Review · Reviewer_NMbH · 2024-11-04

**Soundness:** 3
**Presentation:** 3
**Contribution:** 2
**Rating:** 5
**Confidence:** 4

**Summary:**

A paper provides a very rigorous and scrutiny approach to test the leading ordered top-K Universal Adversarial Perturbations methods on top of multiple architectures and even ensembles. This approach overall guarantees the attack (called AllAttackK) efficiency across models, data, and targets.

**Strengths:**

Main strengths are:
1. Scrutinized testing across models architectures (convolutional, MLP, and ViT-based ones)
2. Single-model and ensemble testing
4. One of the hardest set ups: universal ordered top-K adversarial attack
3. Good ASR across multiple testing settings

**Weaknesses:**

Major thoughts I have after reading the paper:
1. The paper's impact is mostly based on using the known concepts (UAP, Ordered Top-K) and techniques (KL-based, QP-based adversarial generation methods), so that's why we can see a lot of great and rigorous empirical work. In this case, the reader expects to find the answers that were somehow raised in this exploration:
* Why QP-based method is generally working better than KL-based for $K>1$ ? There is an answer about the behavior in case of $K=1$, but nothing about the former case.
* In Table 3, for the test models (last 6 rows) we should have approximately similar ASR values for both $D_{test}$ and $D_{train}$ (because these models were not used for the training at all, we are testing the transferability of the method). Why it is not the case?
2. There are some words about the trade-off between ASR and the norm of perturbations (e.g., in Section 3.2.3, Table 4). While the hypothesis behind it seems valid, in order to justify it the another set of experiments should be done: fix the max norm of perturbations and compare ASRs. In the paper (nor in the Appendix) I haven't found such experiments
3. There is a strong statement from the Conclusion Section (Lines 535-537) that "perturbations themselves in isolation will be classified by the DNN with the top-K predictions equal to the ordered top-K targets (K ≥ 1)", but actually there were no any quantitative check of this statement through the paper text.

Minor remarks:
* a. Line 181: $T \notin \hat{Y}_w(x_i)$ should be something like $T \neq \hat{Y}_w(x_i)[1:K]$ ?
* b. For KL-based criteria, I guess we should use only probabilities induced by $T$ indices, so e.g. the Eq. (10) (Line 230) should be somehow modified so we are not using the full $\hat{P}_{\Theta_s}(x'_i)$, but only the indices from $T$?
* c. There is statement that for ensemble-based UAPs are looking more "semantically meaningful" (Line 491), but no any further discussion on it and its reasons
* d. Lines 793-794: missed the Eq. numbers (referring the main paper)

**Questions:**

A minor question about the calculation of 510 UAP per training. It is understood the first part of the calculation (5 runs per 7 models per 6 K per 2 methods) while for the second one it should be (5 runs per 1 model/ensemble per 3 K per 2 methods) that is different from the provided (15 * 3 * 2). Would highly appreciate the clarification of it.

---

### Official Review · Reviewer_Byrg · 2024-11-05

**Soundness:** 2
**Presentation:** 2
**Contribution:** 3
**Rating:** 6
**Confidence:** 5

**Summary:**

This paper introduces a new attack called AllAttack, as a targeted universal adversarial perturbation against DNNs.
Contribution wise, they consider multiple model architectures CNN, ViT, CLIP, MLP mixers; They consider a large scale of unseen data; They consider multiple targets to find the easiest target label.

**Strengths:**

1. As concluded in summary, their contributions are convincing to me. This is a complete paper;
2. Mini-data-batch and mini-model-batch strategies solve the challenges of universal attacks to some extent. This direction of optimization can be helpful for the community when considering adversarial attack;
3. Comprehensive examination of their proposal. Adversarial attacks have been a classic topic and authors organize them in a comprehensive way in exp part. I enjoy their contributions in experiments and demonstration.

**Weaknesses:**

1. As an ICLR paper, the demonstration is super important. I have stated that the illustration is good in strengths. However, some too small fonts make my eyes exhausted. Figure 1 is fine; figure 2 x, y axis marks are too small; Table 2 headers are no way that I can see clearly; Table 3-5 for me is fine but I don't think they are OK for other reviewers;
2. Please try your UAP on Multi-modality model. For example, feed your image into chat gpt in a black-box manner, will they succeed?

**Questions:**

See weakness.

---

### Official Review · Reviewer_XSXA · 2024-11-08

**Soundness:** 3
**Presentation:** 3
**Contribution:** 2
**Rating:** 5
**Confidence:** 4

**Summary:**

- The draft presents a method (AllAttacK) for learning universal ordered Top-K targeted adversarial perturbations that are both image-agnostic and model-agnostic under the white-box attack setting.
- The draft considers the study of targeted universal adversarial attacks along three axes (model, data, and targets). Built on previous single-model and instance-specific ordered Top-K attack methods, the proposed method presents two optimization methods for learning AllAttacK.
- The proposed attack enables training with disparate deep neural networks (stochastic mini-data-batch and mini-model-batch optimization strategy). It is evaluated in experiments with several universal ordered top-K perturbations.
- Experiments are conducted on the ImageNet-1K dataset considering multiple DNN classifiers, and the performance of the proposed attack is reported.

**Strengths:**

- The draft considers many models (probably the first of such an effort) while crafting the image and model agnostic, ordered top-K UAPs. Large-scale studies can help us better assess the (worst-case) robustness of the DNN classifiers.
- The presentation of the draft contents is comfortable to read and appreciate.
- The proposed method adapts mini-batch optimization to sample models, known as mini-model optimization (since full-batch model optimization needs huge GPU memory). It is a simple technique to handle a pool of disparate DNN models (although no theoretical analysis has been presented about the equivalence of the full-model optimization).
- Experiments considered 18 disparate DNN classifiers trained on the ImageNet-1K (probably the most so far in published research). Ablations are performed to study the robustness of the adversarially trained models against the proposed attack.
- The authors mentioned that the code would be released to help reproduce the results.

**Weaknesses:**

- Although the proposed method presents one of the hardest adversarial attacks, it is built on two existing (relatively simpler) adversarial attacks.
- The main issue with the draft is the ineffectiveness of the crafted UAPs. The draft claims (line 76) that the proposed method crafts a doubly transferable ordered top-K UAPs. However, the Attack Success Rate (ASR) presented in the experiments is not convincing. In particular, Table 3 demonstrates numbers that are not very impressive when attacking unknown (test) models. Some numbers are close to zero (despite crafting on 18 disparate DNN models). Even the ASR for some of the known (train) models also is very low (e.g., MLPMixer).
- A good number of entries in (Tables 2 and 3) do not support the paper's main claim that it crafts a transferable ordered top-K attack.

**Questions:**

- Authors can clarify the discrepancy between the claims and the experimental results (as mentioned in the weaknesses section).

---

### Official Review · Reviewer_vZjp · 2024-11-11

**Soundness:** 3
**Presentation:** 3
**Contribution:** 2
**Rating:** 5
**Confidence:** 3

**Summary:**

This paper considered universal adversarial perturbations (UAPs) and expands conventional the white-box targeted attacks in 3 aspects: models, data and target. Specifically, the authors propose a more challenging setup called AllAttacK, which considers ordered top-K attack protocal and doubly-transferrable to testing models and testing Images. Furthermore, two optimization formulations are proposed for learninig AllAttacK. Experiments are conducted on ImageNet with various models to evaluated the proposed algorithm.

**Strengths:**

1. This paper presents a novel setting of UAPs on the axes of models, data and targets. It provides the first large-scale study of learning data-agnoistic and model-agnostic UAPs.
2. The proposed problem of AllAttacK is challenging and the observation of Figure 2 of learned perturbations is interesting.
3. The presentation is clear and the paper is eazy to understand.
4. The author conducts comprehensive experiments on 18 models with different $K$.

**Weaknesses:**

1. Lack of explanantion of problem formulation of AllAttacK. The authors extend UAPs through 3 dimensions: model, data and target, resulting in the challenging formulation of AllAttacK. However, I believe more explanation are needed on 1) why we need to consider such challenging setup, especially the ordered top-K targeted attacks? What would be the benefit comapred to vallina UAPs? 2) On the model and data dimension, what's the exact difference between AllAttacK and UAPs?

2. The optimization of proposed formulation seems expensive. To solve the challenging model-agnostic and data-agnostic UAPs, the author proposed two optmization formulations (Eq. 9 and Eq. 14). However, both of them requires to train on full training images and training models, which is an combination of $|D^{train}| * |M^{train}|$ examples. I'm wondering the exact computation cost of AllAttacK. How long do one need to train till convergence.

3. The evaluation is limited given the challenging setup. The problem considers data-agnostic and model-agnostic ordered top-K targeted attacks. However, 1) the proposed algorithm is only evaluated on ImageNet, how would it work on other datasets? 2) In the final experiments, 18 models are used for training and 6 unseen models are used for test, how to ensure the empirical conclusion is generalizable to other model architectures? 3) For top-K target attacks, what is the performance variance given different combination of targets?

4. Lack of baselines. How do existing methods work on this setup? More analysis is needed to know on which part the proposed optimization formulations would improve compared to existing methods.

**Questions:**

Please refer to the "Weakness".

---

### Note · Authors · 2024-11-15

**Comment:**

We thank all the reviewers' comments and appreciate their valuable time. We work on continually improving this work.

**Withdrawal Confirmation:**

I have read and agree with the venue's withdrawal policy on behalf of myself and my co-authors.